# Altered visual cortex excitatory/inhibitory ratio following transient congenital visual deprivation in humans

**Rashi Pant[1]\*, Kabilan Pitchaimuthu[1,2], José P Ossandón[1], Idris Shareef[3,4], Sunitha Lingareddy[5], Jürgen Finsterbusch[6], Ramesh Kekunnaya[3], Brigitte Röder[1,3]**

[1]Biological Psychology and Neuropsychology, University of Hamburg, Hamburg, Germany; [2]Department of Medicine and Optometry, Linnaeus University, Kalmar, Sweden; [3]Child Sight Institute, Jasti V Ramanamma Children's Eye Care Centre, LV Prasad Eye Institute, Hyderabad, India; [4]Department of Psychology, University of Nevada, Reno, United States; [5]LUCID Medical Diagnostics, Hyderabad, India; [6]Institute of Systems Neuroscience, University Medical Center Hamburg-Eppendorf, Hamburg, Germany

**\*For correspondence:**
rashi.pant@uni-hamburg.de

## eLife Assessment

This neuroimaging and electrophysiology study in a small cohort of congenital cataract patients with sight recovery aims to characterize the effects of early visual deprivation on excitatory and inhibitory balance in visual cortex. While contrasting sight-recovery with visually intact controls suggested the existence of persistent alterations in Glx/GABA ratio and aperiodic EEG signals, it provided **incomplete** evidence supporting claims about the effects of early deprivation itself. The reported data were considered **valuable**, given the rare study population. However, methodological limitations will likely restrict usefulness to scientists working in this particular subfield.

**Abstract** Non-human animal models have indicated that the ratio of excitation to inhibition (E/I) in neural circuits is experience dependent, and changes across development. Here, we assessed 3T Magnetic Resonance Spectroscopy (MRS) and electroencephalography (EEG) markers of cortical E/I ratio in 10 individuals who had been treated for dense bilateral congenital cataracts, after an average of 12 years of blindness, to test for dependence of the E/I ratio on early visual experience in humans. First, participants underwent MRS scanning at rest with their eyes open and eyes closed, to obtain visual cortex Gamma-Aminobutyric Acid (GABA+) concentration, Glutamate/Glutamine (Glx) concentration, and the concentration ratio of Glx/GABA+, as measures of inhibition, excitation, and E/I ratio, respectively. Subsequently, EEG was recorded to assess aperiodic activity (1–20 Hz) as a neurophysiological measure of the cortical E/I ratio, during rest with eyes open and eyes closed, and during flickering stimulation. Across conditions, congenital cataract-reversal individuals demonstrated a significantly lower visual cortex Glx/GABA+ ratio, and a higher intercept and steeper aperiodic slope at occipital electrodes, compared to age-matched sighted controls. In the congenital cataract-reversal group, a lower Glx/GABA+ ratio was associated with better visual acuity, and Glx concentration correlated positively with the aperiodic intercept in the conditions with visual input. We speculate that these findings result from an increased E/I ratio of the visual cortex as a consequence of congenital blindness, which might require commensurately increased inhibition in order to balance the additional excitation from restored visual input. The lower E/I ratio in congenital cataract-reversal individuals would thus be a consequence of homeostatic plasticity.

## Introduction

Sensitive periods are epochs during the lifespan within which effects of experience on the brain are particularly strong (*Knudsen, 2004*). Non-human animal work has established that structural remodelling (*Bourgeois, 1997*) and the development of local inhibitory neural circuits strongly link to the timing of sensitive periods (*Gianfranceschi et al., 2003*; *Hensch et al., 1998*; *Hensch and Bilimoria, 2012*; *Hensch and Fagiolini, 2005b*; *Takesian and Hensch, 2013*). Early visual experience has been shown to fine-tune local inhibitory circuits (*Benevento et al., 1992*; *Chattopadhyaya et al., 2004*; *Gandhi et al., 2008*; *Toyoizumi et al., 2013*), which dynamically control feedforward excitation (*Tao and Poo, 2005*; *Wu et al., 2022*). The end of the sensitive period has been proposed to coincide with the maturation of inhibitory neural circuits (*Hensch, 2005a*; *Wong-Riley, 2021*; *Zhang et al., 2018*). Within this framework, neural circuit stability following sensitive periods is maintained via a balance between excitatory and inhibitory transmission across multiple spatiotemporal scales (*Froemke, 2015*; *Haider et al., 2006*; *Maffei et al., 2004*; *Takesian and Hensch, 2013*; *Wu et al., 2022*). Such an excitatory/inhibitory (E/I) ratio has been studied at different organizational levels, including the synaptic and neuronal levels, as well as for neural circuits (*van Vreeswijk and Sompolinsky, 1996*; *Wu et al., 2022*).

The experience-dependence of local inhibitory circuit tuning in early development is supported by a large body of work in non-human animals. In particular, studies of the mouse visual cortex demonstrated a disrupted tuning of local inhibitory circuits as a consequence of lacking visual experience at birth (*Hensch and Fagiolini, 2005b*; *Levelt and Hübener, 2012*). In addition, dark-reared mice have been shown to have increased spontaneous neural firing in adulthood (*Benevento et al., 1992*) and a reduced magnitude of inhibition, particularly in layers II/III of the visual cortex (*Morales et al., 2002*), suggesting an overall higher level of excitation.

Human neuroimaging studies have similarly demonstrated that visual experience during the first weeks and months of life is crucial for the development of visual neural circuits (*Baroncelli et al., 2011*; *Lewis and Maurer, 2005*; *Maurer and Hensch, 2012*; *Röder et al., 2021*; *Röder et al., 2021*; *Singh et al., 2018*). As studies manipulating visual experience are impossible in human research, much of our understanding of the experience-dependence of visual circuit development comes from patients who underwent a transient period of congenital blindness due to dense bilateral congenital cataracts. If human infants born with dense bilateral cataracts are treated later than a few weeks from birth, they suffer from a permanent reduction of visual acuity (*Birch et al., 1998*; *Khanna et al., 2013*), stereovision (*Birch et al., 1993*; *Tytla et al., 1993*), and impairments in higher level visual functions such as face perception (*Le Grand et al., 2001*; *Putzar et al., 2010*; *Röder et al., 2013*), coherent motion detection (*Bottari et al., 2018*; *Hadad et al., 2012*; *Maurer and Lewis, 2018*), visual temporal processing (*Badde et al., 2020*), and visual feature binding (*McKyton et al., 2015*; *Putzar et al., 2007*). These visual deficits in congenital cataract-reversal individuals have been attributed to altered neural development due to the absence of early visual experience, as individuals who suffered from developmental cataracts do not typically display a comparable severity of visual impairments (*Lewis and Maurer, 2009*; *Sourav et al., 2020*). While the extant literature has reported correlations between structural changes and behavioral outcomes in congenital cataract-reversal individuals (*Feng et al., 2021*; *Guerreiro et al., 2015*; *Hölig et al., 2023*; *Pedersini et al., 2023*), functional brain imaging (*Heitmann et al., 2023*; *Rączy et al., 2022*) and electrophysiological research (*Bottari et al., 2016*; *Ossandón et al., 2023*; *Pant et al., 2023*; *Pitchaimuthu et al., 2021*) have started to unravel the neural mechanisms which rely on visual experience during early brain development.

Resting-state activity measured via fMRI suggested increased excitation in the visual cortex of congenital cataract-reversal individuals (*Rączy et al., 2022*): The amplitude of low frequency (<1 Hz; blood oxygen level-dependent) fluctuations (ALFF) in the visual cortex was increased in congenital cataract-reversal individuals compared to normally sighted controls when they were scanned with their eyes open. Since similar changes were observed in permanently congenitally blind humans, the authors speculated that congenital visual deprivation resulted in an increased E/I ratio of neural circuits due to impaired neural tuning, which was not reinstated after sight restoration (*Rączy et al., 2022*). Other studies measured resting-state electroencephalogram (EEG) activity and analyzed periodic (alpha oscillations) (*Bottari et al., 2016*; *Ossandón et al., 2023*; *Pant et al., 2023*) as well as aperiodic activity (*Ossandón et al., 2023*). Both measures pointed towards an higher E/I ratio of visual cortex in congenital cataract-reversal individuals (*Ossandón et al., 2023*). In recent research, authors

have interpreted the slope of the aperiodic component of the EEG power spectral density function as an indirect indication of the relative level of excitation; the flatter the slope, the higher the assumed E/I ratio (*Gao et al., 2017*; *Lombardi et al., 2017*; *McSweeney et al., 2023*; *Medel et al., 2020*; *Molina et al., 2020*; *Muthukumaraswamy and Liley, 2018*; *Nanda et al., 2023*; *Schaworonkow and Voytek, 2021*). In fact, prospective studies in children have recently reported a flattening of this slope with age, which was interpreted as increasing levels of excitation with age (*Favaro et al., 2023*; *Hill et al., 2022*). *Ossandón et al., 2023* observed a flatter slope of the aperiodic power spectrum in the high-frequency range (20–40 Hz) but a steeper slope of the low-frequency range (1–19 Hz). This pattern was found in both congenital cataract-reversal individuals, as well as in permanently congenitally blind humans. The low-frequency range has often been associated with inhibition (*Jensen and Mazaheri, 2010*; *Lozano-Soldevilla, 2018*; *Lozano-Soldevilla et al., 2014*). However, it has remained unclear how to reconcile EEG resting-state findings for lower and higher frequency ranges.

Two studies with permanently congenitally blind humans employed Magnetic Resonance Spectroscopy (MRS) to investigate the concentration of both, the inhibitory neurotransmitter Gamma-Aminobutyric Acid (GABA) and the excitatory neurotransmitters Glutamate/Glutamine (Glx) as proxy measures of visual cortex inhibition and excitation, respectively (*Coullon et al., 2015*; *Weaver et al., 2013*). Glutamate/Glutamine concentration was significantly increased in the 'visual' cortex of anophthalmic (n=5) compared to normally sighted individuals, suggesting increased excitability (*Coullon et al., 2015*). Preliminary evidence in congenitally permanently blind individuals (n=9) suggested a decreased GABA concentration in the visual cortex compared to normally sighted individuals (*Weaver et al., 2013*). Thus, these MRS studies corroborated the hypothesis that a lack of visual input at birth enhances relative excitation in visual cortex compared to typical brain development. However, the degree to which neurotransmitter levels recover following sight restoration after a phase of congenital blindness, and how they related to electrophysiological activity, remained unclear.

Here, we filled this gap: we assessed Glutamate/Glutamine (Glx) and GABA+ concentrations using the MEGA-PRESS sequence (*Mescher et al., 1998*) in individuals whose sight had been restored, on average, after 12 years of congenital blindness. The ratio of Glx/GABA+ concentration was used as a proxy for the ratio of excitatory to inhibitory neurotransmission (*Gao et al., 2024*; *Grent-'t-Jong et al., 2022*; *Liu et al., 2015*; *Narayan et al., 2022*; *Steel et al., 2020*; *Takei et al., 2016*; *Zhang et al., 2020*). Ten congenital cataract-reversal individuals were compared to age-matched, normally sighted controls at rest. In addition to MRS, EEG was recorded to assess and compare aperiodic activity in the same participants. Participants were tested with their eyes open, eyes closed (MRS and EEG), and while viewing visual stimuli (EEG) which changed in luminance (*Pant et al., 2023*), since both neurotransmitter levels (*Kurcyus et al., 2018*) and EEG aperiodic activity (*Ossandón et al., 2023*) systematically varies between these conditions. We predicted an altered visual cortex Glx/GABA+ concentration ratio in the edited MRS signal in congenital cataract-reversal individuals. Since the aperiodic intercept has been linked to broad band neuronal firing (*Manning et al., 2009*; *Musall et al., 2014*; *Winawer et al., 2013*) and based on prior findings suggesting higher excitation in congenital cataract-reversal individuals (*Ossandón et al., 2023*; *Rączy et al., 2022*), we predicted a higher intercept as well as an altered slope of the EEG aperiodic component in this group. We further hypothesized that neurotransmitter changes would be concurrent with changes in the slope and intercept of the EEG aperiodic activity in congenital cataract-reversal individuals (*Ossandón et al., 2023*). Finally, we exploratorily assessed the relationship between the MRS and EEG parameters, as well as their possible link to visual deprivation history and visual acuity in congenital cataract-reversal individuals.

## Methods

### Participants

We tested two groups of participants. The first group consisted of 10 individuals with a history of dense bilateral congenital cataracts (CC group, 1 female, Mean Age = 25.8 years, Range = 11–43.5). Participants in this group were all recruited at the LV Prasad Eye Institute (Hyderabad, India) and the presence of dense bilateral cataracts at birth was confirmed by ophthalmologists and optometrists based on a combination of the following criteria: clinical diagnosis of bilateral congenital cataract, drawing of the pre-surgery cataract, occlusion of the fundus, nystagmus (a typical consequence of

congenital visual deprivation), a family history of bilateral congenital cataracts and a visual acuity of fixating and following light (FFL+) or less prior to surgery, barring cases of absorbed lenses. Absorbed lenses occur specifically in individuals with dense congenital cataracts (*Ehrlich, 1948*) and were diagnosed based on the morphology of the lens, anterior capsule wrinkling, and plaque or thickness of stroma. Prior to cataract surgery, the intactness of the retina is typically checked. Thus, we can exclude major retinal damage as source of group differences.

Duration of deprivation was calculated as the age of the participant when cataract removal surgery was performed on the first eye. Two participants were operated within the first year of life (at 3 months and 9 months of age), all other participants underwent cataract removal surgery after the age of 6 years (Mean Age at Surgery = 11.8 years, SD = 9.7, Range = 0.2–31.4). All participants were tested at least 1 year after surgery (Mean Time since Surgery = 14 years, SD = 9.1, Range = 1.8–30.9; *Table 1*). Visual acuity was significantly below typical vision in this group (*Table 1*, Appendix 1.1).

The second group comprised of 10 normally sighted individuals (SC group, 8 males, Mean Age = 26.3 years, Range = 12–41.8). Participants across the two groups were age matched ($t_{(9)}$ = –0.12, p=0.91). Congenital cataract-reversal individuals were clinically screened at the LV Prasad Eye Institute. Both groups did not self-report any neurological or psychiatric conditions, nor any medications. Additionally, all participants were screened for MRI exclusion criteria using a standard questionnaire from the radiology department. One additional individual was tested in each group; they were excluded from data analysis as their data files were corrupted due to inappropriate file transfer from the scanner. All participants (as well as legal guardians for minors) gave written and informed consent. This study was conducted after approval from the Local Ethical Commission of the LV Prasad Eye Institute (Hyderabad Eye Research Foundation LEC-11–086 and LEC-12-15-124, India) as well as of the Faculty of Psychology and Human Movement, University of Hamburg (EK-Röder-102015, Germany).

## Data collection and analysis

The present study consisted of three data acquisition parts on the same day: (1) MRS (45–60 min); (2) EEG (20 min plus time for capping); (3) visual acuity assessment (3–5 min.).

## Magnetic resonance spectroscopy

Participants underwent MRI and MRS scanning at LUCID Diagnostics in Hyderabad (India) with a 3T GE SIGNA Pioneer MRI machine employing a 24-channel head coil. An attendant was present in the scanning room for the duration of the scan to ensure that participants were comfortable and followed the instructions.

A T1 weighted whole brain image was collected for each participant (Repetition Time (TR)=14.97ms, Echo Time (TE)=6.74ms, Matrix size = 512 × 512, In-plane resolution = 0.43 × 0.43 mm, Slice thickness = 1.6 mm, Axial slices = 188, Interslice interval = –0.8 mm, Inversion time = 500ms, Flip angle = 15°). This structural scan enabled registration of every MRS scan to the participants' anatomical landmarks (*Figure 1*). For this scan, participants were instructed to stay as still as possible.

The MRS scans consisted of single-voxel spectroscopy data that were collected using the MEGA-PRESS sequence, which allows for in-vivo quantification of the low-concentration metabolites GABA and glutamate/glutamine (Glu/Gln; *Mescher et al., 1998*; *Mullins et al., 2014*). Due to the spectral overlap of GABA (3.0 ppm) and Glu/Gln (3.75 ppm) with the higher concentration peaks of N-Acetyl Aspartate (NAA) and Creatine (Cr), accurate quantification of GABA and Glu/Gln is challenging. MEGA-PRESS uses spectral editing to obtain these measurements. Spectroscopy data consisted of an edit-ON and an edit-OFF spectrum for each voxel, wherein the 'ON' and 'OFF' refer to whether the frequency of the editing pulse applied is on- or off-resonance with the signal coupled to the GABA complex (applied at approximately 1.9 ppm). Therefore, subtracting repeated acquisitions of the edit-ON and edit-OFF spectra allows for measurement of the magnitude of signals differing in their response to the editing pulse (e.g. GABA), while cancelling out signals that do not (e.g. Cr; *Mescher et al., 1998*). Each MEGA-PRESS scan lasted for 8.5 min and was acquired with the following specifications: TR = 2000ms, TE = 68ms, Voxel size = 40 mm x 30 mm x 25 mm, 256 averages. Additionally, eight unsuppressed water averages were acquired, allowing all metabolites to be referenced to the tissue water concentration. Concentrations of GABA and Glu/Gln quantified from these acquisitions are respectively referred to as GABA+, due to the presence of macromolecular contaminants in the

signal (*Mullins et al., 2014*), and Glx, due to the combined quantification of the Glu, Gln, and Gluta-thione peaks.

Two MRS scans were collected from the visual cortex, centered on the calcarine sulcus of every participant (*Figure 1*). A prior study with normally sighted individuals suggested that visual cortex Glx and GABA+ concentrations depend on whether the participants were scanned with eyes open or eyes closed (*Kurcyus et al., 2018*). Therefore, to ensure any group differences were not potentially driven by differences in eye opening/closure, we tested all participants at rest in two conditions – with eyes open (EO) and eyes closed (EC). Both scans were conducted with regular room illumination, that is, without any explicit visual stimulation.

To ensure that we were identifying neurochemical changes specific to visual regions, we selected the frontal cortex as a control region (*Figure 1*) and collected two scans (EO and EC) from the frontal cortex. The order of the MRS scans was counterbalanced across individuals for both locations and conditions. Two SC subjects did not complete the frontal cortex scan for the EO condition and were excluded from the statistical comparisons of frontal cortex neurotransmitter concentrations. Voxel placement was optimized to avoid the inclusion of the meninges, ventricles, skull and subcortical structures. For each participant, a proper placement was ensured by examining the voxel region across the slices in the acquired T1 volume. Saturation bands to nullify the skull signal were placed at the posterior and anterior edge of the visual cortex and frontal cortex voxel, respectively. Due to limitations of the clinical scanner settings, rotated and skewed voxels were not possible, and therefore voxels were not always located precisely parallel to the calcarine. As documented in Appendix 1.2, the visual cortex voxel showed significant (>60%) overlap with the V1-V6 region in every individual participant.

## MRS data analysis

All data analyses were performed in MATLAB (R2018b, The MathWorks Inc). For MRS data analyses, we used Gannet 3.0, a MATLAB based toolbox specialized for the quantification of GABA+ and Glx from edited spectrum data (*Edden et al., 2014*). Following initial data analysis, all datasets were rean-alyzed for quantification of NAA, GABA+ and Glx using linear combination modelling with the Osprey toolbox (v. 2.5.0) (*Oeltzschner et al., 2020*) in MATLAB 2024a (Appendix 1.3). Osprey had not been released when the study was originally conceptualized. The results did not differ between analysis toolboxes. Here, we present the originally planned analyses with Gannet 3.0.

GABA+ and edited Glx concentration values were obtained and corrected using the *GannetFit*, *GannetCoRegister*, G*annetSegment,* and *GannetQuantify* functions (*Edden et al., 2014*). Briefly, the reported water-normalized, alpha-corrected concentration values, were corrected for the differ-ences in GABA concentration and relaxation times between different tissue types in the voxel (grey matter, white matter, and cerebrospinal fluid; *Harris et al., 2015*). Gannet uses SPM12 to determine the proportion of grey matter, white matter and cerebrospinal fluid in each individual participant's voxel (*Penny et al., 2007*). Note that the tissue fraction values did not differ between groups or conditions (all p's>0.19, see Appendix 1.4). GABA+, Glx and Glx/GABA+ values were compared across groups as proxy measures of inhibition, excitation and E/I ratio, respectively. The use of Glx/GABA+ as a proxy measure of E/I neurotransmission is supported by a study that observed a regional balance between Glx and GABA+ at 3T (*Steel et al., 2020*). Further, the Glx/GABA+ ratio has been employed in prior studies of visual (*Takei et al., 2016*; *Zhang et al., 2020*), cingulate (*Bezalel et al., 2019*), frontal (*Gao et al., 2024*; *Liu et al., 2015*; *Narayan et al., 2022*), and auditory cortex (*Grent-'t-Jong et al., 2022*).

To control for potential unspecified visual cortex changes due to eye pathology, as opposed to genuine changes in neurotransmitter ratio, we compared NAA concentrations in the visual cortex of CC vs SC individuals. NAA forms one of the most prominent peaks in the MR spectrum (2.0 ppm chemical shift). NAA has been quantified with high reproducibility in the visual cortex (*Brooks et al., 1999*) and medial-temporal cortex (*Träber et al., 2006*) of neuro-typical individuals as well as in various pathologies across visual, frontal and temporal cortex (*Paslakis et al., 2014*), for example, schizophrenia (*Mullins et al., 2003*). We did not expect to find differences in NAA concentration between CC and SC individuals as it has not been demonstrated to vary in anophthalmia (*Coullon et al., 2015*) or permanent early blindness (*Weaver et al., 2013*) in humans. TARQUIN 4.3.11 was employed to analyze the OFF-spectrum data (*Wilson et al., 2011*) to assess NAA concentration.

FID-A toolbox was used to correct the data for phase errors across acquisitions arising from temporal changes in the magnetic field strength or participant motion (*Simpson et al., 2017*).

The reported values in the results are water-normalized. All data analyses were repeated with Cr-normalized values from Gannet 3.0, and significant results were replicated (Appendix 1.5).

## MRS data quality

The MRS minimum reporting standards form is found in *Supplementary file 1*. Mean signal-to-noise ratio values for GABA+ and Glx in all groups and conditions were above 19 in the visual cortex and above 8 in the frontal cortex (*Table 2*). A recent study has suggested that an SNR value above 3.8 allows for reliable quantification of GABA+ (*Zöllner et al., 2021*), in conjunction with considering a given study's sample size (*Mikkelsen et al., 2018*). Cramer-Rao lower bound (CRLB) values, that is, the theoretical lower limit of estimated error, were 30% or lower for NAA quantification in both groups and conditions (*Cavassila et al., 2001*). Note that CRLB values above 50% are considered unreliable (*Wilson et al., 2019*). In all quality metrics for Glx, GABA+ and NAA our dataset showed higher quality for the visual cortex voxel than for the frontal cortex voxel, irrespective of group (Main effect of region: all p's<0.004, Appendix 1.6). Such region effects have repeatedly been reported in the MRS literature. They were attributed to magnetic field distortions (*Juchem and de Graaf, 2017*) resulting from the proximity of the frontal cortex voxel to the sinuses. We chose a frontal control voxel rather than a parietal/sensorimotor control voxel (*Coullon et al., 2015*; *Weaver et al., 2013*) due to well-documented changes in multisensory cortical regions as a consequence of congenital blindness (*Harrar et al., 2018*; *Henschke et al., 2018*; *Jiang et al., 2016*; *Röder et al., 1999*; *Sabourin et al., 2022*; *Zatorre et al., 2012*). The fit error for the frontal cortex voxel was below 8.31% for GABA+ and Glx in both groups (*Table 2*). No absolute cutoffs exist for fit errors. However, Mikkelsen et al. reported a mean GABA+ fit error of 6.24+/-1.95% from a posterior cingulate cortex voxel across 8 GE scanners using the Gannet pipeline (*Mikkelsen et al., 2017*). Previous studies in special populations have used frontal cortex data with a fit error of <10% to identify differences between cohorts (*Gao et al., 2024*; *Maier et al., 2022*; *Pitchaimuthu et al., 2017*). Importantly, in the present study, data quality did not significantly differ between groups for GABA+, Glx, or NAA (Appendix 1.6, *Appendix 1—table 1*), making it highly unlikely that data quality differences contributed to group differences.

Prior to in vivo scanning, we confirmed the GABA+ and GABA+/Glx quantification quality with phantom testing (*Henry et al., 2011*; *Jenkins et al., 2019*). Imaging sequences were robust in identifying differences of 0.02 mM in GABA concentration; the known vs. measured concentrations of both GABA (*r*=0.81, p=0.004) and GABA/Glx (*r*=0.71, p=0.019) showed significant agreement. This 0.02 mM difference was documented by *Weaver et al., 2013* between the occipital cortices of early blind and sighted individuals. The detailed procedure and results are described in Appendix 1.7. The spectra from all individual subjects are shown in Appendix 1.8.

## MRS statistical analysis

All statistical analyses were performed using MATLAB R2018b and R v3.6.3.

We compared the visual cortex concentrations of three neurochemicals (GABA+ and Glx from the DIFF spectrum, NAA from the edit-OFF spectrum) between the two groups. For each metabolite, we submitted the concentration values from the visual cortices of CC and SC individuals to a group (2 Levels: CC, SC)-by-condition (2 Levels: EO, EC) ANOVA model. To compare the Glx/GABA+ ratio between groups, we additionally submitted this ratio value to a group-by-condition ANOVA. Identical analyses were performed for the corresponding frontal cortex neurotransmitter values. Wherever necessary, post-hoc comparisons were performed using t-tests. The data were tested for normality (Shapiro-Wilk) and homogeneity of variance (Levene's Test) in R v3.6.3 (Appendix 1.9, *Appendix 1—table 2*). In all ANOVA models, the residuals did not significantly differ from normality.

## Electrophysiological recordings

EEG data analyzed in the present study are a subset of datasets that were included in previous reports (*Ossandón et al., 2023*; *Pant et al., 2023*). The EEG datasets were re-analyzed to investigate aperiodic activity in the same participants who took part in the MRS study. MRS and EEG data were acquired on the same day. The EEG was recorded in three conditions: (1) at rest with eyes open (EO)

(3 min), (2) at rest with eyes closed (EC) (3 min), and (3) during visual stimulation with stimuli that changed in luminance (LU) with equal power at all frequencies (0–30 Hz; *Pant et al., 2023*). We used the slope of the aperiodic (1 /f) component of the EEG spectrum as an estimate of E/I ratio (*Gao et al., 2017*; *Medel et al., 2020*; *Muthukumaraswamy and Liley, 2018*) and the intercept as an estimate of broadband neuronal firing activity (*Haller et al., 2018*; *Manning et al., 2009*; *Miller, 2010*).

The EEG was recorded using Ag/AgCl electrodes attached according the 10/20 system (*Homan et al., 1987*) to an elastic cap (EASYCAP GmbH, Herrsching, Germany; *Figure 1*). We acquired 32 channel EEG using the BrainAmp amplifier, with a bandwidth of 0.01–200 Hz, sampling rate of 5 kHz and a time constant of 0.016 Hz /(10 s; https://www.brainproducts.com/). All scalp recordings were performed against a left ear lobe reference. Electrode impedance was kept below 10 kOhm in all participants.

Participants were asked to sit as still as possible while the EEG was recorded. First, resting-state EEG data were collected. During the EO condition, participants were asked to look towards a blank screen and to avoid eye movements. During the EC condition, participants were instructed to keep their eyes closed. The order of conditions was randomized across participants.

Subsequently, EEG data were recorded during 100 trials of a target detection task with stimuli that changed in luminance (LU). Stimuli were presented with a Dell laptop, on a Dell 22-inch LCD monitor with a refresh rate of 60 Hz. They were created with MATLAB r2018b (The MathWorks, Inc, Natick, MA) and the Psychtoolbox 3 (*Brainard, 1997*; *Kleiner et al., 2007*). On each trial, participants observed a circle at the center of a black screen, subtending a visual angle of 17 degrees. The circle appeared for 6.25 s and changed in luminance with equal power at all frequencies (0–30 Hz). At the end of every trial, participants had to indicate whether a target square, subtending a visual angle of 6 degrees, appeared on that trial. The experiment was performed in a darkened room (for further details, see *Pant et al., 2023*).

## EEG data analysis

Data analysis was performed using the EEGLab toolbox on MATLAB 2018b (*Delorme and Makeig, 2004*). All EEG datasets were filtered using a Hamming windowed sinc FIR filter, with a high-pass cutoff at 1 Hz and a low-pass cutoff at 45 Hz. A prior version of the analysis was conducted with line noise removal via spectrum interpolation (*Ossandón et al., 2023*). However, the analyses reported here did not include this step, since we implemented a low-pass cutoff (20 Hz) which falls far below the typical line noise frequency (50 Hz). Eye movement artifacts were detected in the EEG datasets via independent component analysis using the *runica.m* function's *Infomax* algorithm in EEGLab. Components corresponding to horizontal or vertical eye movements were identified via visual inspection based on criteria discussed in *Plöchl et al., 2012* and removed.

The two 3 min long resting-state recordings (EC, EO) were divided into epochs of 1 s. Epochs with signals exceeding ±120 μV were rejected for all electrodes (see Appendix 1.10, *Appendix 1—table 3* for percentages by group and condition). We then calculated the power spectral density of the EO and EC resting-state data using the *pwelch* function (sampling rate = 1000 Hz, window length = 1000 samples, overlap = 0).

Datasets collected while participants viewed visual stimuli that changed in luminance (LU) were downsampled to 60 Hz (antialiasing filtering performed by EEGLab's *pop_resample* function) to match the stimulation rate. The datasets were divided into 6.25 s long epochs corresponding to the duration of visual stimulation per trial. Subsequently, baseline removal was conducted by subtracting the mean activity across the length of an epoch (1 s for the EO and EC conditions, 6.25 s for the LU condition) from every data point. After baseline removal, epochs with signals exceeding a threshold of ±120 μV were rejected in order to exclude potential artifacts. Finally, we calculated the power spectral density of the LU data using the *pwelch* function (sampling rate = 60 Hz, window length = 60 samples, overlap = 0).

We derived the aperiodic (1 /f) component of the power spectrum for the EO, EC, and LU conditions (*Donoghue et al., 2020b*; *Schaworonkow and Voytek, 2021*). First, we fit the 1 /f distribution function to the frequency spectrum of each participant, separately for each electrode. The 1 /f distribution was fit to the normalized spectrum converted to log-log scale (range = 1–20 Hz; *Donoghue et al., 2020a*; *Gyurkovics et al., 2021*; *Schaworonkow and Voytek, 2021*). We excluded the alpha range (8–14 Hz) for this fit to avoid biasing the results due to documented differences in alpha activity

**Table 1.** Clinical and demographic information of the participants with a history of dense bilateral congenital cataracts (CC) as well as demographic information and visual acuity of age-matched normally sighted control participants (SC).

NA indicates that patient's data for the field were not available. FFL: Fixating and Following Light; CF: Counting Fingers; PL: Perceiving Light. Duration of visual deprivation was calculated by subtracting the date of birth from the date of surgery on the first eye (and thus corresponds to the age at surgery). Time since surgery was calculated by subtracting the date of surgery on the first eye from the date of testing. Visual acuity on the date of testing was measured binocularly with the Freiburg Vision Test (FrACT).

| | Gender | Age | Visual acuity on date tested (logMAR) | Comorbidities | | | Visual acuity pre surgery | | Duration of visual deprivation (Years) | Time since surgery (Years) | Family history |
| | | | | Absorbed lenses | Strabismus | Nystagmus | OD | OS | | | |
|---|---|---|---|---|---|---|---|---|---|---|---|
| CC1 | Male | 17.0 | 0.17 | No | Yes | Yes | FFL - | FFL + | 0.2 | 16.8 | No |
| CC2 | Male | 43.5 | 0.9 | Yes | Yes | Yes | 1.18 | 1 | 20.8 | 22.7 | Yes |
| CC3 | Male | 18.7 | 0.9 | Yes | Yes | Yes | 1.48 | 1.77 | 15.6 | 3.1 | No |
| CC4 | Male | 15.2 | 0.62 | Yes | NA | Yes | CF at 1.5 m | CF at 3 m | 7.0 | 8.2 | No |
| CC5 | Male | 32.4 | 0.88 | No | Yes | Yes | NA | NA | 14.0 | 18.4 | Yes |
| CC6 | Male | 23.9 | 0.78 | No | Yes | Yes | NA | NA | 6.0 | 17.9 | Yes |
| CC7 | Male | 13.1 | 0.54 | No | Yes | Yes | PL+ | PL+ | 0.8 | 12.4 | No |
| CC8 | Male | 18.3 | 0.66 | Yes | No | Yes | 1.2 | 1.3 | 16.4 | 1.9 | Yes |
| CC9 | Male | 36.9 | 1.34 | No | NA | Yes | NA | NA | 6.0 | 30.9 | Yes |
| CC10 | Female | 38.8 | 1.04 | Yes | Yes | Yes | 1.48 | 1.48 | 31.4 | 7.4 | Yes |
| SC1 | Male | 17.7 | 0.2 | | | | | | | | |
| SC2 | Male | 41.9 | −0.27 | | | | | | | | |
| SC3 | Male | 19.5 | −0.25 | | | | | | | | |
| SC4 | Male | 16.0 | −0.11 | | | | | | | | |
| SC5 | Male | 33.3 | −0.12 | | | | | | | | |
| SC6 | Male | 24.0 | −0.16 | | | | | | | | |
| SC7 | Male | 12.2 | −0.25 | | | | | | | | |
| SC8 | Female | 25.1 | −0.28 | | | | | | | | |
| SC9 | Female | 36.0 | −0.21 | | | | | | | | |
| SC10 | Male | 37.3 | −0.22 | | | | | | | | |

between CC and SC individuals (*Bottari et al., 2016*; *Ossandón et al., 2023*; *Pant et al., 2023*). This 1 /f fit resulted in a value of the aperiodic slope, an aperiodic intercept value corresponding to the broadband power of 1–20 Hz, and a fit error value for the spectrum of every participant, individually for each electrode. Spectra from individual subjects are displayed in Appendix 1.11. The visual cortex aperiodic slope and intercept values were obtained by averaging across the pre-selected occipital electrodes O1 and O2, resulting in one value of broadband slope and one value of intercept per participant and condition (*Figure 1*). This procedure yielded average $R^2$ values >0.91 for the aperiodic fit in each group and condition (Appendix 1.11, *Appendix 1—table 4*).

## EEG statistical analysis

We compared the average visual cortex aperiodic slope and intercept in separate group (two Levels: CC, SC) by condition (three levels: EC, EO, LU) ANOVA models. The data were tested for normality (Shapiro-Wilk) and homogeneity of variance (Levene's Test) in R v3.6.3 (see Appendix 1.9, *Appendix 1—table 2*); in all ANOVA models, the residuals did not significantly differ from normality.

## Visual acuity

Visual acuity was measured binocularly for every participant on the date of testing, using the Freiburg Visual Acuity Test (FrACT; *Bach, 1996*, *Bach, 2007*, https://michaelbach.de/fract/). Visual acuity is reported as the logarithm of the minimum angle of resolution (logMAR, *Table 1*), wherein higher values indicate worse vision (*Elliott, 2016*). Analogous to previous studies, we ran a number of exploratory

correlation analyses between GABA+, Glx and Glx/GABA+ concentrations and visual acuity at the date of testing, duration of visual deprivation, and time since surgery, respectively, in the CC group (*Birch et al., 2009*; *Guerreiro et al., 2015*; *Kalia et al., 2014*; *Rajendran et al., 2020*). As expected from normal vision in the SC group, they did not show considerable variance in visual acuity (*Table 1*); thus, we refrained from calculation correlations between visual acuity and MRS/EEG parameters in the SC group. Based on the literature, we additionally tested the correlation between the neurotransmitter levels and chronological age across the CC and SC groups. All reported correlation coefficients are Pearson correlations, and 95% confidence intervals were calculated for all correlation coefficients.

### Exploratory correlation analyses between MRS and EEG measures

Exploratory correlation analyses between EEG and MRS measures were run separately for CC and SC individuals. We calculated Pearson correlations between the aperiodic intercept and GABA+, Glx, and Glx/GABA+ concentrations. Further, Pearson correlations between the aperiodic slope, and the concentrations of GABA+, Glx, and Glx/GABA+ were assessed. MRS measures collected at rest with EO and EC were correlated with the corresponding resting-state EEG conditions (EO, EC). EEG metrics for the visual stimulation (LU) condition with flickering stimuli were tested for correlation with GABA+, Glx, and Glx/GABA+ concentration measured while participants' eyes were open at rest. We did not have prior hypotheses as to the best of our knowledge no extant literature has tested the correlation between aperiodic EEG activity and MRS measures of GABA+, Glx, and Glx/GABA+. Therefore, we corrected for multiple comparisons using the Bonferroni correction (six comparisons).

## Results
### Transient congenital visual deprivation lowered the Glx/GABA+ concentration in the visual cortex

The Glx/GABA+ concentration ratio was significantly lower in the visual cortex of congenital cataract-reversal (CC) than age-matched, normally sighted control (SC) individuals (main effect of group: $F_{(1,39)}$ = 5.80, p0.021, $\eta_p^2$=0.14) (*Figure 2*). This effect did not vary with eye closure (main effect of condition: $F_{(1,39)}$ = 2.29, p=0.139, $\eta_p^2$=0.06, group-by-condition interaction: $F_{(1,39)}$ = 1.15, p=0.290, $\eta_p^2$=0.03). As a control for unspecific effects of surgery unrelated to visual deprivation on neurochemistry, the frontal cortex Glx/GABA+ concentration was compared between groups. There was no difference between CC and SC individuals in their frontal cortex Glx/GABA+ concentration (main

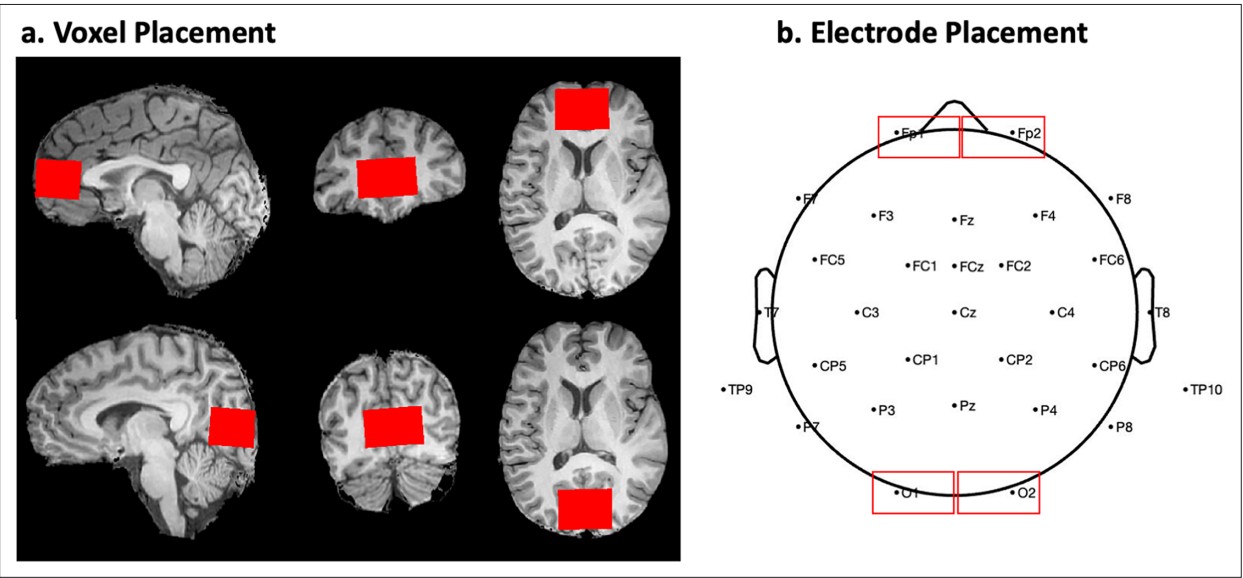

**Figure 1.** Voxel placement for Magnetic Resonance Spectroscopy and electrode placement for Electroencephalography. (**a**) Position of the frontal cortex (top) and visual cortex (bottom) voxels in a single subject. Skull-stripped figures output from SPM12. (**b**) Electrode montage according to the 10/20 electrode system with marked occipital electrodes preselected for analyses, and frontal electrodes used for control analyses.

effect of group: F(1,37) = 0.05, p=0.82, $\eta_p^2$<0.01, main effect of condition: F(1,37) = 2.98, p=0.093, $\eta_p^2$=0.07, group-by-condition interaction: F(1,37) = 0.09, p=0.76, $\eta_p^2$<0.01; *Figure 2*).

When separately comparing CC and SC individuals' GABA+ and Glx concentrations in the visual cortex, we did not find any significant group difference (GABA+ main effect of group: F(1,39) = 2.5, p=0.12, $\eta_p^2$=0.06, main effect of condition: F(1,39) = 0.6, p=0.43, $\eta_p^2$=0.02, group-by-condition interaction: F(1,39) = 0.03, p=0.86, $\eta_p^2$<0.001; Glx main effect of group: F(1,39) = 2.8, p=0.103, $\eta_p^2$=0.07, main effect of condition: F(1,39) = 1.8, p=0.19, $\eta_p^2$=0.05, group-by-condition interaction: F(1,39) = 1.27, p=0.27, $\eta_p^2$ –0.03; *Figure 2*). In the frontal cortex, GABA+ and Glx concentrations did not vary either with group or condition (all p-values >0.19, all $\eta_p^2$<0.05; *Figure 2*). Note that these findings were replicated when Osprey's quantification method was used: Glx/GABA+ concentration was lower in the visual cortex of CC than SC individuals, while GABA+ and Glx concentration did not significantly differ (Appendix 1.3). When analyses were repeated with Cr-normalized Glx/

**Table 2.** Quality metrics for Magnetic Resonance Spectroscopy data.

Mean quality metrics in each group are reported with the standard deviation in parentheses. The displayed quality metrics for signal-to-noise ratio, full-width half maxima and fit error are those output by Gannet 3.0: signal-to-noise-ratio (SNR), was calculated in GannetFit.m by estimating the noise in the GABA+/Glx/NAA signal across acquisitions and by dividing the absolute peak height of the GABA+/Glx/NAA signal by the estimated noise; full-width-half-maxima (FWHM), is defined as the width of the peak in Hertz (Hz); and fit error, is defined as the standard deviation of the residual of the GABA+/Glx/NAA peak fit. The fit error is expressed as a percentage of the GABA+/Glx peak height. The Cramer-Rao lower bound is reported as output by TARQUIN 4.3.11 for the NAA signal (not calculated for GABA+ or Glx as these metabolites were quantified using Gannet 3.0).

| Signal-to-noise ratio | | CC | SC |
|---|---|---|---|
| | NAA | 293.16 (47.50) | 289.01 (50.91) |
| | GABA+ | 21.53 (3.66) | 19.08 (3.99) |
| Visual cortex | Glx | 23.75 (3.75) | 22.18 (5.26) |
| | NAA | 108.37 (21.84) | 97.20 (28.08) |
| | GABA+ | 10.311 (2.20) | 8.30 (1.93) |
| Frontal cortex | Glx | 15.82 (4.85) | 13.58 (3.86) |
| **Full-width-half maxima** | | **CC** | **SC** |
| | NAA | 9.04 (0.94) | 8.69 (0.75) |
| | GABA+ | 19.84 (1.13) | 19.10 (0.71) |
| Visual cortex | Glx | 16.62 (1.63) | 16.46 (1.63) |
| | NAA | 19.26 (2.33) | 21.42 (3.79) |
| | GABA+ | 21.69 (3.15) | 23.23 (3.41) |
| Frontal cortex | Glx | 27.54 (8.70) | 30.63 (12.64) |
| **Fit error** | | **CC** | **SC** |
| | NAA | 0.81 (0.20) | 0.77 (0.15) |
| | GABA+ | 3.42 (0.63) | 3.68 (0.63) |
| Visual cortex | Glx | 3.10 (0.58) | 3.18 (0.47) |
| | NAA | 1.33 (0.41) | 1.70 (0.57) |
| | GABA+ | 6.57 (2.20) | 8.31 (3.65) |
| Frontal cortex | Glx | 4.44 (1.54) | 5.15 (1.90) |
| **Cramer-Rao lower bound** | | **CC** | **SC** |
| Visual cortex | NAA | 0.13 (0.02) | 0.14 (0.03) |
| Frontal cortex | NAA | 0.33 (0.22) | 0.26 (0.24) |

GABA+ concentrations, the Glx concentration was found to be significantly lower in CC vs SC individuals' visual cortices, in addition to the lower Glx/GABA+ concentration ratio (Appendix 1.5). Since this finding was not replicated with water normalized Glx concentration in Gannet or Osprey, we refrain from interpreting this additional group effect for Glx.

The Glx/GABA+ concentration measured when CC individuals' eyes were closed correlated positively with visual acuity on the logMAR scale ($r$=0.65, $p$=0.044), indicating that CC individuals with higher Glx/GABA+ values had worse visual acuity (*Figure 2C*, Appendix 1.12). The same correlation was not significant for the eyes opened condition ($r$=–0.042, $p$=0.908; *Figure 2C*). Duration of deprivation and time since surgery did not significantly predict Glx/GABA+, GABA+ or Glx concentrations in the CC group (all p-values >0.088, Appendix 1.12).

## No difference in NAA concentration between CC and SC individuals' visual cortices

As a control measure to ensure that between-group differences were specific to hypothesized changes in Glx and GABA+ concentrations, we compared the NAA concentration between CC and SC individuals. The NAA concentration did not significantly differ between groups, neither in visual (main effect of group: $F_{(1,39)}$ = 0.03, $p$=0.87, $\eta_p^2$<0.001, main effect of condition: $F_{(1,39)}$ = 0.31, $p$=0.58, $\eta_p^2$<0.01, group-by-condition interaction: $F_{(1,39)}$ = 0.09, $p$=0.76, $\eta_p^2$<0.01) nor frontal cortex (main effect of group: $F_{(1,37)}$ = 1.1, $p$=0.297, $\eta_p^2$=0.02, main effect of condition: $F_{(1,37)}$ = 0.14, $p$=0.71, $\eta_p^2$=0.01, group-by-condition interaction: $F_{(1,37)}$ = 0.03, $p$=0.86, $\eta_p^2$<0.001) (Appendix 1.13, *Appendix 1—figure 14*).

## Transient congenital visual deprivation resulted in a steeper aperiodic slope and higher aperiodic intercept at occipital sites

The aperiodic slope (1–20 Hz), measured via EEG as an electrophysiological estimate of the E/I ratio (*Gao et al., 2017*; *Muthukumaraswamy and Liley, 2018*), was compared between CC and SC individuals. The aperiodic slope was significantly steeper, that is, more negative, at occipital electrodes in CC than in SC individuals ($F_{(1,59)}$ = 13.1, $p$<0.001, $\eta_p^2$=0.19; *Figure 3*). Eye closure and visual stimulation did not affect the steepness of the aperiodic slope ($F_{(2,59)}$ = 0.78, $p$=0.465, $\eta_p^2$=0.03, group-by-condition interaction: $F_{(2,59)}$ = 0.12, $p$=0.885, $\eta_p^2$<0.01).

The aperiodic intercept (1–20 Hz) was compared between CC and SC individual to estimate group differences in broadband neural activity (*Manning et al., 2009*; *Musall et al., 2014*; *Winawer et al., 2013*) and was found to be significantly larger at occipital electrodes in CC than SC individuals (main effect of group: $F_{(1,59)}$ = 5.2, $p$=0.026, $\eta_p^2$=0.09; (*Figure 3*)). Eye closure did not affect the magnitude of the aperiodic intercept in either group (main effect of condition: $F_{(2,59)}$ = 0.16, $p$=0.848, $\eta_p^2$<0.01, group-by-condition interaction: $F_{(2,59)}$ = 0.11, $p$=0.892, $\eta_p^2$<0.01). No significant group differences in slope and intercept were found for frontal electrodes (Appendix 1.14).

Within the CC group, visual acuity, time since surgery and duration of blindness did not significantly correlate with the aperiodic slope or the intercept (all p's>0.083, Appendix 1.15). Age negatively correlated with the aperiodic intercept across CC and SC individuals, that is, a reduction of the intercept was observed with age. Similar effects of chronological age have been previously observed (*Hill et al., 2022*; *Voytek et al., 2015*) (Appendix 1.15).

## Glx concentration predicted the aperiodic intercept in CC individuals' visual cortices during ambient and flickering visual stimulation

We exploratorily tested the relationship between Glx, GABA+ and Glx/GABA+ measured at rest and the EEG aperiodic intercept measured at rest and during flickering visual stimulation, separately for the CC and the SC group. Visual cortex Glx concentration in CC individuals was positively correlated with the aperiodic intercept either when participants had their eyes open during rest ($r$=0.91, $p$=0.001, Bonferroni corrected) or when they viewed flickering stimuli ($r$=0.90, $p$<0.001, Bonferroni corrected). Corresponding correlations were not significant for Glx concentrations in the eyes closed condition ($r$=0.341, $p$>0.99, Bonferroni corrected). Moreover, in SC individuals, no significant correlation was observed between visual cortex Glx concentration and aperiodic intercept in any condition (all p's>0.99, Bonferroni corrected) (*Figure 4*). Given the correlation between the aperiodic intercept and chronological age across groups (Appendix 1.15), we performed a post-hoc linear regression analysis

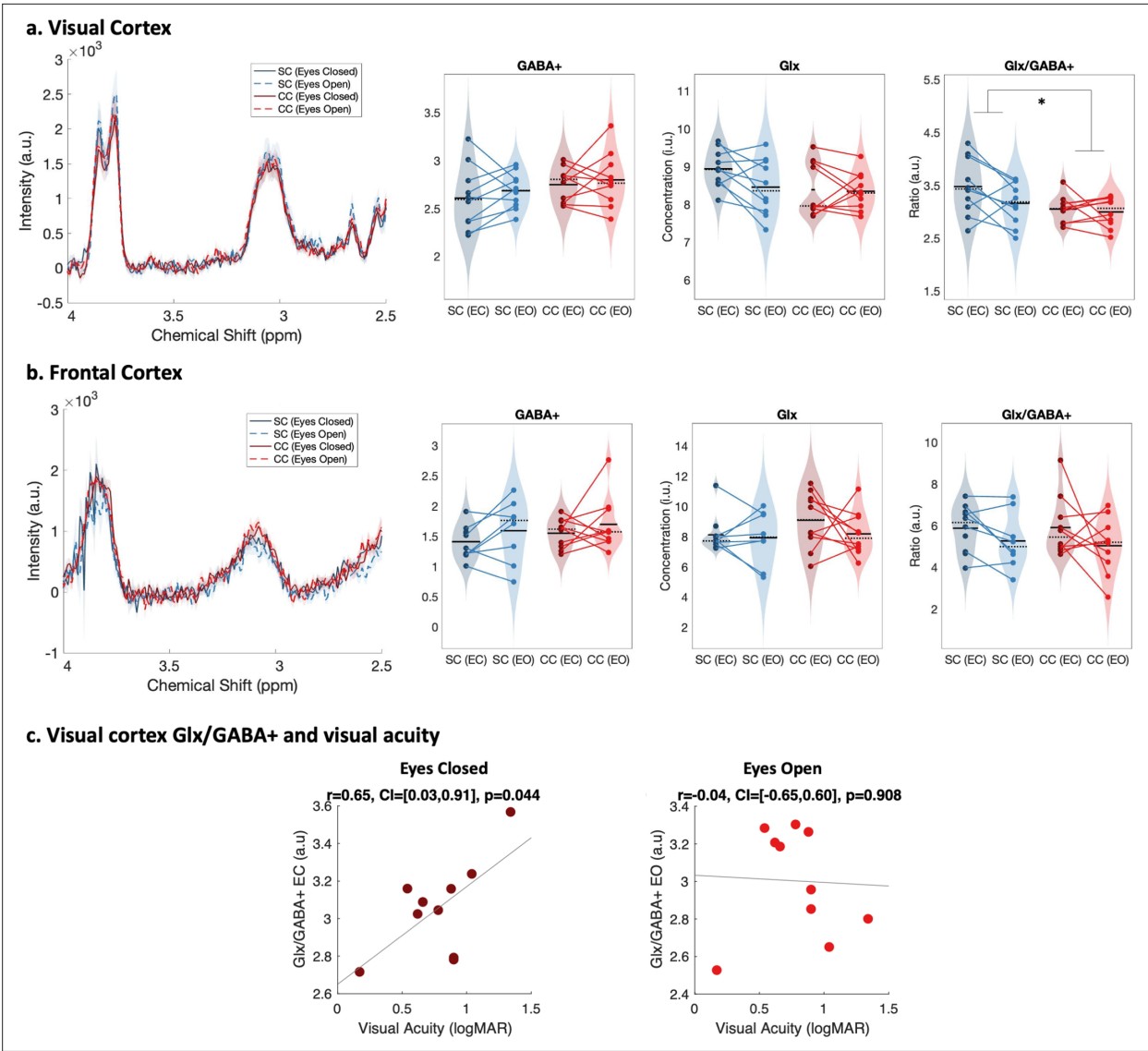

**Figure 2.** Edited spectra obtained from Magnetic Resonance Spectroscopy (MRS). (**a**) Average edited spectra showing GABA+ and edited Glx peaks in the visual cortices of normally sighted individuals (SC, green) and individuals with reversed congenital cataracts (CC, red) are shown. Edited MRS DIFF spectra are separately displayed for the eyes open (EO), and eyes closed (EC) conditions using dashed and solid lines respectively. The standard error of the mean is shaded. Water-normalized GABA+, water-normalized Glx, and Glx/GABA+ concentration distributions for each group and condition are depicted as violin plots on the right. The solid black lines indicate mean values, and dotted lines indicate median values. The colored lines connect values of individual participants across conditions. (**b**) Corresponding average edited MRS spectra and water-normalized GABA+, water-normalized Glx and Glx/GABA+ concentration distributions measured from the frontal cortex are displayed. (**c**) Correlations between visual cortex Glx/GABA+ concentrations in the visual cortex of CC individuals and visual acuity in logMAR units are depicted for the eyes closed (EC, left) and eyes open (EO, right) conditions. The 95% confidence intervals (CI) of the correlation coefficients (r) are reported.

to model the aperiodic intercept in the CC group with both age and Glx concentration as covariates. Glx concentration, but not age, significantly predicted the aperiodic intercept within the CC group during rest with eyes open, and during visual stimulation (Appendix 1.16).

A negative correlation between the aperiodic slope and Glx concentration in CC individuals (i.e. steeper slopes with increasing Glx concentration) was observed during visual stimulation, but did not survive correction for multiple comparisons (Appendix 1.17). No such correlation was observed between Glx concentration and aperiodic slope in the eyes open or closed conditions. Visual cortex GABA+ concentration and Glx/GABA+ concentration ratios did not significantly correlate with the aperiodic intercept or slope in either CC or SC individuals, during any experimental condition (Appendix 1.17).

## Discussion

Research in non-human animals has provided convincing evidence that the ratio of excitation to inhibition (E/I) in the visual cortex is reliant on early visual experience (*Froemke, 2015*; *Haider et al., 2006*; *Hensch et al., 1998*; *Takesian and Hensch, 2013*; *Wu et al., 2022*). Studies in humans who were born blind due to dense bilateral cataracts, and who had received sight restoration surgery in childhood or as adults, have found limited recovery of both basic visual and higher order visual functions (*Birch et al., 2009*; *Röder and Kekunnaya, 2021*). Here, we tested whether neurotransmitter concentrations and electrophysiological markers of cortical E/I ratio depend on early visual experience in humans, and how possible changes in visual cortex E/I ratio relate to sight recovery. First, we employed MRS and assessed Glutamate/Glutamine (Glx) and GABA+ concentrations, as well as their ratio, in the visual cortex (*Shibata et al., 2017*; *Steel et al., 2020*; *Takei et al., 2016*). Second, the slope and intercept of the aperiodic resting-state EEG activity with eyes open and closed (*Gao et al., 2017*; *Muthukumaraswamy and Liley, 2018*), as well as during flickering visual stimulation (*Pant et al., 2023*), were measured over the occipital cortex in the same individuals. The EEG measures allowed us to exploratorily relate neurotransmitter changes to neural activity changes in congenital cataract-reversal individuals.

We found a lower Glx/GABA+ concentration ratio in the visual cortex of congenital cataract-reversal (CC) individuals as compared to normally sighted controls (SC). Additionally, the slope of the aperiodic EEG power spectrum was steeper for the low-frequency range (1–20 Hz), and its intercept was higher in CC than SC individuals. In the CC group, Glx concentration correlated with the intercept of the aperiodic component during flickering visual stimulation. The Glx/GABA+ concentration ratio during the eyes closed condition predicted visual acuity of CC individuals. Together, the present results provide initial evidence for experience-dependent development of the E/I ratio in the human visual cortex, with consequences for behavior.

Previous MRS studies in the visual cortex of permanently congenitally blind humans reported higher Glx concentrations (*Coullon et al., 2015*) in five anophthalmic humans, and numerically lower GABA concentrations in congenitally blind humans (*Weaver et al., 2013*) (n=9), as compared to normally sighted individuals. These results were interpreted as suggesting a higher E/I ratio in the

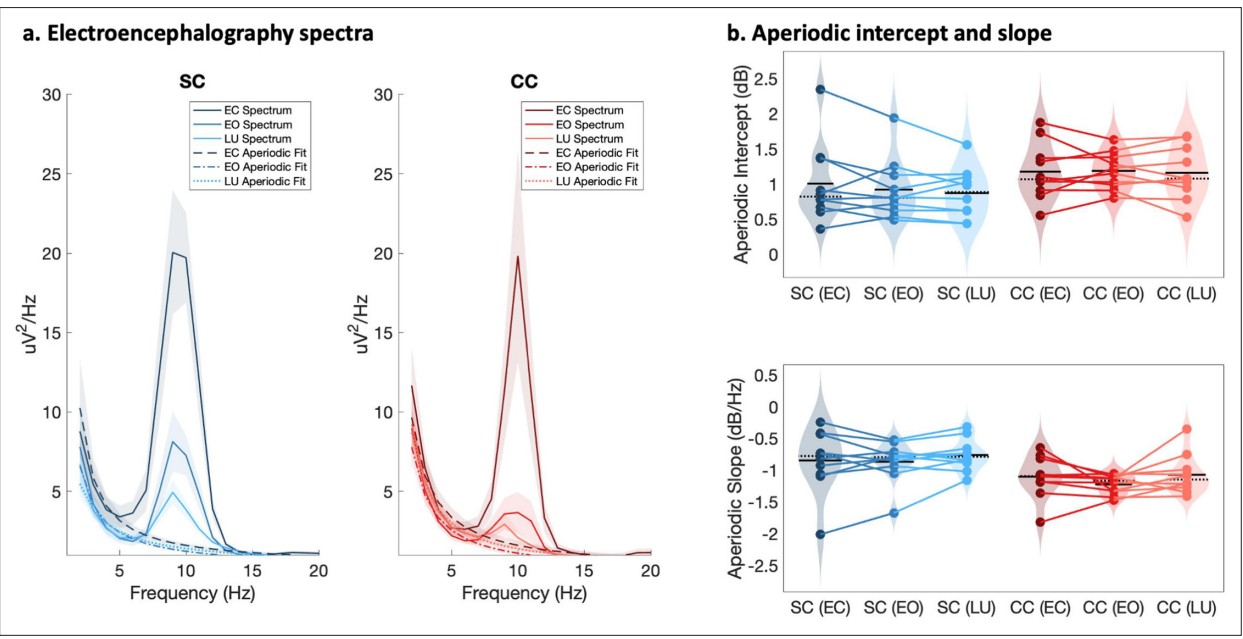

**Figure 3.** Full spectrum and aperiodic activity of the electroencephalogram (EEG). (**a**) EEG spectra across O1 and O2 with the corresponding aperiodic (1 /f) fits for normally sighted individuals (SC, blue, left) and individuals with reversed congenital cataracts (CC, red, right). Spectra of EEG recordings are displayed for the eyes closed (EC) and eyes opened (EO) conditions, as well as while viewing stimuli that changed in luminance (LU). Shaded regions represent the standard error of the mean. (**b**) Aperiodic intercept (top) and slope (bottom) value distributions for each group and condition are displayed as violin plots. Solid black lines indicate mean values, dotted black lines indicate median values. Colored lines connect values of individual participants across conditions.

visual cortex of permanently congenitally blind humans, which would be consistent with the extant literature on higher BOLD activity in the visual cortices of the same population (*Bedny, 2017*; *Rączy et al., 2022*; *Röder and Kekunnaya, 2022*). We observed a lower Glx/GABA+ ratio and a steeper slope of the aperiodic EEG activity (1–20 Hz) at occipital electrodes, both of which suggest a lower rather than higher E/I ratio in the visual cortex of CC individuals. Here, we speculate that our results imply a change in neurotransmitter concentrations as a consequence of *restoring* vision following congenital blindness. Further, we hypothesize that due to limited structural plasticity after a phase of congenital blindness, the neural circuits of CC individuals, which had adapted to blindness after birth, likely employ physiological plasticity mechanisms (*Knudsen, 1998*; *Mower et al., 1985*; *Röder et al., 2021*), in order to re-adapt to the newly available visual excitation following sight restoration later in life.

Structural remodeling (*Bourgeois and Rakic, 1996*) for typical E/I balance requires visual experience following birth (*Hensch and Fagiolini, 2005b*; *Takesian and Hensch, 2013*; *Zhang et al., 2018*) and is linked to a sensitive period (*Desai et al., 2002*; *Hensch and Fagiolini, 2005b*). A repeatedly

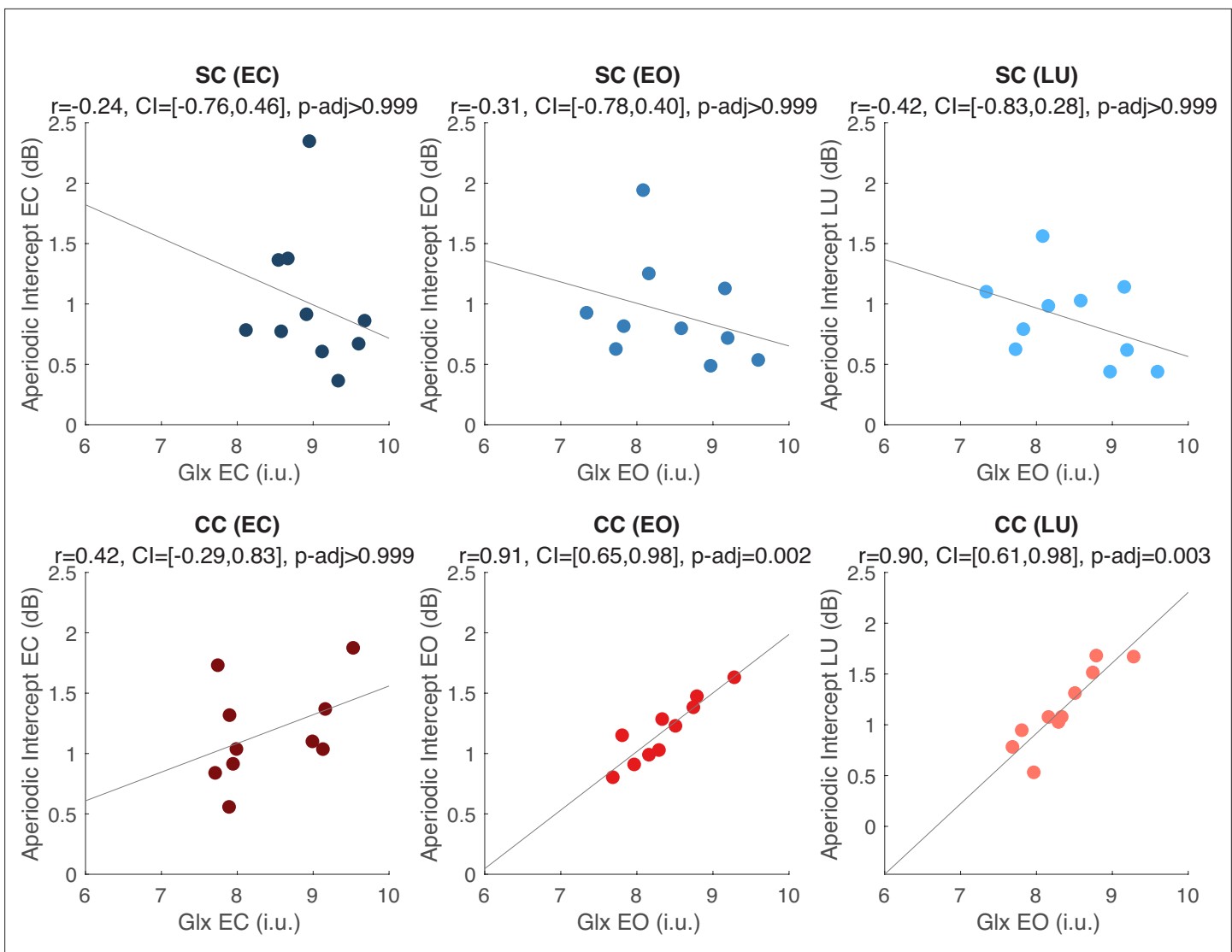

**Figure 4.** Exploratory correlation analyses between the aperiodic intercept (1–20 Hz) and glutamate/glutamine (Glx) concentration in the visual cortex. Correlations between water-normalized Glx concentration and aperiodic intercept are shown for the eyes closed (EC, left), eyes open (EO, middle) and visual stimulation (LU, right) conditions for sighted controls (SC, green, top) and individuals with reversed congenital cataracts (CC, red, bottom). The reported adjusted p values (p-adj) are Bonferroni corrected for multiple comparisons. The 95% confidence intervals (CI) of the correlation coefficients (**r**) are reported.

documented finding in permanently congenitally blind humans is the increased thickness of visual cortex (*Anurova et al., 2015*; *Hölig et al., 2023*; *Jiang et al., 2009*). These structural changes in permanently congenitally blind individuals were interpreted as a lack of experience-dependent pruning of exuberant synapses and/or reduced myelination, the latter typically leading to a shift of the grey-white matter boundary (*Natu et al., 2019*). In parallel, it was observed in non-human primates that the overproduction of synapses during the initial phase of brain development was independent of experience, but that synaptic pruning, predominantly of excitatory synapses, depended on visual experience (*Bourgeois and Rakic, 1996*; *Bourgeois et al., 1989*). The lack of excitatory synapse pruning was thought to underlie the observed higher excitability of visual cortex due to congenital visual deprivation (*Benevento et al., 1992*; *Huang et al., 2015*; *Morales et al., 2002*). Crucially, increased visual cortex thickness (*Feng et al., 2021*; *Guerreiro et al., 2015*; *Hölig et al., 2023*) and higher BOLD activity during rest with the eyes open *Rączy et al., 2022* have been observed for CC individuals following congenital blindness, suggesting incomplete recovery of cortical structure and function *after* sight restoration in humans. Thus, the restored feedforward drive to visual cortex after cataract removal surgery might reach a visual cortex with a lower threshold for excitation.

As to the best of our knowledge, the present study was the first assessing MRS markers of cortical excitation in humans following sight restoration after congenital blindness, we allude to non-human animal work for the interpretation of our findings. Such studies have often demonstrated that excitation and inhibition go hand-in-hand (*Froemke, 2015*; *Haider et al., 2006*; *Isaacson and Scanziani, 2011*; *Tao and Poo, 2005*). Analogously, we speculate that an overall reduction in Glx/GABA ratio might be effective in counteracting the aforementioned adaptations to congenital blindness, that is a lower threshold for excitation. Higher overall visual cortex excitation as a consequence of congenital blindness (*Benevento et al., 1992*; *Morales et al., 2002*) might come with the risk of runaway excitation in the presence of restored visually-elicited excitation. Phrased differently, we postulate that significantly lowering overall excitation of an originally hyper-excited visual cortex (during the phase of blindness) after sight restoration might be instrumental for maintaining neural circuit stability. Evidence for such homeostatic adjustments comes from studies with normally sighted humans that observed a reduction of GABA concentrations in visual cortex (*Lunghi et al., 2015*) and an increase in the BOLD response *Binda et al., 2018* following monocular blindfolding. Further, studies in adult mice have provided support for a homeostatic adjustment of the E/I ratio following prolonged changes in neural activity (*Chen et al., 2022*; *Goel and Lee, 2007*; *Keck et al., 2017*; *Whitt et al., 2014*). For example, a long period of decreased activity following enucleation in adult mice commensurately decreased inhibitory drive (*Keck et al., 2011*), primarily onto excitatory neurons (*Barnes et al., 2015*). In line with the lowered Glx/GABA+ ratio being a compensatory measure to prevent runaway excitation during visual stimulation, the link between the Glx/GABA+ ratio during eye closure and visual acuity in an exploratory correlation analysis suggests that the more successful such assumed downregulation of the E/I ratio in visual cortex, the better the visual recovery. In fact, this correlation with visual acuity recovery is reminiscent of a previously reported correlation in a larger group of CC individuals, between decreased visual cortex thickness and better visual acuity (*Hölig et al., 2023*). Hence, CC individuals with more advanced structural normalization appear to have a better starting point for functional recovery, the latter possibly mediated by physiological plasticity. Yet, future work has to explicitly test these hypotheses.

An increased intercept of the aperiodic component of occipital EEG activity was observed in the same CC individuals who underwent MRS assessment, irrespective of condition, that is, during rest with eyes open and eyes closed, as well as during flickering stimulation. The intercept of the aperiodic component has been linked to overall neuronal spiking activity (*Manning et al., 2009*; *Musall et al., 2014*) and fMRI BOLD activity (*Winawer et al., 2013*). The higher aperiodic intercept may therefore signal increased spontaneous spiking activity in the visual cortex of CC individuals. This interpretation would be consistent with the previously observed increase in visual cortex BOLD activity of CC compared to SC individuals (*Rączy et al., 2022*).

In CC individuals, the intercept of the aperiodic activity was highly correlated with the Glx concentration during rest with eyes open and during flickering stimulation. This exploratory finding needs replication in a larger sample. If reliable, the correlation between the EEG aperiodic intercept and Glx concentration in CC individuals might indicate more broadband firing (*Manning et al., 2009*; *Winawer et al., 2013*) in CC than SC individuals during active and passive visual stimulation.

## Limitations

The sample size of the present study was rather high for rare population of carefully diagnosed CC individuals, but undoubtedly overall small. Access to CC individuals was limited by the constraints of the COVID-19 pandemic. Hence, all the group differences, the exploratory correlations with visual history metrics, and between MRS-EEG parameters, are reported for further investigation in a larger sample. Moreover, our speculative accounts for the present findings need to be validated with pre- and post-surgery assessments. Finally, a comparison of CC individuals with a control group of developmental cataract-reversal individuals would be instrumental to test the hypothesis that the observed group differences are specific to early brain development.

We are aware that MRS and EEG has a low spatial specificity. Moreover, MRS measures do not allow us to distinguish between presynaptic, postsynaptic and vesicular neurotransmitter concentrations. However, all reported group differences in MRS and EEG parameters were specific to visual cortex and were not found for the frontal control voxel or at frontal electrodes, respectively. While data quality was lower for the frontal compared to the visual cortex voxels, as has been observed previously (*Juchem and de Graaf, 2017*; *Rideaux et al., 2022*), this was not an issue for the EEG recordings. Thus, lower sensitivity of frontal measures cannot easily explain the lack of group differences for frontal measures. Crucially, data quality did not differ between groups.

While interpretations of new data in the absence of similar data sets are necessarily speculative, the validity of the neurochemical findings was supported by quality assessments; phantom testing showed high correlations between the experimentally varied metabolite concentrations and the extracted GABA+ and Glx concentrations (Appendix 1.3). The neurochemical results were robust to analysis pipelines (Appendix 1.3) as well as normalization method (Appendix 1.5). The EEG results from the present group of CC individuals replicated effects observed in a larger sample of 28 additional CC individuals (Appendix 1.18; *Ossandón et al., 2023*), as well as prior findings from another sample reporting lower alpha power (Appendix 1.19; *Bottari et al., 2016*; *Pant et al., 2023*). Further, the aperiodic intercept of EEG activity decreased with chronological age irrespective of group or condition, replicating earlier reports (*Hill et al., 2022*; *Voytek et al., 2015*) (Appendix 1.15). Finally, group differences were observed despite the considerable variance of blindness duration and time since surgery, demonstrating the crucial role of early visual experience.

## Conclusion

The present study in sight recovery individuals with a history of congenital blindness indicates that E/I balance is a result of early experience and crucial for human behavior. We provide initial evidence that the E/I ratio in congenital cataract-reversal individuals is altered even years after surgery, which may be due to previous adaptation to congenital blindness.

## Acknowledgements

We thank the technical staff of the Lucid Medical Diagnostics Center, Banjara Hills, Hyderabad, India, in particular Mr. Balakrishna Vaddepally, for technical assistance during collection of MRS/MRI data. We would like to acknowledge Dr. Suddha Sourav for technical support, and Ms. Prativa Regmi for assistance with phantom testing and data collection. We are grateful to D Balasubramanian of the LV Prasad Eye Institute for initiating and supporting our research. The study was funded by the German Research Foundation (DFG Ro 2625/10–1 and SFB 936–178316478-B11) and Landesforschungsförderung (LFF-FV 6) of the Free and Hanseatic City of Hamburg to Brigitte Röder. Rashi Pant was supported by a PhD student fellowship from the Hector Fellow Academy GmbH.

## Additional information

### Competing interests

Sunitha Lingareddy: is the Managing Director Radiology at Lucid Medical Diagnostics, Hyderabad, India. The other authors declare that no competing interests exist.

## Funding

| Funder | Grant reference number | Author |
|---|---|---|
| Hector Fellow Academy | PhD Student Fellowship | Rashi Pant |
| Deutsche Forschungsgemeinschaft | DFG Ro 2625/10-1 | Brigitte Röder |
| Landesforschungsförderung | LFF-FV 6 | Brigitte Röder |
| Deutsche Forschungsgemeinschaft | SFB 936-178316478-B11 | Brigitte Röder |

The funders had no role in study design, data collection and interpretation, or the decision to submit the work for publication.

## Author contributions

Rashi Pant, Conceptualization, Data curation, Formal analysis, Funding acquisition, Visualization, Methodology, Writing – original draft, Project administration, Writing – review and editing; Kabilan Pitchaimuthu, Conceptualization, Data curation, Formal analysis, Supervision, Methodology, Writing – original draft, Project administration, Writing – review and editing; José P Ossandón, Conceptualization, Data curation, Formal analysis, Supervision, Methodology, Writing – review and editing; Idris Shareef, Data curation, Methodology, Project administration, Writing – review and editing; Sunitha Lingareddy, Resources, Project administration, Writing – review and editing; Jürgen Finsterbusch, Conceptualization, Supervision, Validation, Methodology, Writing – review and editing; Ramesh Kekunnaya, Resources, Supervision, Funding acquisition, Validation, Methodology, Project administration, Writing – review and editing; Brigitte Röder, Conceptualization, Resources, Supervision, Funding acquisition, Validation, Investigation, Methodology, Writing – original draft, Project administration, Writing – review and editing

## Author ORCIDs

Rashi Pant ⓘ https://orcid.org/0000-0001-5242-4145
Kabilan Pitchaimuthu ⓘ https://orcid.org/0000-0001-9090-5206
José P Ossandón ⓘ https://orcid.org/0000-0002-2539-390X
Idris Shareef ⓘ https://orcid.org/0000-0001-9258-2199
Ramesh Kekunnaya ⓘ https://orcid.org/0000-0001-5789-2300
Brigitte Röder ⓘ https://orcid.org/0000-0003-3088-8023

## Ethics

All participants (as well as legal guardians for minors) gave written and informed consent. This study was conducted after approval from the Local Ethical Commission (LEK) of the Faculty of Psychology and Human Movement Science (EK-Röder-102015) and the Local Ethical Committee of the Hyderabad Eye Research Foundation (LEC 11-086, LEC-12-15-124).

Reviewer #1 (Public review): https://doi.org/10.7554/eLife.98143.4.sa1
Reviewer #2 (Public review): https://doi.org/10.7554/eLife.98143.4.sa2
Reviewer #3 (Public review): https://doi.org/10.7554/eLife.98143.4.sa3
Author response https://doi.org/10.7554/eLife.98143.4.sa4

# Additional files

## Supplementary files

Supplementary file 1. MRS Minimum Reporting Standards Form as published by *Lin et al., 2021*.
MDAR checklist

## Data availability

Data necessary to replicate the main manuscript figures and results have been made accessible at https://doi.org/10.25592/uhhfdm.17349.

The following dataset was generated:

| Author(s) | Year | Dataset title | Dataset URL | Database and Identifier |
|---|---|---|---|---|
| Pant R, Pitchaimuthu K, Ossandón JP, Shareef I, Lingaredd S, Finsterbusch J, Kekunnaya R, Röder B | 2025 | Data for "Altered visual cortex excitatory/ inhibitory ratio following transient congenital visual deprivation in humans" | https://doi.org/10. 25592/uhhfdm.17349 | UHH Research Data Repository, 10.25592/ uhhfdm.17349 |

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

# Appendix 1

## 1.1. Visual acuity

The lower visual acuity (evidenced by higher logMAR values) in congenital cataract-reversal individuals was expected from a large number of previous reports (*Khanna et al., 2013*). Binocular visual acuity values measured on the day of MRS/EEG testing with the Frieburg Visual Acuity test (*Bach, 2007*) are seen in *Appendix 1—figure 1* and reported in *Table 1* of the Methods.

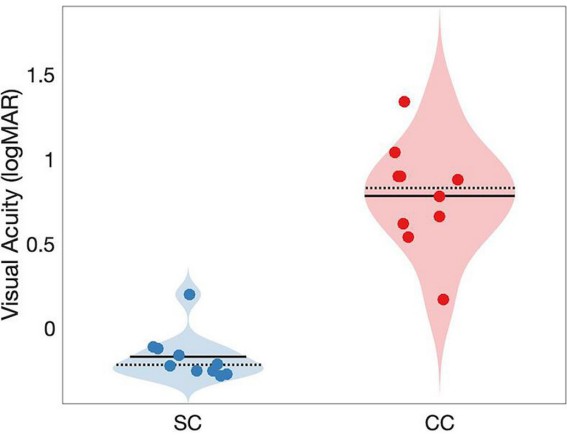

**Appendix 1—figure 1.** Visual acuity in normally sighted individuals (SC) and congenital cataract-reversal (CC individuals). Binocularly measured visual acuity distributions in logarithmic of minimum angle of resolution (logMAR) are displayed as violin plots. Solid black lines indicate mean values, dotted black lines indicate median values.

## 1.2. Percentage overlap of visual cortex MRS voxel with anatomically defined visual cortex region

Percentage of overlap between the visual cortex MRS voxel and an anatomically defined visual cortex region of interest (ROI) was calculated for every subject. First, the visual ROI mask was obtained using the Anatomy toolbox in SPM 12 (*Eickhoff et al., 2005*), including all areas V1-V6 of the occipital lobe. This mask was aligned, co-registered and resliced to every subject's T1 scan. Subsequently, the proportion of vertices in the MRS visual cortex voxel mask (as generated by Gannet 3.0 using SPM 12) overlapping with the T1-aligned occipital lobe mask of each participant was calculated as a percentage of the total number of vertices in the MRS visual cortex voxel. The percentage overlap in both groups did not significantly differ (Mean CC = 67.1%, Mean SC = 70%, t(18) = –1.14, p=0.269).

## 1.3. MRS data analysis using linear combination modeling (Osprey)

In the context of MEGA-PRESS data analysis, a recent study suggested that linear combination modeling offers superior reproducibility compared to peak fitting methods, such as using three gaussian peaks (as implemented in Gannet) (*Hupfeld et al., 2024*). Osprey is an open-source toolbox which uses linear combination modelling for analysis of MEGA-PRESS as well as PRESS datasets (*Oeltzschner et al., 2020*). As our experiment was conceptualized prior to the release of this toolbox, we originally analyzed our data using Gannet 3.0 by the same authors, and TARQUIN for OFF-spectrum analysis. Subsequently, we re-analyzed our data using Osprey v2.6.0 and found that the main findings of the analysis corresponded to those obtained with Gannet and TARQUIN (*Appendix 1—figure 2*, *Appendix 1—figure 3*). While visual cortex GABA+ (Main effect of group F(1,39) = 2.48, p=0.124, $\eta_p^2$=0.06, Main effect of condition F(1,39) = 0.92, p=0.345, $\eta_p^2$=0.02, Group-by-condition interaction F(1,39) = 0.35, p=0.555, $\eta_p^2$<0.01) and Glx concentration (Main effect of group F(1,39) = 2.75, p=0.106, $\eta_p^2$=0.07, Main effect of condition F(1,39) = 0.44, p=0.512, $\eta_p^2$=0.01, Group-by-condition interaction F(1,39) = 1.46, p=0.234, $\eta_p^2$=0.04) did not significantly differ between CC and SC individuals, the Glx/GABA+ concentration ratio was lower in the visual cortex of CC than SC individuals across conditions (Main effect of group F(1,39) = 7.67, p=0.009, $\eta_p^2$=0.17, Main effect of condition F(1,39) = 1.6, p=0.214, $\eta_p^2$=0.04, Group-by-condition interaction F(1,39) = 0.11, p0.743, $\eta_p^2$<0.01; *Appendix 1—figure 2*). NAA concentration did not differ between the visual cortices of CC vs SC individuals, regardless of condition (Main effect of group F(1,39) =

0.93, p=0.342, $\eta_p^2$=0.02, Main effect of condition F(1,39) = 0.53, p=0.471, $\eta_p^2$=0.01, Group-by-condition interaction F(1,39) = 0.14, p=0.714, $\eta_p^2$<0.01; *Appendix 1—figure 3*).

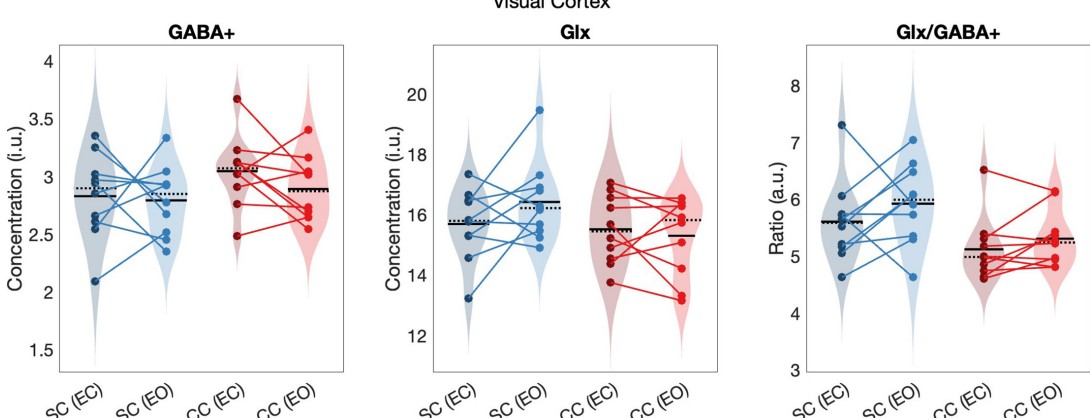

**Appendix 1—figure 2.** Edited (DIFF) spectrum metabolites quantified via Osprey. Water-normalized and tissue corrected GABA+, water-normalized and tissue-corrected Glx, and Glx/GABA+ concentration distributions from the visual cortex are depicted as violin plots for each group and condition (left to right). The solid black lines indicate mean values, and dotted lines indicate median values. The colored lines connect values of individual participants across conditions. Results for congenitally cataract-reversal individuals (CC) and for normally sighted controls (SC) are shown in blue and red, respectively. EC = Eyes closed, EO = Eyes open.,

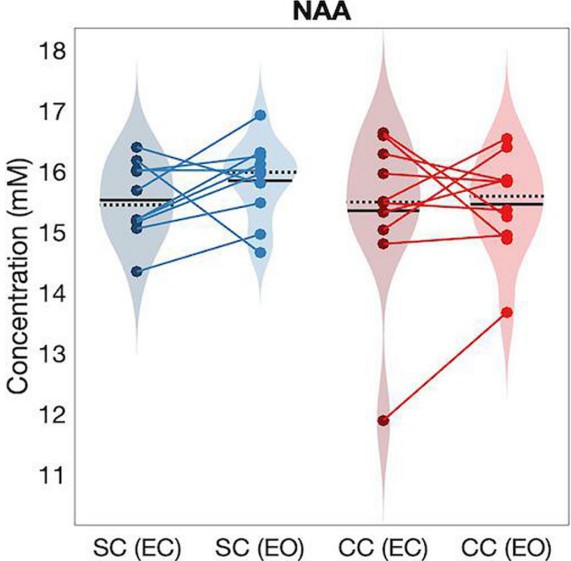

**Appendix 1—figure 3.** OFF spectrum metabolites quantified via Osprey. Water-normalized NAA concentration distributions from the visual cortex are depicted as violin plots for each group and condition (left to right). The solid black lines indicate mean values, and dotted lines indicate median values. The colored lines connect values of individual participants across conditions. For abbreviations see *Appendix 1—figure 2*.

## 1.4. Tissue fractions

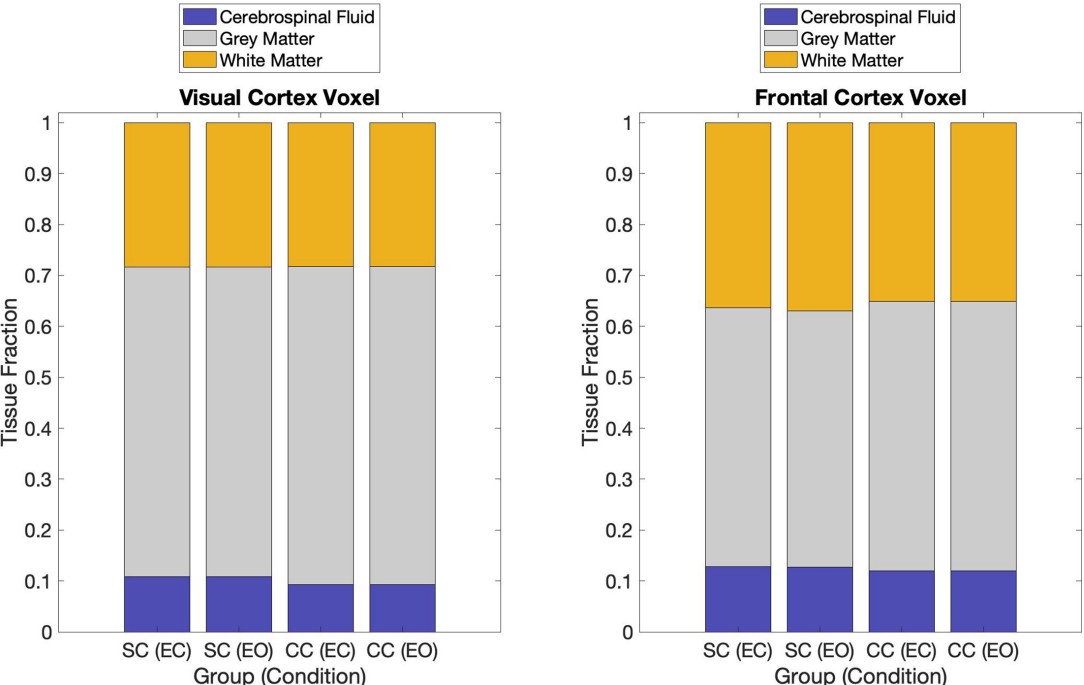

**Appendix 1—figure 4.** Tissue fractions for Magnetic Resonance Spectroscopy voxels. The fractions of white matter (yellow), grey matter (grey) and cerebrospinal fluid (blue) are displayed for the eyes open (EO), and eyes closed (EC) conditions in the congenital cataract-reversal group (CC) and the normally sighted control group (SC). Tissue fractions were separately calculated for the visual (left) and frontal (right) cortex voxels.

## 1.5. Cr-Normalized analysis of MRS data

To ensure that our results were not specific to water-normalized quantification of Glx/GABA+, we reran all analyses with the same pipeline specified in the methods section using Creatine (Cr) normalized GABA+ and Glx quantities across the visual cortex of congenital cataract-reversal (CC) and normally sighted control (SC) individuals. Cr is often used as an internal reference as its concentration is relatively stable in most brain regions. Similar to the water-normalized values, a lower Glx/GABA+ concentration ratio was observed in the visual cortex of CC than SC individuals with Cr-normalization (Main effect of group: $F(1,39) = 5.80$, p=0.021, $\eta_p^2$=0.14), regardless of eye opening or eye closure (Main effect of condition: $F(1,39) = 2.29$, p=0.138, $\eta_p^2$=0.06, Group-by-condition interaction: $F(1,39) = 1.15$, p=0.290, $\eta_p^2$=0.03) (Figure S22). Further, Cr-normalized GABA+ concentration did not differ between groups or conditions (Main effect of group: $F(1,39) = 0.82$, p=0.369, $\eta_p^2$=0.02, Main effect of condition: $F(1,39) = 0.94$, p=0.339, $\eta_p^2$=0.02, Group-by-condition interaction: $F(1,39) = 0.09$, p=0.762, $\eta_p^2$<0.01). Notably, unlike water-normalized Glx values (Results, *Figure 2*), Cr-normalized Glx concentration was lower in the visual cortex of CC than SC individuals (Main effect of group: $F(1,39) = 4.73$, p=0.036, $\eta_p^2$=0.12), regardless of condition (Main effect of condition: $F(1,39) = 0.91$, P=0.346, $\eta_p^2$=0.02, Group-by-condition interaction: $F(1,39) = 0.94$, p=0.339, $\eta_p^2$=0.02) (*Appendix 1—figure 5*). None of the Cr-normalized values differed by group or condition, nor where there any significant group-by-condition interactions in the corresponding frontal cortex comparison (all $F(1,39) < 2.54$, p's>0.119, all $\eta_p^2$<0.07).

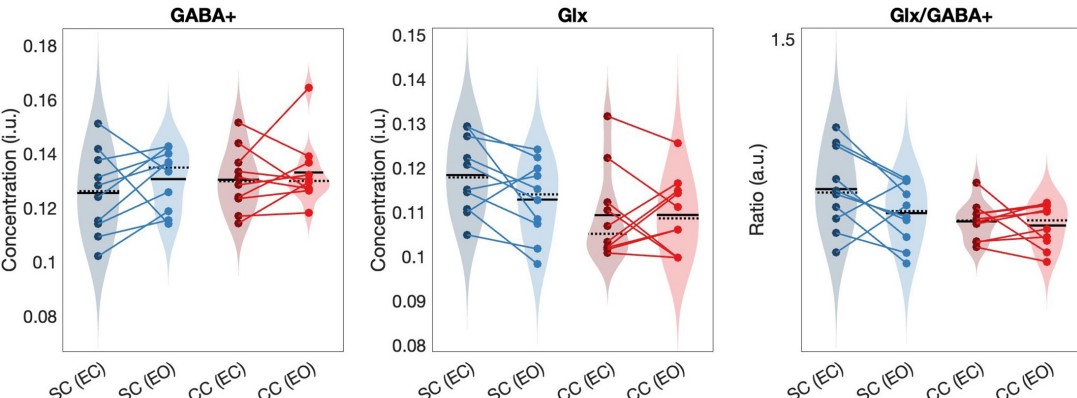

**Appendix 1—figure 5.** Cr-normalized edited (DIFF) spectrum metabolites. Creatine (Cr)-normalized GABA+, Cr-normalized Glx, and Glx/GABA+ concentration distributions from the visual cortex are depicted as violin plots for each group and condition (left to right). The solid black lines indicate mean values, and dotted lines indicate median values. The colored lines connect values of individual participants across conditions.

## 1.6. MRS quality metrics analysis

**Appendix 1—table 1.** ANOVA results for quality metrics on Magnetic Resonance Spectroscopy data.

Quality metrics were compared for each signal (GABA+, Glx and NAA) in a group (congenital cataract-reversal, normally sighted control)-by-region (visual cortex, frontal cortex) ANOVA.

|  |  | Main effect of group | | | Main effect of region | | | Group-by-region interaction | | |
|---|---|---|---|---|---|---|---|---|---|---|
|  |  | $F_{(1,39)}$ | $\eta_p^2$ | p | $F_{(1,39)}$ | $\eta_p^2$ | p | $F_{(1,39)}$ | $\eta_p^2$ | p |
|  | NAA | 0.38 | 0.011 | 0.539 | 232.00 | 0.865 | <0.001 | 0.08 | 0.002 | 0.778 |
|  | GABA+ | 3.37 | 0.084 | 0.080 | 127.12 | 0.779 | <0.001 | 0.01 | <0.001 | 0.936 |
| Signal-to-noise ratio | Glx | 0.39 | 0.011 | 0.534 | 26.75 | 0.426 | <0.001 | <0.001 | <0.001 | 0.989 |
|  | NAA | 1.53 | 0.041 | 0.224 | 247.71 | 0.873 | <0.001 | 2.94 | 0.076 | 0.095 |
|  | GABA+ | 0.09 | 0.002 | 0.765 | 21.71 | 0.376 | <0.001 | 0.56 | 0.015 | 0.457 |
| Full-width half maxima | Glx | 0.20 | 0.005 | 0.660 | 31.56 | 0.467 | <0.001 | 0.11 | 0.003 | 0.743 |
|  | NAA | 1.97 | 0.052 | 0.168 | 38.36 | 0.515 | <0.001 | 3.00 | 0.077 | 0.092 |
|  | GABA+ | 2.78 | 0.070 | 0.104 | 69.14 | 0.657 | <0.001 | 1.65 | 0.043 | 0.206 |
| Fit error | Glx | 0.26 | 0.007 | 0.610 | 12.91 | 0.264 | <0.001 | 0.22 | 0.006 | 0.643 |
| Cramer-Rao lower bound | NAA | 0.05 | 0.001 | 0.821 | 9.34 | 0.206 | 0.004 | 0.09 | 0.002 | 0.760 |

## 1.7. Phantom testing

Phantom testing was conducted to confirm the quality of the MRS data and analysis pipeline. GABA concentration was varied from 0 to 2 mM in a 1 liter, 7.2 pH phantom with fixed metabolite concentrations of Cr (8 mM), NAA (15 mM), Glutamate (12 mM), and Glutamine (3 mM; *Jenkins et al., 2019*). Phosphate Buffer Saline (PBS) was used to maintain the pH at room temperature. The base solution of several metabolites was included to gauge the quality of the overall signal as well as ensuring the similarity of the phantom to metabolites present in vivo (*Jenkins et al., 2019*). The range of used GABA concentrations included the previously reported GABA concentration in the visual cortex of congenitally blind individuals (*Weaver et al., 2013*).

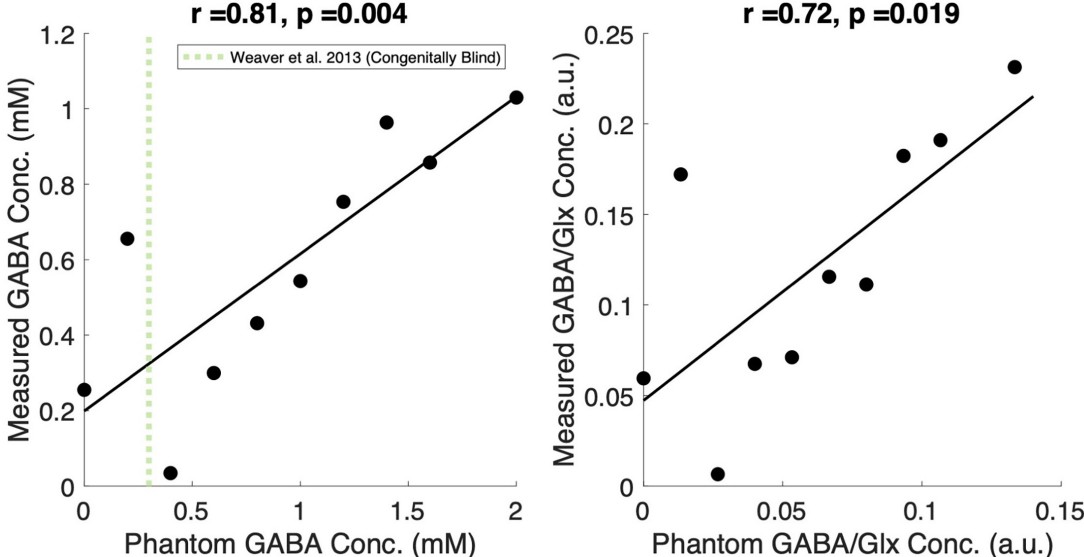

**Appendix 1—figure 6.** Phantom testing of GABA concentrations. Plots depicting the correlation between known and measured concentrations from phantom scans of Gamma-Aminobutyric Acid (GABA; left) and the ratio of GABA to Glutamate/Glutamine (GABA/Glx, right) concentration. In the left panel, previously reported GABA concentration from the visual cortex of congenitally blind individuals (**Weaver et al., 2013**) is marked with a vertical dotted line.

Eleven phantom scans were obtained varying the known concentration of GABA in steps of 0.2 mM (corresponding to 0.0206 g) (**Appendix 1—figure 6**), the reported difference in visual cortex GABA concentration between early blind (mean = 0.3 mM) and normally sighted (mean = 0.5 mM) individuals' visual cortex (**Weaver et al., 2013**). Note that Weaver et al. reported that this group difference did not survive the Bonferroni-Holm correction. Nevertheless, to the best of our knowledge, no other study has reported significant GABA concentration differences between permanently congenitally blind humans and sighted controls based on MRS assessments in humans.

For both GABA and the concentration ratio of GABA/Glx (calculated instead of Glx/GABA due to the 0-GABA concentration solution), our measured values showed significant agreement with the known phantom concentration values (**Appendix 1—figure 6**). These results demonstrate that our data acquisition and analysis pipeline were adequate to identify differences between CC and SC individuals' visual cortices within previously reported concentration ranges.

## 1.8. Individual subjects' MRS edited spectra (visual cortex)

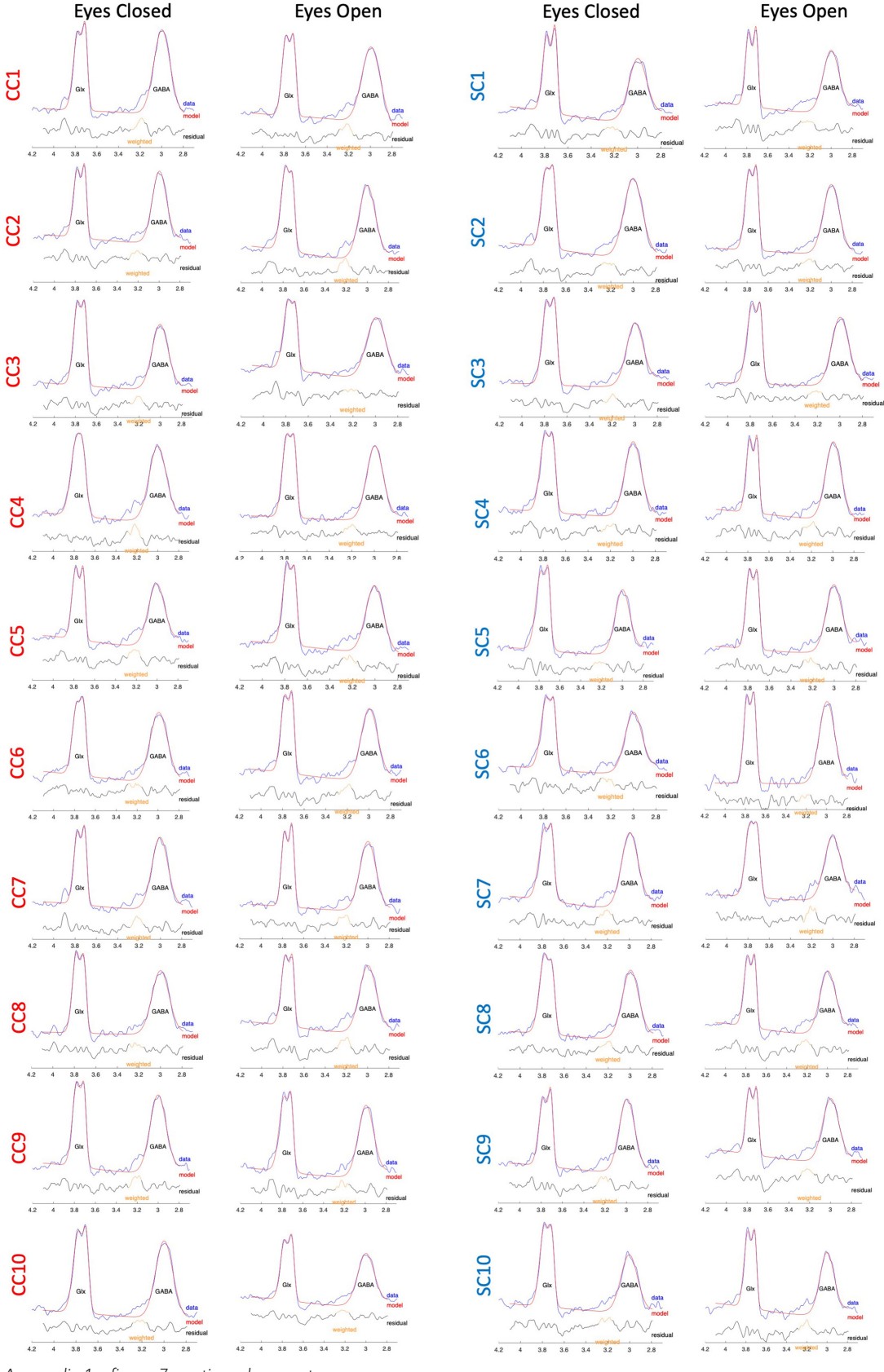

*Appendix 1—figure 7 continued on next page*

**Appendix 1—figure 7.** Edited spectra of participants showing GABA+ and Glx peaks. Individual participants' edited spectra and the respective model fits for congenital cataract-reversal (CC, left) and normally sighted control (SC, right) individuals. Spectra are shown as output by GannetFit.m for the eyes closed and eyes open conditions for each subject.

## 1.9. Tests for normality and homogeneity of data

**Appendix 1—table 2.** Results from the Shapiro-Wilk test for normality within each group and Levene's test for homogeneity of variance across groups, for all dependent variables in the reported analyses.

The assumption of normality or homogeneity of variance was rejected if p was smaller than 0.05.

| Dependent variable | Shapiro-Wilk test for normality | | | | Levene's Test for homogeneity of variance | |
|---|---|---|---|---|---|---|
| | CC (W value) | CC (p-value) | SC (W value) | SC (p-value) | F(1,18) | p-value |
| Aperiodic intercept (EO) | 0.86 | 0.076 | 0.98 | 0.975 | 0.48 | 0.499 |
| Aperiodic slope (EO) | 0.94 | 0.574 | 0.92 | 0.363 | 0.71 | 0.411 |
| Aperiodic intercept (EC) | 0.84 | 0.050 | 0.96 | 0.761 | 0.15 | 0.700 |
| Aperiodic slope (EC) | 0.88 | 0.141 | 0.93 | 0.431 | 0.64 | 0.434 |
| Aperiodic intercept (LU) | 0.94 | 0.526 | 0.95 | 0.650 | 0.00 | 0.993 |
| Aperiodic slope (LU) | 0.96 | 0.810 | 0.88 | 0.121 | 0.15 | 0.700 |
| GABA+ (EC) | 0.94 | 0.515 | 0.90 | 0.204 | 2.27 | 0.149 |
| GABA+ (EO) | 0.97 | 0.913 | 0.97 | 0.864 | 0.60 | 0.450 |
| Glx (EC) | 0.97 | 0.881 | 0.87 | 0.102 | 0.69 | 0.416 |
| Glx (EO) | 0.95 | 0.720 | 0.97 | 0.906 | 3.38 | 0.083 |
| GABA+/Glx (EC) | 0.96 | 0.834 | 0.94 | 0.501 | 2.07 | 0.172 |
| GABA+/Glx (EO) | 0.92 | 0.347 | 0.89 | 0.173 | 1.51 | 0.235 |

## 1.10. Rejected epochs from electroencephalography data

**Appendix 1—table 3.** Mean percentage of rejected epochs in each condition for the congenital cataract reversal (CC) and normally sighted control (SC) groups.

| Group | Eyes open | Eyes closed | Visual stimulation |
|---|---|---|---|
| CC | 13% | 3.1% | 0.2% |
| SC | 18.37% | 2% | 0.1% |

## 1.11. Individual subjects' aperiodic fits

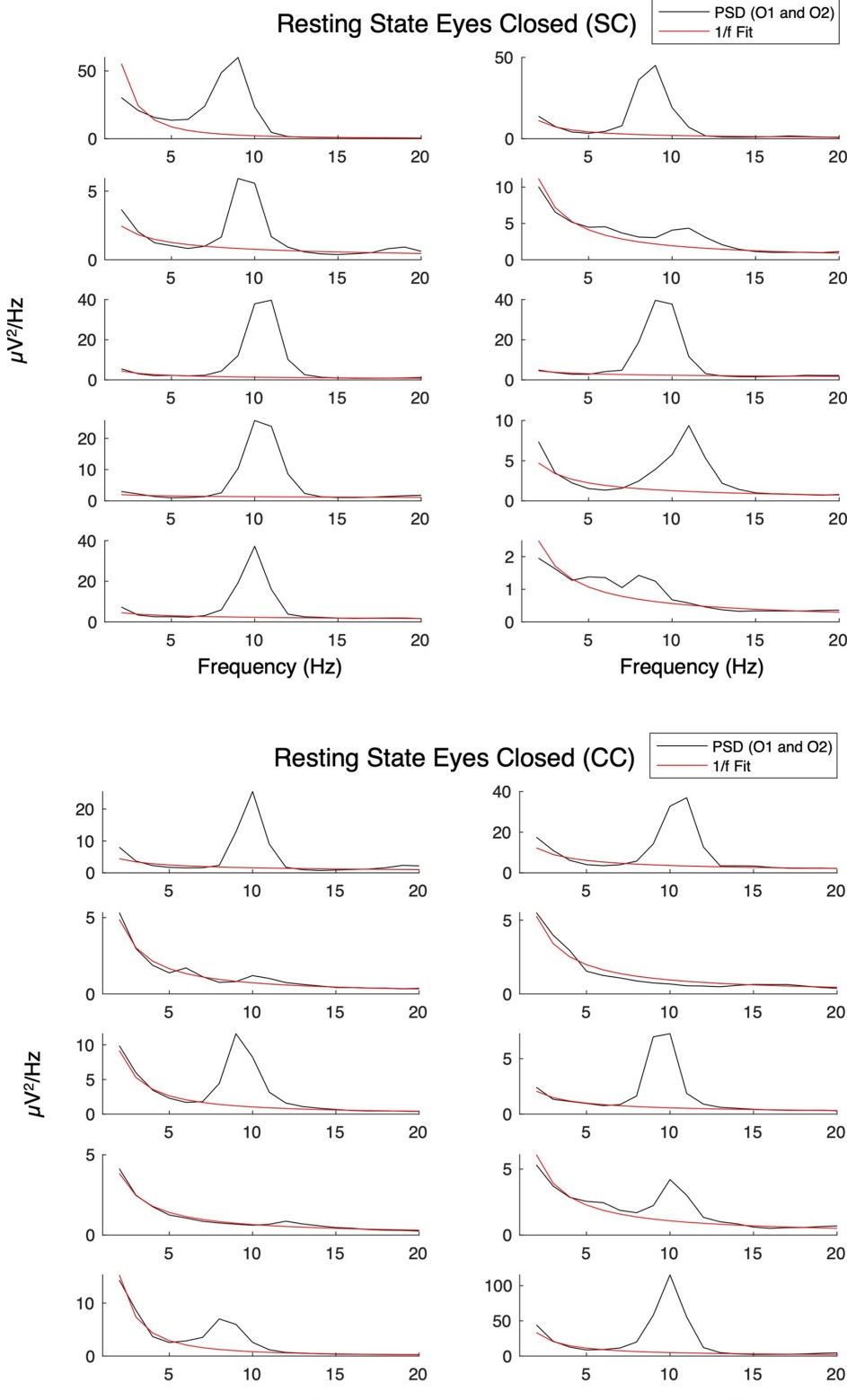

**Appendix 1—figure 8.** Aperiodic fits for normally sighted control (SC, top) and congenital cataract-reversal (CC, bottom) individuals at occipital electrodes during rest with eyes closed. Solid black lines indicate the power spectral density, red lines indicate the aperiodic (1 /f) fit in the 1–20 Hz range, excluding alpha frequencies.

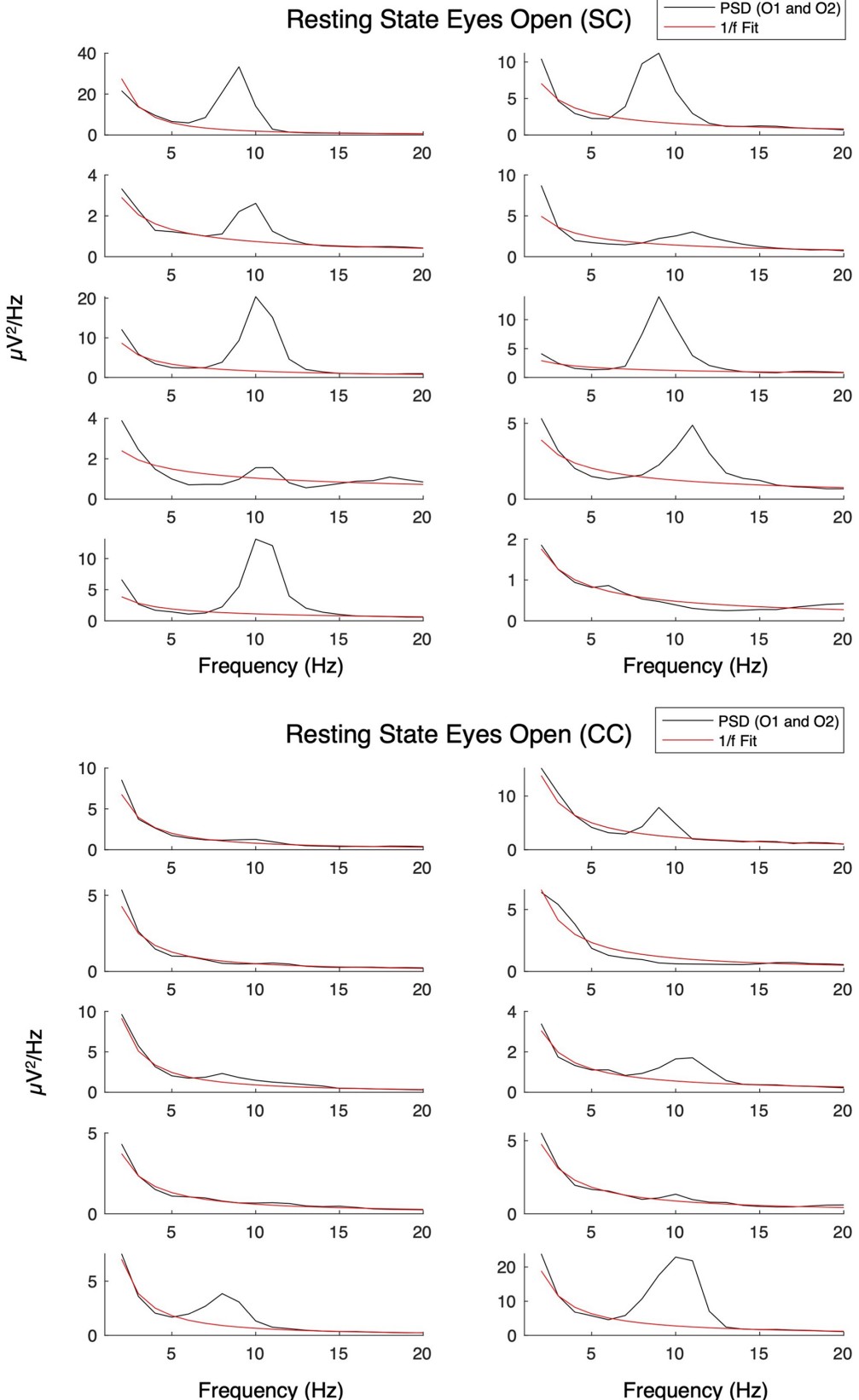

**Appendix 1—figure 9.** Aperiodic fits for normally sighted control (SC, top) and congenital cataract-reversal (CC, bottom) individuals at occipital electrodes during rest with eyes open. Solid black lines indicate the power spectral density, red lines indicate the aperiodic (1 /f) fit in the 1–20 Hz range, excluding alpha frequencies.

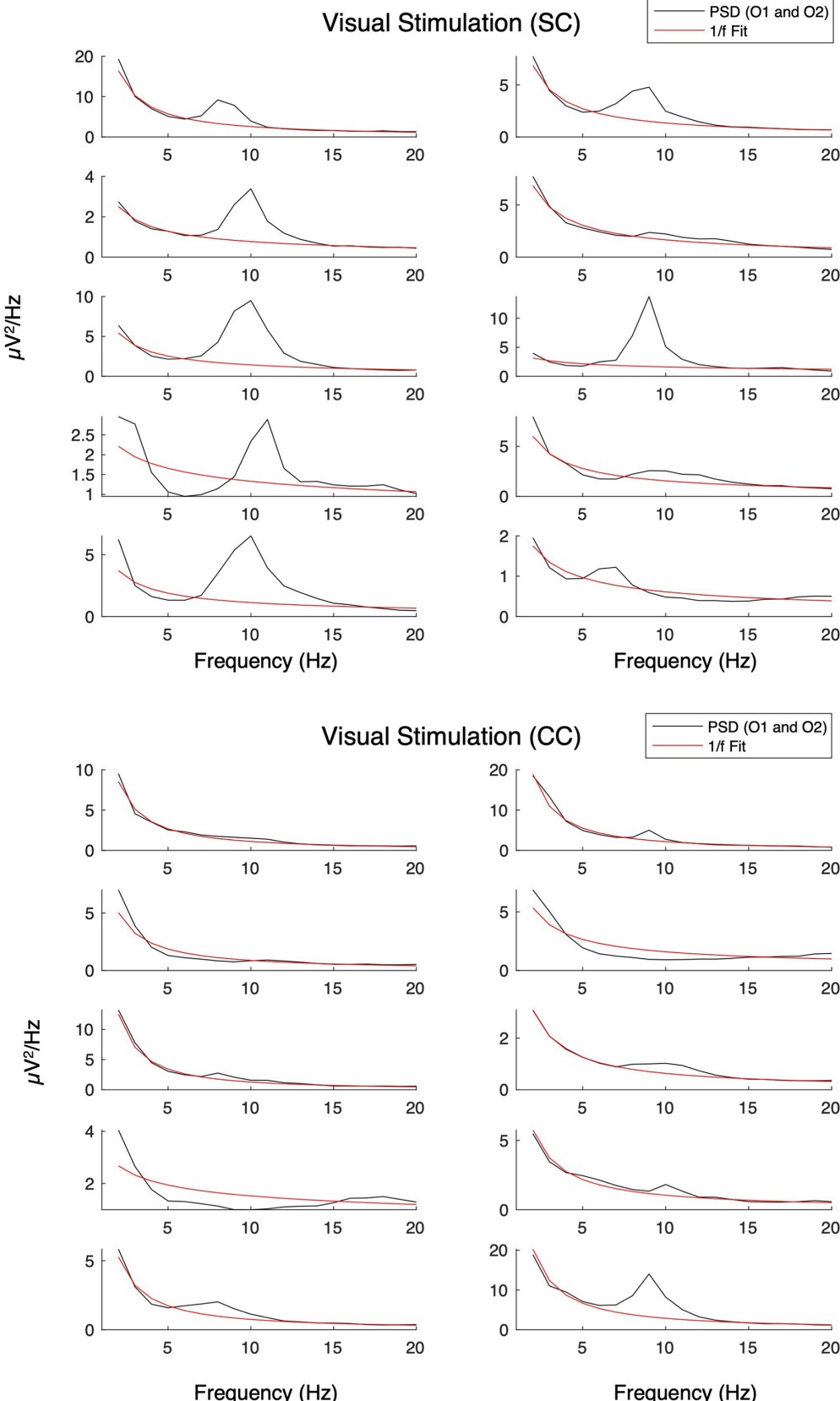

**Appendix 1—figure 10.** Aperiodic fits for normally sighted control (SC, top) and congenital cataract-reversal (CC, bottom) individuals at occipital electrodes during visual stimulation. Solid black lines indicate the power spectral density, red lines indicate the aperiodic (1 /f) fit in the 1–20 Hz range, excluding alpha frequencies.

**Appendix 1—table 4.** Goodness of fit ($R^2$) values for the EEG aperiodic spectrum. Average $R^2$ values are reported for each group and condition.

|  | Eyes closed (EC) | Eyes open (EO) | Visual stimulation (LU) |
|---|---|---|---|
| Sighted control (SC) | 0.96 | 0.96 | 0.91 |
| Congenital cataract reversal (CC) | 0.95 | 0.98 | 0.99 |

## 1.12. Exploratory correlation analysis between MRS measures and visual deprivation history

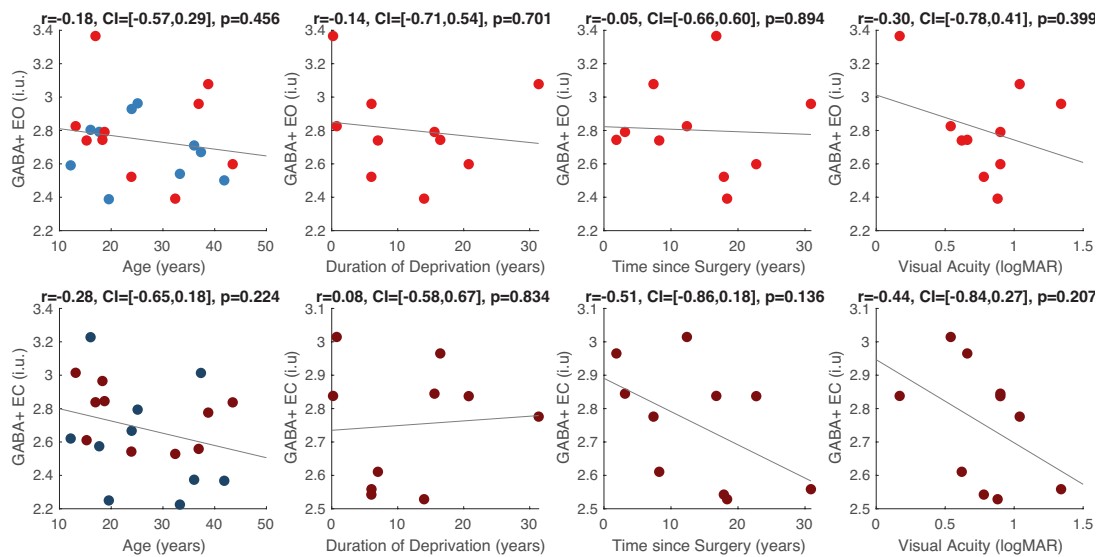

**Appendix 1—figure 11.** Effect of visual deprivation history on GABA+ concentration. Correlations between visual cortex GABA+ concentration and chronological age of the congenital cataract-reversal (CC, red) and normally sighted individuals (SC, blue, see left panel). Second to fourth panels depict correlations between visual cortex GABA+ concentration and duration of visual deprivation, time since surgery and visual acuity in the CC individuals, respectively. Correlations were separately calculated for the eyes open (EO, top row) and eyes closed (EC, bottom row) conditions. The 95% confidence intervals (CI) of the correlation coefficients (r) are reported.

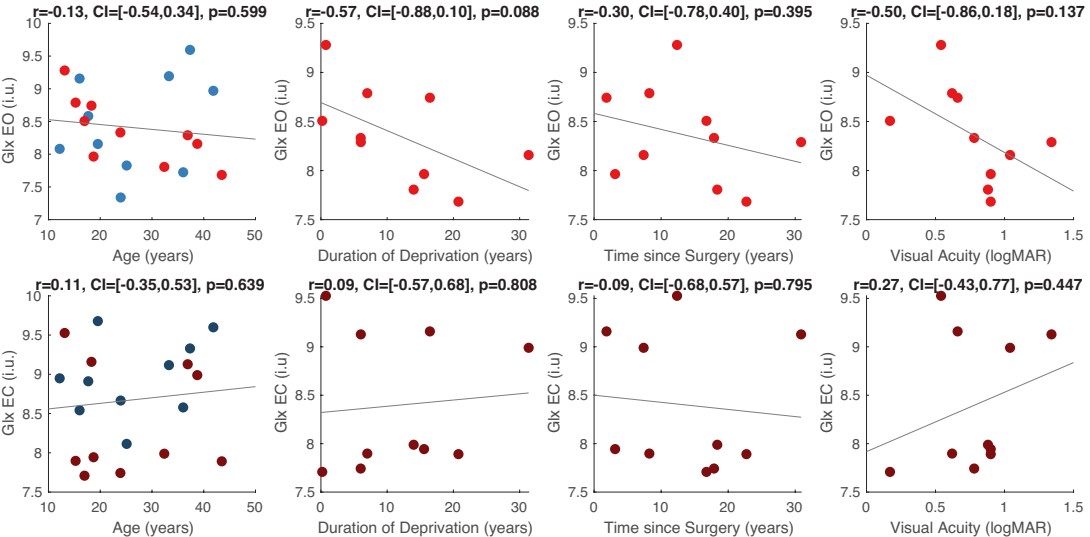

**Appendix 1—figure 12.** Effect of visual deprivation history on Glx concentration. Correlations between visual cortex Glx concentration and chronological age of the congenital cataract-reversal (CC, red) and normally sighted individuals (SC, blue, see left panel). Second to fourth panels depict correlations between visual cortex Glx concentration and duration of visual deprivation, time since surgery and visual acuity in the CC individuals, respectively. Correlations were separately calculated for the eyes open (EO, top row) and eyes closed (EC, bottom row) conditions. The 95% confidence intervals (CI) of the correlation coefficients (r) are reported.

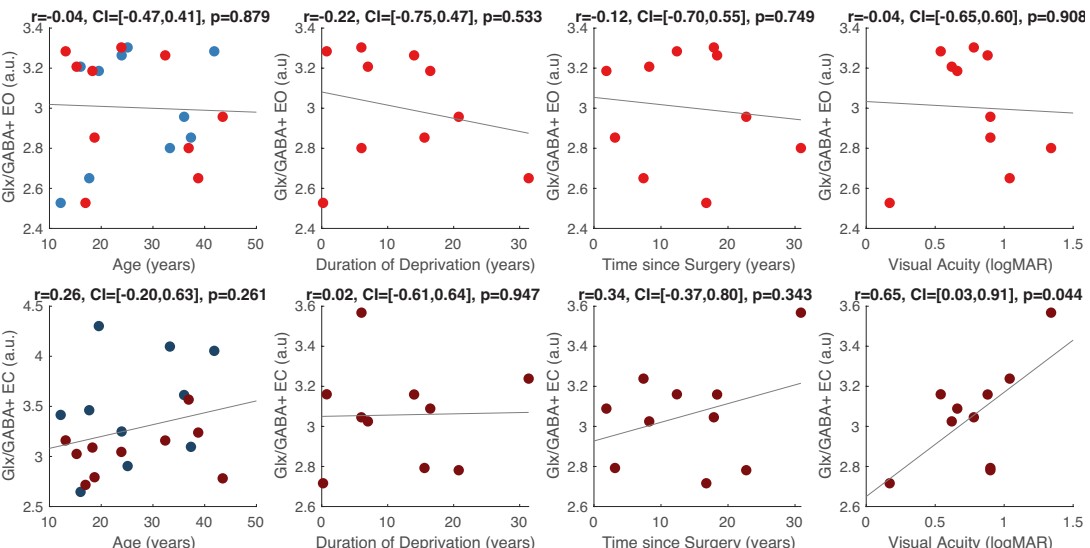

**Appendix 1—figure 13.** Effect of visual deprivation history on Glx/GABA concentration. Correlations between visual cortex Glx/GABA+ concentration and chronological age of the congenital cataract-reversal (CC, red) and normally sighted individuals (SC, blue, see left panel). Second to fourth panels depict correlations between visual cortex Glx/GABA+ concentration and duration of visual deprivation, time since surgery and visual acuity in the CC individuals, respectively. Correlations were separately calculated for the eyes open (EO, top row) and eyes closed (EC, bottom row) conditions. The 95% confidence intervals (CI) of the correlation coefficients (r) are reported.

## 1.13. MRS OFF-spectra

### a. Visual Cortex

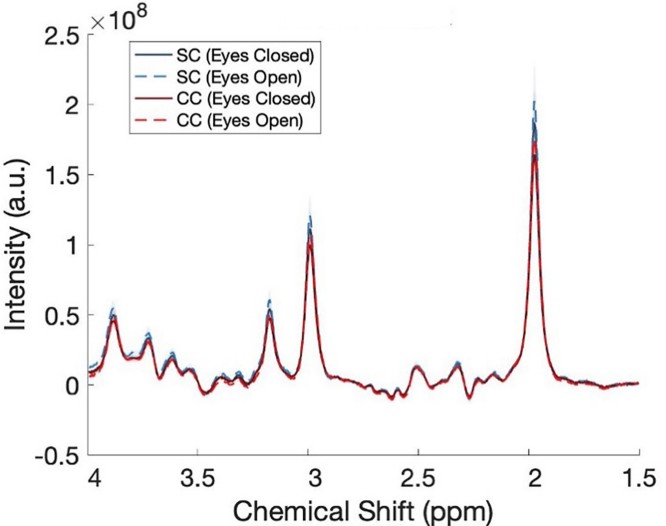
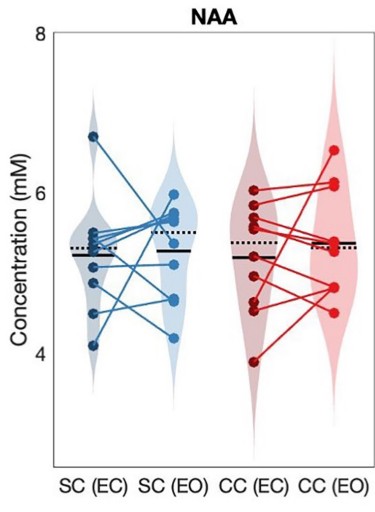

### b. Frontal Cortex

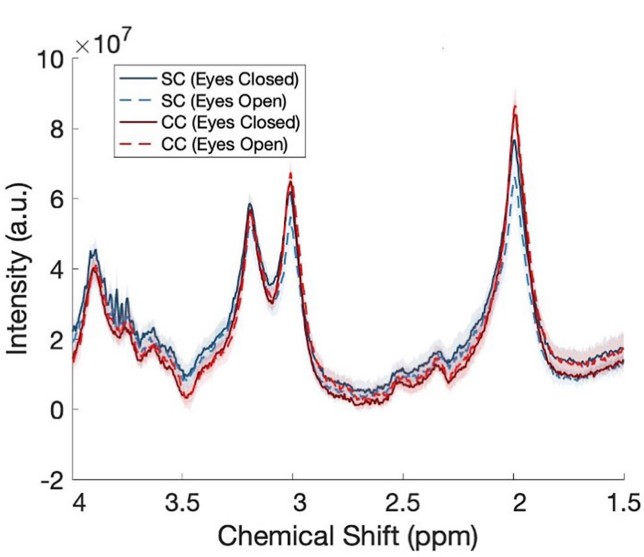
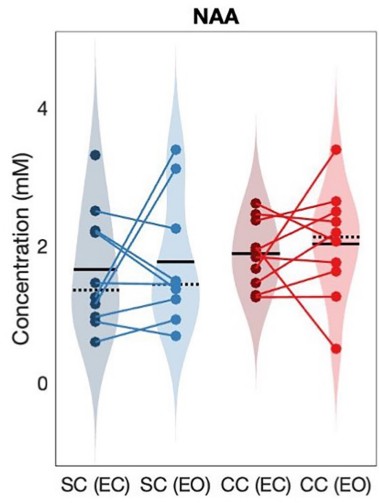

**Appendix 1—figure 14.** OFF spectra obtained from Magnetic Resonance Spectroscopy (MRS). (**a**) The average spectra show NAA peaks in the visual cortices of normally sighted individuals (SC, green) and individuals with reversed congenital cataracts (CC, red) are shown. Spectra are displayed for the eyes open (EO), and eyes closed (EC) conditions. The standard error of the mean is shaded. NAA concentration distributions for each group and condition are demonstrated as violin plots on the right. The solid black lines indicate mean values, and dotted lines indicate median values. The colored lines connect values of individual participants across conditions. (**b**) Corresponding average MRS spectra and NAA concentration distributions measured from the frontal cortex are displayed.

## 1.14. Aperiodic measures across frontal electrodes

To assess the spatial specificity of aperiodic EEG measures, we compared the aperiodic slope and intercept calculated across the frontal electrodes FP1 and FP2 between congenital cataract-reversal (CC) and age-matched sighted control individuals (SC). We found that neither group nor condition significantly predicted the aperiodic offset across frontal electrodes (Main effect of group $F_{(1,59)}$ = 0.11, p=0.746, Main effect of condition $F_{(1,59)}$ = 0.14, p=0.712, Group-by-condition interaction

F(1,59) = 0.05, p=0.885). Moreover, the aperiodic slope did not vary with either group or condition in frontal electrodes (Main effect of group F(1,59) = 0.09, p=0.771, Main effect of condition F(2,59) = 0.93, p=0.400, Group-by-condition interaction F(2,59) = 0.22, p0.801; *Appendix 1—figure 15*).

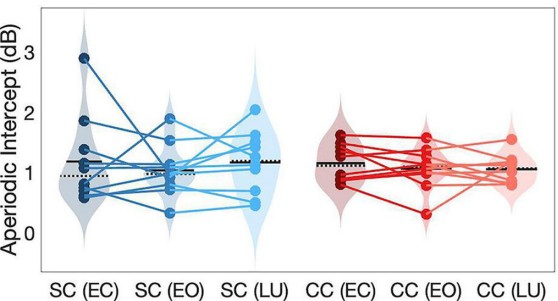

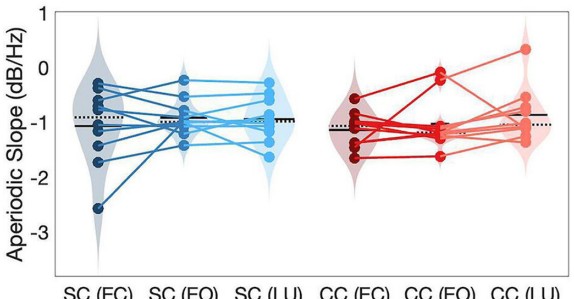

**Appendix 1—figure 15.** Aperiodic intercept (top) and slope (bottom) for congenital cataract-reversal (CC, red) and age-matched normally sighted control (SC, blue) individuals in frontal electrodes. Distributions of these parameters are displayed as violin plots for three conditions; at rest with eyes closed (EC), at rest with eyes open (EO) and during visual stimulation (LU). Aperiodic parameters were calculated across electrodes Fp1 and Fp2. Solid black lines indicate mean values, dotted black lines indicate median values. Colored lines connect values of individual participants across conditions.

## 1.15. Exploratory correlation analysis between the aperiodic slope and intercept of the EEG power spectrum and visual deprivation history

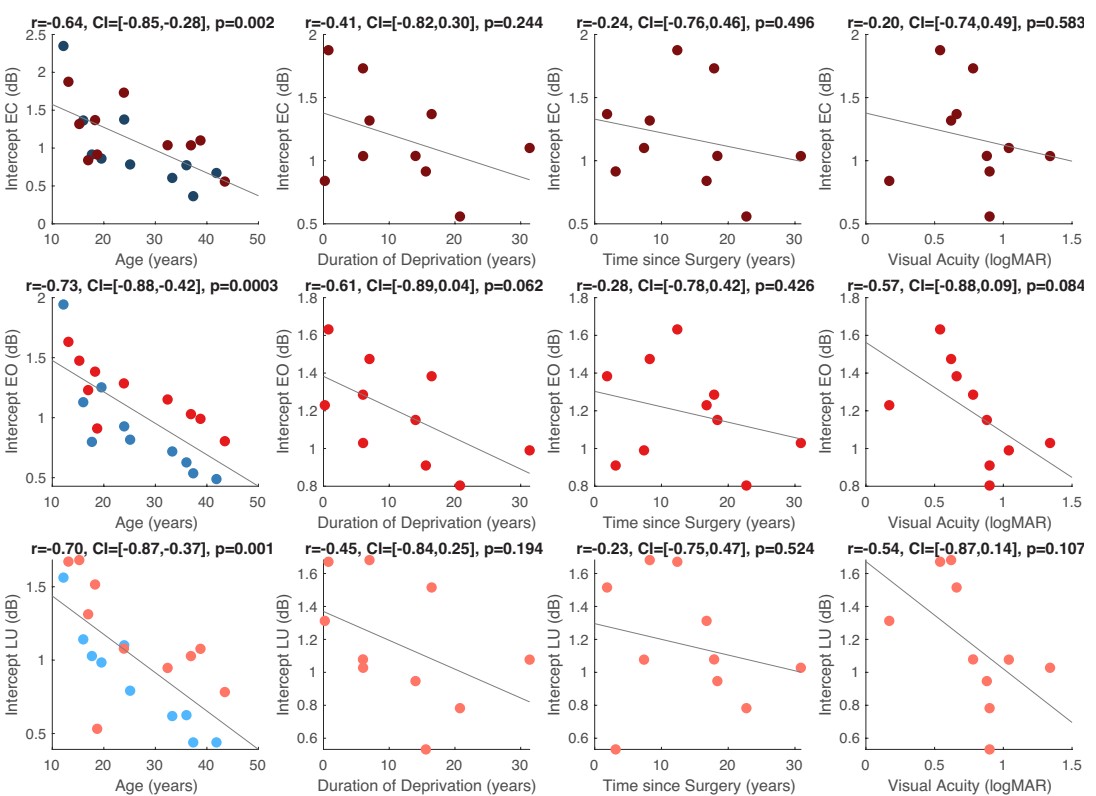

**Appendix 1—figure 16.** Effect of visual deprivation history on aperiodic intercept. Correlations between aperiodic intercept at occipital electrodes and chronological age of the congenital cataract-reversal (CC, red) and normally sighted individuals (SC, blue, see left panel). Second to fourth panels depict correlations between aperiodic intercept and duration of visual deprivation, time since surgery and visual acuity in the CC individuals, respectively. Correlations were separately calculated for the aperiodic intercept while participants viewed stimuli that changed in luminance (LU, top row) and the eyes open (EO, middle row) and eyes closed (EC, bottom row) conditions. The 95% confidence intervals (CI) of the correlation coefficients (r) are reported.

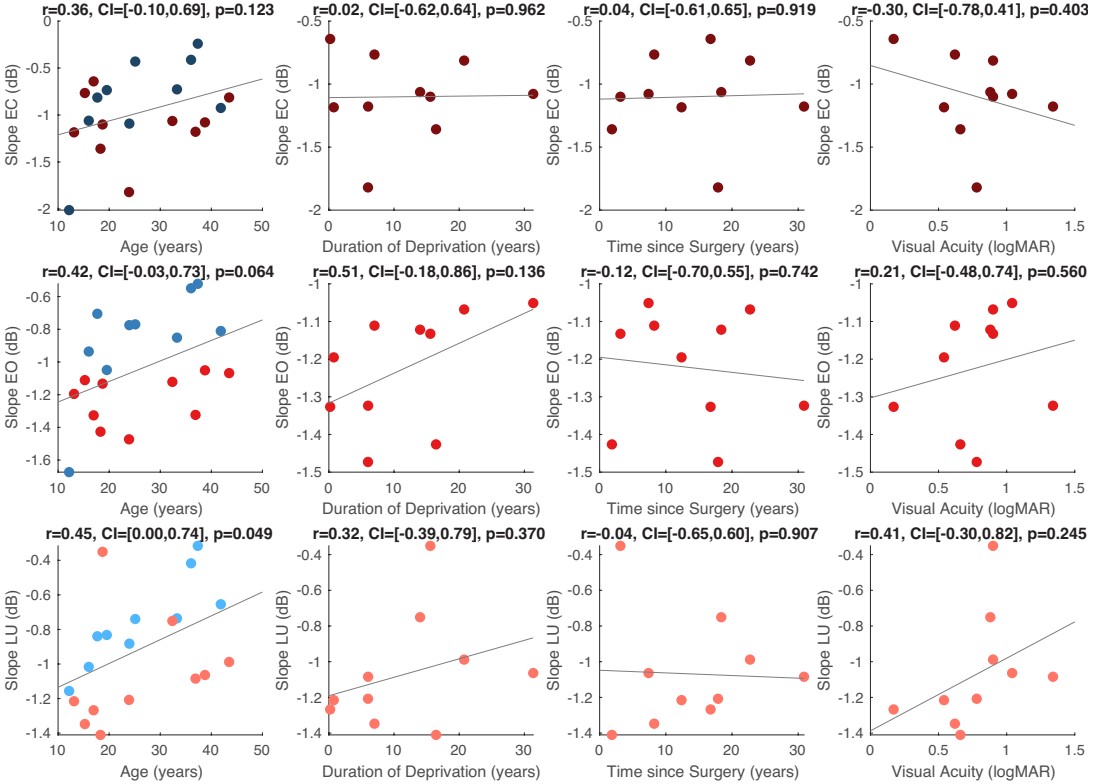

**Appendix 1—figure 17.** Effect of visual deprivation history on aperiodic slope. Correlations between aperiodic slope across occipital electrodes and chronological age of the congenital cataract-reversal (CC, red) and normally sighted individuals (SC, blue, see left panel). Second to fourth panels depict correlations between aperiodic slope and duration of visual deprivation, time since surgery and visual acuity in the CC individuals, respectively. Correlations separately calculated for the aperiodic slope while participants viewed flickering stimuli (LU, top row) and the eyes open (EO, middle row) and eyes closed (EC, bottom row) conditions. The 95% confidence intervals (CI) of the correlation coefficients (r) are reported.

## 1.16. Linear Regression between Glx and aperiodic intercept with age as covariate

A linear regression was conducted within the CC group to predict the aperiodic intercept during visual stimulation, based on age and visual cortex Glx concentration. The results of the regression analysis indicated that the model explained a significant proportion of the variance in the aperiodic intercept, $R^2$=0.82, $t(2,7)$=16.1, $p$=0.0024. Note that the coefficient for age was not significant, β=0.007, t(7)=0.82, $p$=0.439. The regression coefficients and their respective statistics are presented in *Appendix 1—table 5*.

**Appendix 1—table 5.** Regression summary for the effects of Glutamate/Glutamine (Glx) and age on aperiodic intercept (Visual Stimulation) in the congenital cataract reversal (CC) group in the visual stimulation (LU) condition.

| Predictor | Estimate | SE | t | p |
|---|---|---|---|---|
| Model intercept | −5.75 | 1.71 | −3.36 | 0.012 |
| Age | 0.007 | 0.008 | 0.82 | 0.439 |
| Glx | 0.81 | 0.19 | 4.36 | 0.003 |

A second regression was conducted to predict the aperiodic intercept in the CC group during eye opening at rest, based on age and visual cortex Glx concentration. The results of the regression analysis indicated that the model explained a significant proportion of the variance in the aperiodic

intercept, $R^2$=0.842, $t(2,7)$=18.6, $p$=0.00159. Note that the coefficient for age was not significant, $\beta$=−0.005, $t(7)$=−0.90, $p$=0.400. The regression coefficients and their respective statistics are presented in *Appendix 1—table 6*.

**Appendix 1—table 6.** Regression summary for the effects of Glutamate/glutamine (Glx) concentration and age on aperiodic intercept during eye opening at rest (EO) in the congenital cataract reversal (CC) group.

| Predictor | Estimate | SE | t | p |
|---|---|---|---|---|
| Model intercept | −2.07 | 1.11 | −1.86 | 0.106 |
| Age | −0.005 | 0.005 | −0.90 | 0.400 |
| Glx | 0.40 | 0.12 | 3.35 | 0.012 |

Given that the Glx coefficient was significant in both models, and age did not significantly predict either outcome. we concluded that Glx predicted the intercept of the aperiodic intercept.

## 1.17. Exploratory correlation analysis between Electroencephalography and MRS measures

We tested the correlations between Glx, GABA+ and Glx/GABA+ measured at rest, and EEG aperiodic broadband intercept as well as slope in CC and SC individuals, measured at rest and while participants observed a flickering visual stimulus. Below, we report the exploratory correlations prior to Bonferroni correction for 6 comparisons (*Appendix 1—figure 18*; *Appendix 1—figure 22*). Note that the correlation found between the aperiodic slope (1–20 Hz) and Glx concentration (see *Appendix 1—figure 22*) was not significant (all p's>0.219) after correcting for multiple comparisons.

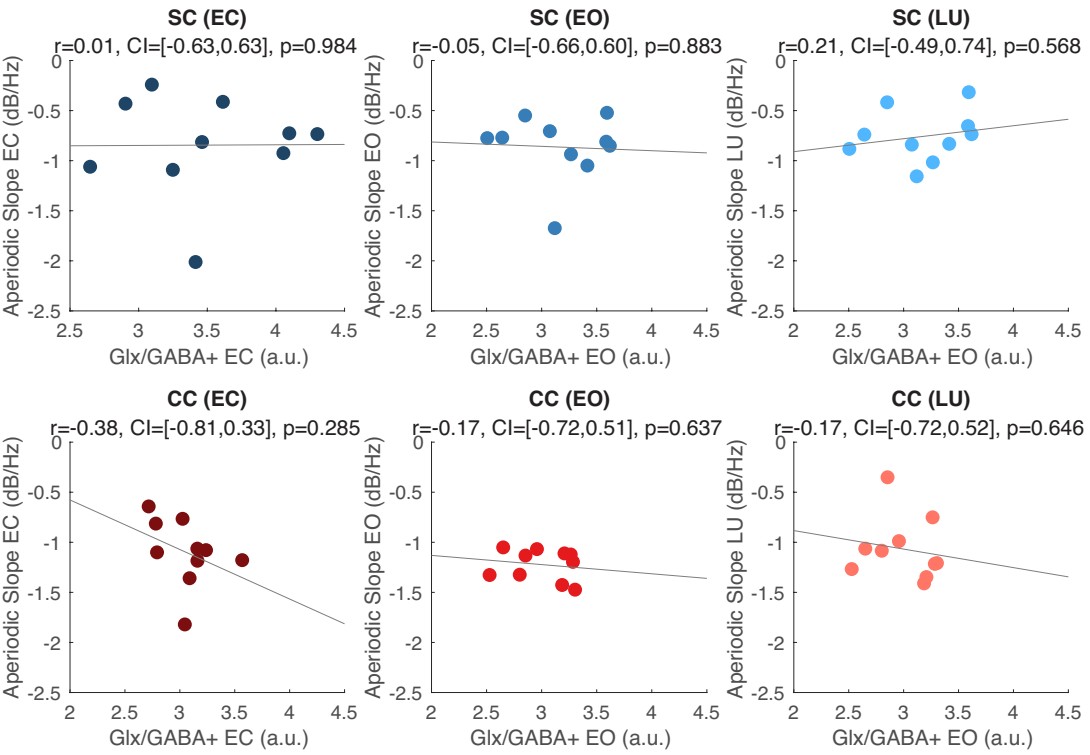

**Appendix 1—figure 18.** Correlation between aperiodic slope and Glx/GABA+ concentration. Correlations between the aperiodic slope and visual cortex Glx/GABA+ concentration measured at rest with eyes closed (EC) (left panels) and eyes open (EO) (middle panels), and the correlation between aperiodic slope measured while subjects viewed flickering stimuli (LU) and visual cortex Glx/GABA+ concentration measured in the EO condition (right panels), are depicted. Correlations were calculated separately for normally sighted control (SC, blue, top row) and congenital cataract-reversal (CC, red, bottom row) individuals. The 95% confidence intervals (CI) of the correlation coefficients (r) are reported.

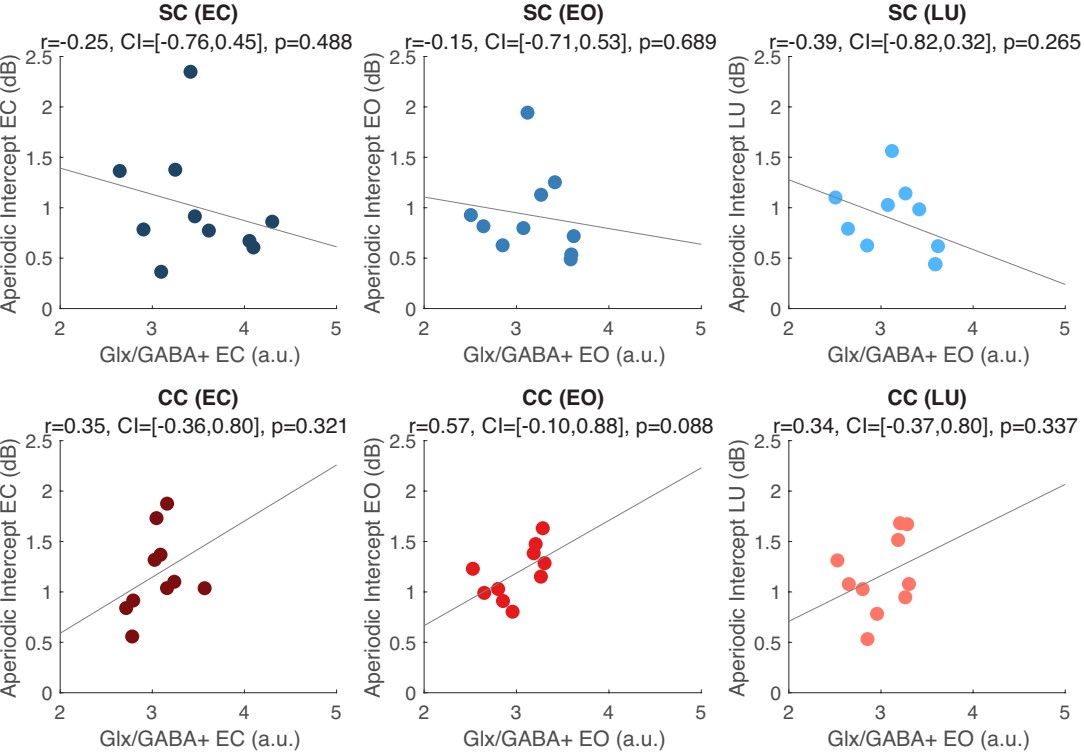

**Appendix 1—figure 19.** Correlation between aperiodic intercept and Glx/GABA+ concentration. Correlations between the aperiodic intercept and visual cortex Glx/GABA+ concentration measured at rest with eyes closed (EC) (left panels) and eyes open (EO) (middle panels), and the correlation between aperiodic intercept measured while subjects viewed flickering stimuli (LU) and visual cortex Glx/GABA+ concentration measured in the EO condition (right panels), are depicted. Correlations were calculated separately for normally sighted control (SC, blue, top row) and congenital cataract-reversal (CC, red, bottom row) individuals. The 95% confidence intervals (CI) of the correlation coefficients (r) are reported.

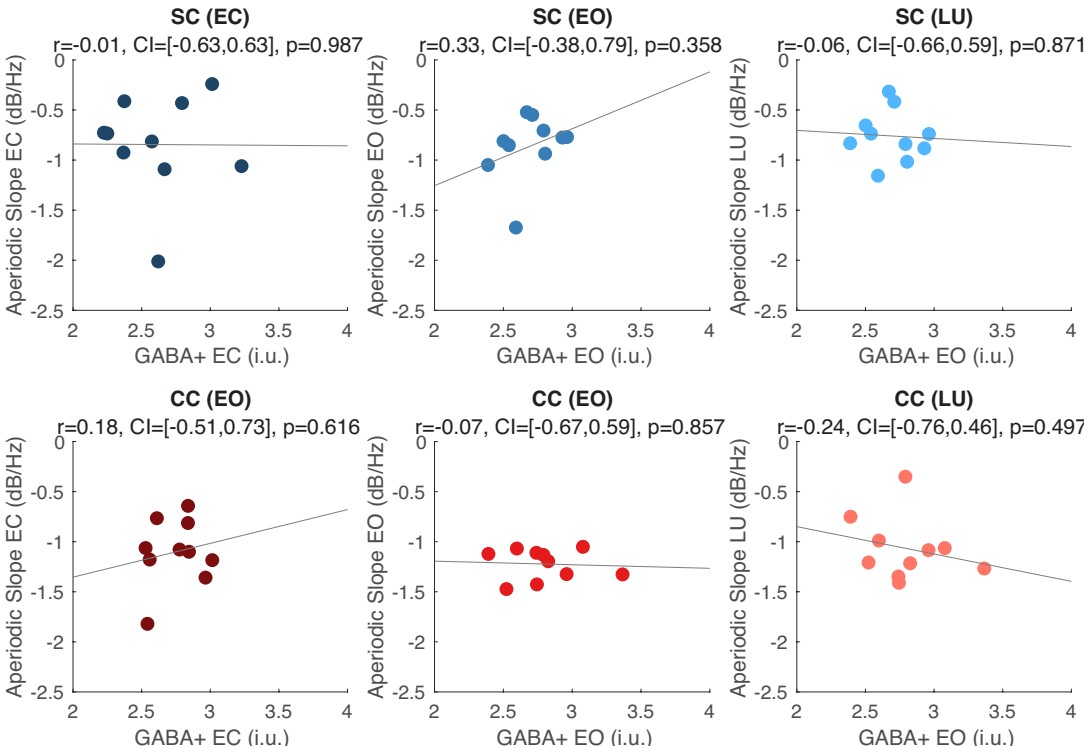

**Appendix 1—figure 20.** Correlation between aperiodic slope and GABA+ concentration. Correlations between the aperiodic slope and visual cortex GABA+ concentration measured at rest with eyes closed (EC) (left panels) and eyes open (EO) (middle panels), and the correlation between aperiodic slope measured while subjects viewed flickering stimuli (LU) and visual cortex GABA+ concentration measured in the EO condition (right panels), are depicted. Correlations were calculated separately for normally sighted control (SC, blue, top row) and congenital cataract-reversal (CC, red, bottom row) individuals. The 95% confidence intervals (CI) of the correlation coefficients (r) are reported.

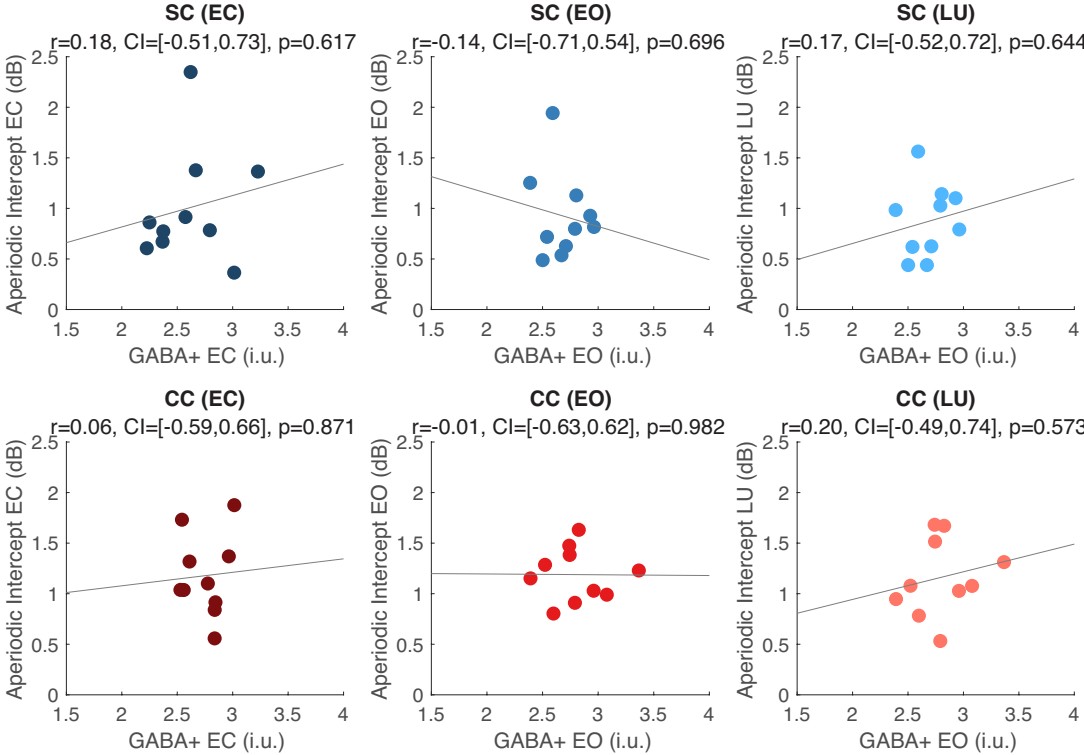

**Appendix 1—figure 21.** Correlation between aperiodic intercept and GABA+ concentration. Correlations between the aperiodic intercept and visual cortex GABA+ concentration measured at rest with eyes closed (EC) (left panels) and eyes open (EO) (middle panels), and the correlation between aperiodic intercept measured while subjects viewed flickering stimuli (LU) and visual cortex GABA+ concentration measured in the EO condition (right panels), are depicted. Correlations were calculated separately for normally sighted control (SC, blue, top row) and congenital cataract-reversal (CC, red, bottom row) individuals. The 95% confidence intervals (CI) of the correlation coefficients (r) are reported.

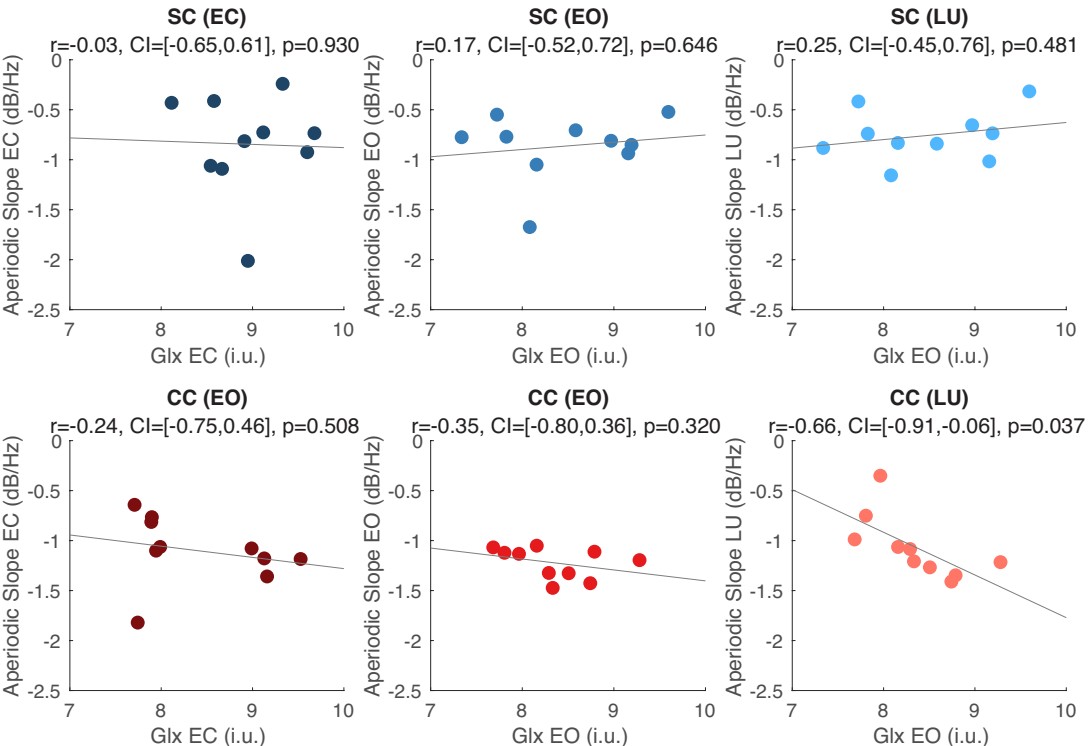

**Appendix 1—figure 22.** Correlation between aperiodic slope and Glx concentration. Correlations between the aperiodic slope and visual cortex Glx concentration measured at rest with eyes closed (EC) (left panels) and eyes open (EO) (middle panels), and the correlation between aperiodic slope measured while subjects viewed flickering stimuli (LU) and visual cortex Glx concentration measured in the EO condition (right panels), are depicted. Correlations were calculated separately for normally sighted control (SC, blue, top row) and congenital cataract-reversal (CC, red, bottom row) individuals. The 95% confidence intervals (CI) of the correlation coefficients (r) are reported.

## 1.18. Correspondence of 1–20 Hz findings with *Ossandón et al., 2023*

The resting-state EEG data from the 10 congenital cataract reversal (CC) individuals in the present study corresponded to that of 28 additional CC subjects tested by *Ossandón et al., 2023*.

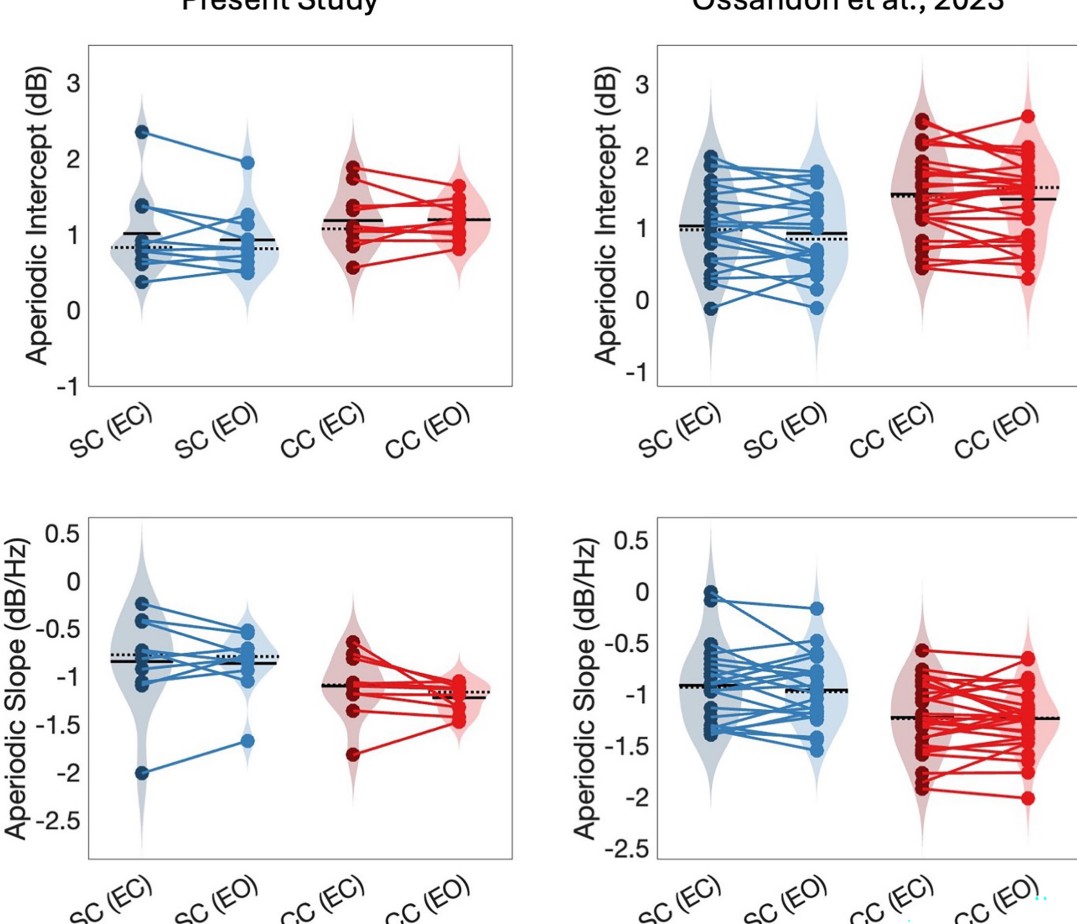

**Appendix 1—figure 23.** Aperiodic offset and slope in the 1–20 Hz range from occipital electrodes in congenital cataract reversal (CC) and normally sighted control (SC) individuals of the present study (left) and additional 28 subjects of *Ossandón et al., 2023*. Aperiodic intercepts (top) and slope (bottom) distributions for each group and condition are displayed as violin plots. Solid black lines indicate mean values, dotted black lines indicate median values. Colored lines connect values of individual participants across conditions.

## 1.19. Alpha amplitude compared between congenital cataract-reversal and sighted control individuals

This dataset is a subset of prior findings of reduced alpha amplitude in congenital cataract-reversal (CC) vs normally sighted control individuals (SC) (*Ossandón et al., 2023*; *Pant et al., 2023*). We tested for differences in alpha amplitude between the 10 CC individuals of the MRS study and their controls and replicated the results of Ossandon et al. and Pant et al., in the present sample (*Appendix 1—figure 24*). An ANOVA revealed that the alpha amplitude was lower in CC than in SC individuals across conditions (main effect of group: $F_{(1,59)} = 8.95$, p=0.004, $\eta_p^2$=0.14, group-by-condition interaction: $F_{(2,59)} = 0.8$, p=0.454, $\eta_p^2$=0.03). As expected, eye closure increased alpha activity compared to eye opening and visual stimulation (main effect of condition: $F_{(2,59)} = 13.12$, p<0.001, $\eta_p^2$=0.33).

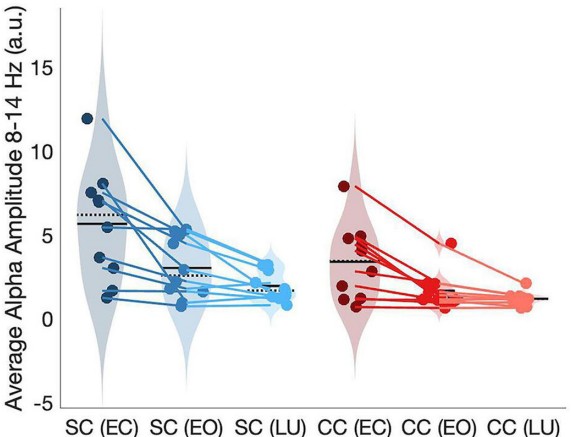

**Appendix 1—figure 24.** Aperiodic-corrected alpha amplitude in congenital cataract-reversal and normally sighted individuals. Aperiodic-corrected alpha amplitudes (8–14 Hz) distributions for each group and condition are displayed as violin plots. Solid black lines indicate mean values, dotted black lines indicate median values. Colored lines connect values of individual participants across conditions.

