## [Editor Report · eLife Assessment]

This neuroimaging and electrophysiology study in a small cohort of congenital cataract patients with sight recovery aims to characterize the effects of early visual deprivation on excitatory and inhibitory balance in visual cortex. While contrasting sight-recovery with visually intact controls suggested the existence of persistent alterations in Glx/GABA ratio and aperiodic EEG signals, it provided **incomplete** evidence supporting claims about the effects of early deprivation itself. The reported data were considered **valuable**, given the rare study population. However, methodological limitations will likely restrict usefulness to scientists working in this particular subfield.

---

## [Referee Report · Reviewer #1 (Public review)]

Summary

In this human neuroimaging and electrophysiology study, the authors aimed to characterise effects of a period of visual deprivation in the sensitive period on excitatory and inhibitory balance in the visual cortex. They attempted to do so by comparing neurochemistry conditions ('eyes open', 'eyes closed') and resting state, and visually evoked EEG activity between ten congenital cataract patients with recovered sight (CC), and ten age-matched control participants (SC) with normal sight. First, they used magnetic resonance spectroscopy to measure in vivo neurochemistry from two locations, the primary location of interest in the visual cortex, and a control location in the frontal cortex. Such voxels are used to provide a control for the spatial specificity of any effects because the single-voxel MRS method provides a single sampling location. Using MR-visible proxies of excitatory and inhibitory neurotransmission, Glx and GABA+ respectively, the authors report no group effects in GABA+ or Glx, no difference in the functional conditions 'eyes closed' and 'eyes open'. They found an effect of group in the ratio of Glx/GABA+ and no similar effect in the control voxel location. They then perform multiple exploratory correlations between MRS measures and visual acuity and report a weak positive correlation between the 'eyes open' condition and visual acuity in CC participants. The same participants then took part in an EEG experiment. The authors selected two electrodes placed in the visual cortex for analysis and report a group difference in an EEG index of neural activity, the aperiodic intercept, as well as the aperiodic slope, considered a proxy for cortical inhibition. Control electrodes in the frontal region did not present with the same pattern. They report an exploratory correlation between the aperiodic intercept and Glx in one out of three EEG conditions.

The authors report the difference in E/I ratio and interpret the lower E/I ratio as representing an adaptation to visual deprivation, which would have initially caused a higher E/I ratio. Although intriguing, the strength of evidence in support of this view is not strong. Amongst the limitations are the low sample size, a critical control cohort that could provide evidence for higher E/I ratio in CC patients without recovered sight for example, and lower data quality in the control voxel. Nevertheless, the study provides a rare and valuable insight into experience-dependent plasticity in the human brain.

Strengths of study

How sensitive period experience shapes the developing brain is an enduring and important question in neuroscience. This question has been particularly difficult to investigate in humans. The authors recruited a small number of sight-recovered participants with bilateral congenital cataracts to investigate the effect of sensitive period deprivation on the balance of excitation and inhibition in the visual brain using measures of brain chemistry and brain electrophysiology. The research is novel, and the paper was interesting and well-written.

Limitations

Low sample size. Ten for CC and ten for SC, and further two SC participants were rejected due to lack of frontal control voxel data. The sample size limits the statistical power of the dataset and increases the likelihood of effect inflation.

In the updated manuscript, the authors have provided justification for their sample size by pointing to prior studies and the inherent difficulties in recruiting individuals with bilateral congenital cataracts. Importantly, this highlights the value the study brings to the field while also acknowledging the need to replicate the effects in a larger cohort.

Lack of specific control cohort. The control cohort has normal vision. The control cohort is not specific enough to distinguish between people with sight loss due to different causes and patients with congenital cataracts with co-morbidities. Further data from a more specific populations, such as patients whose cataracts have not been removed, with developmental cataracts, or congenitally blind participants, would greatly improve the interpretability of the main finding. The lack of a more specific control cohort is a major caveat that limits a conclusive interpretation of the results.

In the updated version, the authors have indicated that future studies can pursue comparisons between congenital cataract participants and cohorts with later sight loss.

MRS data quality differences. Data quality in the control voxel appears worse than in the visual cortex voxel. The frontal cortex MRS spectrum shows far broader linewidth than the visual cortex (Supplementary Figures). Compared to the visual voxel, the frontal cortex voxel has less defined Glx and GABA+ peaks; lower GABA+ and Glx concentrations, lower NAA SNR values; lower NAA concentrations. If the data quality is a lot worse in the FC, then small effects may not be detectable.

In the updated version, the authors have added more information that informs the reader of the MRS quality differences between voxel locations. This increases the transparency of their reporting and enhances the assessment of the results.

Because of the direction of the difference in E/I, the authors interpret their findings as representing signatures of sight improvement after surgery without further evidence, either within the study or from the literature. However, the literature suggests that plasticity and visual deprivation drives the E/I index up rather than down. Decreasing GABA+ is thought to facilitate experience dependent remodelling. What evidence is there that cortical inhibition increases in response to a visual cortex that is over-sensitised to due congenital cataracts? Without further experimental or literature support this interpretation remains very speculative.

The updated manuscript contains key reference from non-human work to justify their interpretation.

Heterogeneity in patient group. Congenital cataract (CC) patients experienced a variety of duration of visual impairment and were of different ages. They presented with co-morbidities (absorbed lens, strabismus, nystagmus). Strabismus has been associated with abnormalities in GABAergic inhibition in the visual cortex. The possible interactions with residual vision and confounds of co-morbidities are not experimentally controlled for in the correlations, and not discussed.

The updated document has addressed this caveat.

Multiple exploratory correlations were performed to relate MRS measures to visual acuity (shown in Supplementary Materials), and only specific ones shown in the main document. The authors describe the analysis as exploratory in the 'Methods' section. Furthermore, the correlation between visual acuity and E/I metric is weak, not corrected for multiple comparisons. The results should be presented as preliminary, as no strong conclusions can be made from them. They can provide a hypothesis to test in a future study.

This has now been done throughout the document and increases the transparency of the reporting.

P.16 Given the correlation of the aperiodic intercept with age ("Age negatively correlated with the aperiodic intercept across CC and SC individuals, that is, a flattening of the intercept was observed with age"), age needs to be controlled for in the correlation between neurochemistry and the aperiodic intercept. Glx has also been shown to negatively correlates with age.

This caveat has been addressed in the revised manuscript.

Multiple exploratory correlations were performed to relate MRS to EEG measures (shown in Supplementary Materials), and only specific ones shown in the main document. Given the multiple measures from the MRS, the correlations with the EEG measures were exploratory, as stated in the text, p.16, and in Fig.4. yet the introduction said that there was a prior hypothesis "We further hypothesized that neurotransmitter changes would relate to changes in the slope and intercept of the EEG aperiodic activity in the same subjects." It would be great if the text could be revised for consistency and the analysis described as exploratory.

This has been done throughout the document and increases the transparency of the reporting.

The analysis for the EEG needs to take more advantage of the available data. As far as I understand, only two electrodes were used, yet far more were available as seen in their previous study (Ossandon et al., 2023). The spatial specificity is not established. The authors could use the frontal cortex electrode (FP1, FP2) signals as a control for spatial specificity in the group effects, or even better, all available electrodes and correct for multiple comparisons. Furthermore, they could use the aperiodic intercept vs Glx in SC to evaluate the specificity of the correlation to CC.

This caveat has been addressed. The authors have added frontal electrodes to their analysis, providing an essential regional control for the visual cortex location.

Comments on revisions:

In the first revision, the authors made reasonable adjustments to their manuscript that addressed most of my comments by adding further justification for their methodology, essential literature support, pointing out exploratory analyses, limitations and adding key control analyses. Their revised manuscript was overall improved, providing valuable information, though the evidence that supports their claims is still incomplete.

In their second revision, the authors pointed to justifications for their analyses, careful interpretation and tempered claims to clarify their response to the initial feedback. However, my assessment of the first revision has not been changed after the second revision, because there were no further modifications of their responses to my feedback.

---

## [Referee Report · Reviewer #2 (Public review)]

Summary:

The study examined 10 congenitally blind patients who recovered vision through the surgical removal of bilateral dense cataracts, measuring neural activity and neuro chemical profiles from the visual cortex. The declared aim is to test whether restoring visual function after years of complete blindness impacts excitation/inhibition balance in the visual cortex. The manuscript reports precious behavioural, electrophysiological and magnetic resonance data from a rare population. Although the findings are useful for stimulating further research in the field, they only provide incomplete support to the authors' claims.

The main claim is that sight recovery impacts the excitation/inhibition balance in the visual cortex; however, the paradigm does not allow to distinguish the effects of sight recovery from those of visual deprivation (i.e. in patients who were born blind but recovered vision after several months/years vs. patients who were born blind and never recovered vision); moreover, the link between electrophysiological findings and cortical excitation/inhibition is tentative and its interpretation remains speculative.

Strengths:

The findings are undoubtedly useful for the community, as they contribute towards characterising the many ways in which this special population differs from normally sighted individuals. The combination of MRS and EEG measures is a promising strategy to estimate a fundamental physiological parameter - the balance between excitation and inhibition in the visual cortex, which animal studies show to be heavily dependent upon early visual experience. Thus, the reported results pave the way for further studies, which may use a similar approach to evaluate more patients and control groups.

Weaknesses:

The main methodological limitation is the lack of an appropriate comparison group or condition to delineate the effect of sight recovery (as opposed to the effect of congenital blindness). Few previous studies suggested that Excitation/Inhibition ratio in the visual cortex is increased in congenitally blind patients; the present study reports that E/I ratio decreases instead. The authors claim that this implies a change of E/I ratio following sight recovery. However, supporting this claim would require showing a shift of E/I after vs. before the sight-recovery surgery, or at least it would require comparing patients who did and did not undergo the sight-recovery surgery (as common in the field).

There are also more technical limitations related to the correlation analyses, which are partly acknowledged in the manuscript. A bland correlation between GLX/GABA and the visual impairment is reported, but this is specific to the patients group (N=10) and would not hold across groups (the correlation is positive, predicting the lowest GLX/GABA ratio values for the sighted controls - opposite of what is found). There is also a strong correlation between GLX concentrations and the EEG power at the lowest temporal frequencies. Although this relation is intriguing, it only holds for a very specific combination of parameters (of the many tested): only with eyes open, only in the patients group.

Conclusions:

The main claim of the study is that sight recovery impacts the excitation/inhibition balance in the visual cortex, estimated with MRS or through indirect EEG indices. However, due to the weaknesses outlined above, the study cannot distinguish the effects of sight recovery from those of visual deprivation. Moreover, many aspects of the results are interesting but their validation and interpretation require additional experimental work.

Comments on revisions:

The authors' revisions did not substantially alter the manuscript. As such, my assessment above remains unaltered.

---

## [Referee Report · Reviewer #3 (Public review)]

Summary:

This manuscript examines the impact of congenital visual deprivation on the excitatory/inhibitory (E/I) ratio in the visual cortex using Magnetic Resonance Spectroscopy (MRS) and electroencephalography (EEG) in individuals whose sight was restored. Ten individuals with reversed congenital cataracts were compared to age-matched, normally sighted controls, assessing the cortical E/I balance and its interrelationship and to visual acuity. The study reveals that the Glx/GABA ratio in the visual cortex and the intercept and aperiodic signal are significantly altered in those with a history of early visual deprivation, suggesting persistent neurophysiological changes despite visual restoration. First of all, I would like to disclose that I am not an expert in congenital visual deprivation, nor in MRS. My expertise is in EEG (particularly in the decomposition of periodic and aperiodic activity) and statistical methods. Second, although the authors addressed some of my concerns on the previous version of this manuscript, major concerns and flaws remain in terms of methodological and statistical approaches along with the (over) interpretation of the results.

Persistent specific concerns include:

(1 3.1) Response to Variability in Visual Deprivation

Rather than listing the advantages and disadvantages of visual deprivation, I recommend providing at least a descriptive analysis of how the duration of visual deprivation influenced the measures of interest. This would enhance the depth and relevance of the discussion.

(2 3.2) Small Sample Size

The issue of small sample size remains problematic. The justification that previous studies employed similar sample sizes does not adequately address the limitation in the current study. I strongly suggest that the correlation analyses should not feature prominently in the main manuscript or the abstract, especially if the discussion does not substantially rely on these correlations. Please also revisit the recommendations made in the section on statistical concerns.

(3 3.3) Statistical Concerns

While I appreciate the effort of conducting an independent statistical check, it merely validates whether the reported statistical parameters, degrees of freedom (df), and p-values are consistent. However, this does not address the appropriateness of the chosen statistical methods.

Several points require clarification or improvement:

(4) Correlation Methods: The manuscript does not specify whether the reported correlation analyses are based on Pearson or Spearman correlation.

This has been addressed in the final revision

(5) Confidence Intervals: Include confidence intervals for correlations to represent the uncertainty associated with these estimates.

This has been addressed in the final revision

(6) Permutation Statistics: Given the small sample size, I recommend using permutation statistics, as these are exact tests and more appropriate for small datasets.

(7) Adjusted P-Values: Ensure that reported Bonferroni corrected p-values (e.g., p > 0.999) are clearly labeled as adjusted p-values where applicable.

This has been addressed in the final revision

(8) Figure 2C

Figure 2C still lacks crucial information that the correlation between Glx/GABA ratio and visual acuity was computed solely in the control group (as described in the rebuttal letter). Why was this analysis restricted to the control group? Please provide a rationale.

(9 3.4) Interpretation of Aperiodic Signal

Relying on previous studies to interpret the aperiodic slope as a proxy for excitation/inhibition (E/I) does not make the interpretation more robust.

(10) Additionally, the authors state:

"We cannot think of how any of the exploratory correlations between neurophysiological measures and MRS measures could be accounted for by a difference e.g. in skull thickness."

(11) This could be addressed directly by including skull thickness as a covariate or visualizing it in scatterplots, for instance, by representing skull thickness as the size of the dots.

(12 3.5) Problems with EEG Preprocessing and Analysis

Downsampling: The decision to downsample the data to 60 Hz "to match the stimulation rate" is problematic. This choice conflates subsequent spectral analyses due to aliasing issues, as explained by the Nyquist theorem. While the authors cite prior studies (Schwenk et al., 2020; VanRullen & MacDonald, 2012) to justify this decision, these studies focused on alpha (8-12 Hz), where aliasing is less of a concern compared of analyzing aperiodic signal. Furthermore, in contrast, the current study analyzes the frequency range from 1-20 Hz, which is too narrow for interpreting the aperiodic signal asE/I. Typically, this analysis should include higher frequencies, spanning at least 1-30 Hz oreven 1-45 Hz (not 20-40 Hz).

(13) Baseline Removal: Subtracting the mean activity across an epoch as a baseline removal step is inappropriate for resting-state EEG data. This preprocessing step undermines the validity of the analysis. The EEG dataset has fundamental flaws, many of which were pointed out in the previous review round but remain unaddressed. In its current form, the manuscript falls short of standards for robust EEG analysis.

(14) The authors mention: "The EEG data sets reported here were part of data published earlier (Ossandón et al.,2023; Pant et al., 2023)." Thus, the statement "The group differences for the EEG assessments corresponded to those of a larger sample of CC individuals (n=38) " is a circular argument and should be avoided."

The authors addressed this comment and adjusted the statement. However, I do not understand, why the full sample published earlier (Ossandón et al., 2023) was not used in the current study?

Comments on revisions:

The current version of the manuscript is almost unchanged compared to the last version. Unfortunately, I observed that the authors have not adequately addressed most of my previous suggestions; rather, they provided justifications for not incorporating them.

Given this, I do not see the need to modify my initial assessment.

---

## [Author Response]

The following is the authors’ response to the previous reviews.

**eLife Assessment**
This neuroimaging and electrophysiology study in a small cohort of congenital cataract patients with sight recovery aims to characterize the effects of early visual deprivation on excitatory and inhibitory balance in visual cortex. While contrasting sight-recovery with visually intact controls suggested the existence of persistent alterations in Glx/GABA ratio and aperiodic EEG signals, it provided only incomplete evidence supporting claims about the effects of early deprivation itself. The reported data were considered valuable, given the rare study population. However, the small sample sizes, lack of a specific control cohort and multiple methodological limitations will likely restrict usefulness to scientists working in this particular subfield.

We thank the reviewing editors for their consideration and updated assessment of our manuscript after its first revision.

In order to assess the effects of early deprivation, we included an age-matched, normally sighted control group recruited from the same community, measured in the same scanner and laboratory. This study design is analogous to numerous studies in permanently congenitally blind humans, which typically recruited sighted controls, but hardly ever individuals with a different, e.g. late blindness history. In order to improve the specificity of our conclusions, we used a frontal cortex voxel in addition to a visual cortex voxel (MRS). Analogously, we separately analyzed occipital and frontal electrodes (EEG).

Moreover, we relate our findings in congenital cataract reversal individuals to findings in the literature on permanent congenital blindness. Note, there are, to the best of our knowledge, neither MRS nor resting-state EEG studies in individuals with permanent late blindness.

Our participants necessarily have nystagmus and low visual acuity due to their congenital deprivation phase, and the existence of nystagmus is a recruitment criterion to diagnose congenital cataracts.

It might be interesting for future studies to investigate individuals with transient late blindness. However, such a study would be ill-motivated had we not found differences between the most “extreme” of congenital visual deprivation conditions and normally sighted individuals (analogous to why earlier research on permanent blindness investigated permanent congenitally blind humans first, rather than permanently late blind humans, or both in the same study). Any result of these future work would need the reference to our study, and neither results in these additional groups would invalidate our findings.

Since all our congenital cataract reversal individuals by definition had visual impairments, we included an eyes closed condition, both in the MRS and EEG assessment. Any group effect during the eyes closed condition cannot be due to visual acuity deficits changing the bottom-up driven visual activation.

As we detail in response to review 3, our EEG analyses followed the standards in the field.

**Public Reviews:**

**Reviewer #1 (Public review):**
SummaryIn this human neuroimaging and electrophysiology study, the authors aimed to characterise effects of a period of visual deprivation in the sensitive period on excitatory and inhibitory balance in the visual cortex. They attempted to do so by comparing neurochemistry conditions ('eyes open', 'eyes closed') and resting state, and visually evoked EEG activity between ten congenital cataract patients with recovered sight (CC), and ten age-matched control participants (SC) with normal sight.First, they used magnetic resonance spectroscopy to measure in vivo neurochemistry from two locations, the primary location of interest in the visual cortex, and a control location in the frontal cortex. Such voxels are used to provide a control for the spatial specificity of any effects, because the single-voxel MRS method provides a single sampling location. Using MR-visible proxies of excitatory and inhibitory neurotransmission, Glx and GABA+ respectively, the authors report no group effects in GABA+ or Glx, no difference in the functional conditions 'eyes closed' and 'eyes open'. They found an effect of group in the ratio of Glx/GABA+ and no similar effect in the control voxel location. They then perform multiple exploratory correlations between MRS measures and visual acuity, and report a weak positive correlation between the 'eyes open' condition and visual acuity in CC participants.The same participants then took part in an EEG experiment. The authors selected two electrodes placed in the visual cortex for analysis and report a group difference in an EEG index of neural activity, the aperiodic intercept, as well as the aperiodic slope, considered a proxy for cortical inhibition. Control electrodes in the frontal region did not present with the same pattern. They report an exploratory correlation between the aperiodic intercept and Glx in one out of three EEG conditions.The authors report the difference in E/I ratio, and interpret the lower E/I ratio as representing an adaptation to visual deprivation, which would have initially caused a higher E/I ratio. Although intriguing, the strength of evidence in support of this view is not strong. Amongst the limitations are the low sample size, a critical control cohort that could provide evidence for higher E/I ratio in CC patients without recovered sight for example, and lower data quality in the control voxel. Nevertheless, the study provides a rare and valuable insight into experience-dependent plasticity in the human brain.Strengths of studyHow sensitive period experience shapes the developing brain is an enduring and important question in neuroscience. This question has been particularly difficult to investigate in humans. The authors recruited a small number of sight-recovered participants with bilateral congenital cataracts to investigate the effect of sensitive period deprivation on the balance of excitation and inhibition in the visual brain using measures of brain chemistry and brain electrophysiology. The research is novel, and the paper was interesting and well written.LimitationsLow sample size. Ten for CC and ten for SC, and further two SC participants were rejected due to lack of frontal control voxel data. The sample size limits the statistical power of the dataset and increases the likelihood of effect inflation.In the updated manuscript, the authors have provided justification for their sample size by pointing to prior studies and the inherent difficulties in recruiting individuals with bilateral congenital cataracts. Importantly, this highlights the value the study brings to the field while also acknowledging the need to replicate the effects in a larger cohort.Lack of specific control cohort. The control cohort has normal vision. The control cohort is not specific enough to distinguish between people with sight loss due to different causes and patients with congenital cataracts with co-morbidities. Further data from a more specific populations, such as patients whose cataracts have not been removed, with developmental cataracts, or congenitally blind participants, would greatly improve the interpretability of the main finding. The lack of a more specific control cohort is a major caveat that limits a conclusive interpretation of the results.In the updated version, the authors have indicated that future studies can pursue comparisons between congenital cataract participants and cohorts with later sight loss.MRS data quality differences. Data quality in the control voxel appears worse than in the visual cortex voxel. The frontal cortex MRS spectrum shows far broader linewidth than the visual cortex (Supplementary Figures). Compared to the visual voxel, the frontal cortex voxel has less defined Glx and GABA+ peaks; lower GABA+ and Glx concentrations, lower NAA SNR values; lower NAA concentrations. If the data quality is a lot worse in the FC, then small effects may not be detectable.In the updated version, the authors have added more information that informs the reader of the MRS quality differences between voxel locations. This increases the transparency of their reporting and enhances the assessment of the results.Because of the direction of the difference in E/I, the authors interpret their findings as representing signatures of sight improvement after surgery without further evidence, either within the study or from the literature. However, the literature suggests that plasticity and visual deprivation drives the E/I index up rather than down. Decreasing GABA+ is thought to facilitate experience dependent remodelling. What evidence is there that cortical inhibition increases in response to a visual cortex that is over-sensitised to due congenital cataracts? Without further experimental or literature support this interpretation remains very speculative.The updated manuscript contains key reference from non-human work to justify their interpretation.Heterogeneity in patient group. Congenital cataract (CC) patients experienced a variety of duration of visual impairment and were of different ages. They presented with co-morbidities (absorbed lens, strabismus, nystagmus). Strabismus has been associated with abnormalities in GABAergic inhibition in the visual cortex. The possible interactions with residual vision and confounds of co-morbidities are not experimentally controlled for in the correlations, and not discussed.The updated document has addressed this caveat.Multiple exploratory correlations were performed to relate MRS measures to visual acuity (shown in Supplementary Materials), and only specific ones shown in the main document. The authors describe the analysis as exploratory in the 'Methods' section. Furthermore, the correlation between visual acuity and E/I metric is weak, not corrected for multiple comparisons. The results should be presented as preliminary, as no strong conclusions can be made from them. They can provide a hypothesis to test in a future study.This has now been done throughout the document and increases the transparency of the reporting.P.16 Given the correlation of the aperiodic intercept with age ("Age negatively correlated with the aperiodic intercept across CC and SC individuals, that is, a flattening of the intercept was observed with age"), age needs to be controlled for in the correlation between neurochemistry and the aperiodic intercept. Glx has also been shown to negatively correlates with age.This caveat has been addressed in the revised manuscript.Multiple exploratory correlations were performed to relate MRS to EEG measures (shown in Supplementary Materials), and only specific ones shown in the main document. Given the multiple measures from the MRS, the correlations with the EEG measures were exploratory, as stated in the text, p.16, and in Fig.4. yet the introduction said that there was a prior hypothesis "We further hypothesized that neurotransmitter changes would relate to changes in the slope and intercept of the EEG aperiodic activity in the same subjects." It would be great if the text could be revised for consistency and the analysis described as exploratory.This has been done throughout the document and increases the transparency of the reporting.The analysis for the EEG needs to take more advantage of the available data. As far as I understand, only two electrodes were used, yet far more were available as seen in their previous study (Ossandon et al., 2023). The spatial specificity is not established. The authors could use the frontal cortex electrode (FP1, FP2) signals as a control for spatial specificity in the group effects, or even better, all available electrodes and correct for multiple comparisons. Furthermore, they could use the aperiodic intercept vs Glx in SC to evaluate the specificity of the correlation to CC.This caveat has been addressed. The authors have added frontal electrodes to their analysis, providing an essential regional control for the visual cortex location.Comments on the latest version:The authors have made reasonable adjustments to their manuscript that addressed most of my comments by adding further justification for their methodology, essential literature support, pointing out exploratory analyses, limitations and adding key control analyses. Their revised manuscript has overall improved, providing valuable information, though the evidence that supports their claims is still incomplete.

We thank the reviewer for suggesting ways to improve our manuscript and carefully reassessing our revised manuscript.

**Reviewer #2 (Public review):**
Summary:The study examined 10 congenitally blind patients who recovered vision through the surgical removal of bilateral dense cataracts, measuring neural activity and neuro chemical profiles from the visual cortex. The declared aim is to test whether restoring visual function after years of complete blindness impacts excitation/inhibition balance in the visual cortex.Strengths:The findings are undoubtedly useful for the community, as they contribute towards characterising the many ways in which this special population differs from normally sighted individuals. The combination of MRS and EEG measures is a promising strategy to estimate a fundamental physiological parameter - the balance between excitation and inhibition in the visual cortex, which animal studies show to be heavily dependent upon early visual experience. Thus, the reported results pave the way for further studies, which may use a similar approach to evaluate more patients and control groups.Weaknesses:The main methodological limitation is the lack of an appropriate comparison group or condition to delineate the effect of sight recovery (as opposed to the effect of congenital blindness). Few previous studies suggested that Excitation/Inhibition ratio in the visual cortex is increased in congenitally blind patients; the present study reports that E/I ratio decreases instead. The authors claim that this implies a change of E/I ratio following sight recovery. However, supporting this claim would require showing a shift of E/I after vs. before the sight-recovery surgery, or at least it would require comparing patients who did and did not undergo the sight-recovery surgery (as common in the field).

We thank the reviewer for suggesting ways to improve our manuscript and carefully reassessing our revised manuscript.

Since we have not been able to acquire longitudinal data with the experimental design of the present study in congenital cataract reversal individuals, we compared the MRS and EEG results of congenital cataract reversal individuals to published work in congenitally permanent blind individuals. We consider this as a resource saving approach. We think that the results of our cross-sectional study now justify the costs and enormous efforts (and time for the patients who often have to travel long distances) associated with longitudinal studies in this rare population.

There are also more technical limitations related to the correlation analyses, which are partly acknowledged in the manuscript. A bland correlation between GLX/GABA and the visual impairment is reported, but this is specific to the patients group (N=10) and would not hold across groups (the correlation is positive, predicting the lowest GLX/GABA ratio values for the sighted controls - opposite of what is found). There is also a strong correlation between GLX concentrations and the EEG power at the lowest temporal frequencies. Although this relation is intriguing, it only holds for a very specific combination of parameters (of the many tested): only with eyes open, only in the patients group.

Given the exploratory nature of the correlations, we do not base the majority of our conclusions on this analysis. There are no doubts that the reported correlations need replication; however, replication is only possible after a first report. Thus, we hope to motivate corresponding analyses in further studies.

It has to be noted that in the present study significance testing for correlations were corrected for multiple comparisons, and that some findings replicate earlier reports (e.g. effects on EEG aperiodic slope, alpha power, and correlations with chronological age).

Conclusions:The main claim of the study is that sight recovery impacts the excitation/inhibition balance in the visual cortex, estimated with MRS or through indirect EEG indices. However, due to the weaknesses outlined above, the study cannot distinguish the effects of sight recovery from those of visual deprivation. Moreover, many aspects of the results are interesting but their validation and interpretation require additional experimental work.

We interpret the group differences between individuals tested years after congenital visual deprivation and normally sighted individuals as supportive of the E/I ratio being impacted by congenital visual deprivation. In the absence of a sensitive period for the development of an E/I ratio, individuals with a transient phase of congenital blindness might have developed a visual system indistinguishable from normally sighted individuals. As we demonstrate, this is not so. Comparing the results of congenitally blind humans with those of congenitally permanently blind humans (from previous studies) allowed us to identify changes of E/I ratio, which add to those found for congenital blindness.

We thank the reviewer for the helpful comments and suggestions related to the first submission and first revision of our manuscript. We are keen to translate some of them into future studies.

**Reviewer #3 (Public review):**
This manuscript examines the impact of congenital visual deprivation on the excitatory/inhibitory (E/I) ratio in the visual cortex using Magnetic Resonance Spectroscopy (MRS) and electroencephalography (EEG) in individuals whose sight was restored. Ten individuals with reversed congenital cataracts were compared to age-matched, normally sighted controls, assessing the cortical E/I balance and its interrelationship and to visual acuity. The study reveals that the Glx/GABA ratio in the visual cortex and the intercept and aperiodic signal are significantly altered in those with a history of early visual deprivation, suggesting persistent neurophysiological changes despite visual restoration.First of all, I would like to disclose that I am not an expert in congenital visual deprivation, nor in MRS. My expertise is in EEG (particularly in the decomposition of periodic and aperiodic activity) and statistical methods.Although the authors addressed some of the concerns of the previous version, major concerns and flaws remain in terms of methodological and statistical approaches along with the (over)interpretation of the results. Specific concerns include:(1 3.1) Response to Variability in Visual DeprivationRather than listing the advantages and disadvantages of visual deprivation, I recommend providing at least a descriptive analysis of how the duration of visual deprivation influenced the measures of interest. This would enhance the depth and relevance of the discussion.

Although Review 2 and Review 3 (see below) pointed out problems in interpreting multiple correlational analyses in small samples, we addressed this request by reporting such correlations between visual deprivation history and measured EEG/MRS outcomes.

Calculating the correlation between duration of visual deprivation and behavioral or brain measures is, in fact, a common suggestion. The existence of sensitive periods, which are typically assumed to not follow a linear gradual decline of neuroplasticity, does not necessary allow predicting a correlation with duration of blindness. Daphne Maurer has additionally worked on the concept of “sleeper effects” (Maurer et al., 2007), that is, effects on the brain and behavior by early deprivation which are observed only later in life when the function/neural circuits matures.

In accordance with this reasoning, we did not observe a significant correlation between duration of visual deprivation and any of our dependent variables.

(2 3.2) Small Sample SizeThe issue of small sample size remains problematic. The justification that previous studies employed similar sample sizes does not adequately address the limitation in the current study. I strongly suggest that the correlation analyses should not feature prominently in the main manuscript or the abstract, especially if the discussion does not substantially rely on these correlations. Please also revisit the recommendations made in the section on statistical concerns.

In the revised manuscript, we explicitly mention that our sample size is not atypical for the special group investigated, but that a replication of our results in larger samples would foster their impact. We only explicitly mention correlations that survived stringent testing for multiple comparisons in the main manuscript.

Given the exploratory nature of the correlations, we have not based the majority of our claims on this analysis.

(3 3.3) Statistical ConcernsWhile I appreciate the effort of conducting an independent statistical check, it merely validates whether the reported statistical parameters, degrees of freedom (df), and p-values are consistent. However, this does not address the appropriateness of the chosen statistical methods.

We did not intend for the statcheck report to justify the methods used for statistics, which we have done in a separate section with normality and homogeneity testing (Supplementary Material S9), and references to it in the descriptions of the statistical analyses (Methods, Page 13, Lines 326-329 and Page 15, Lines 400-402).

Several points require clarification or improvement:(4) Correlation Methods: The manuscript does not specify whether the reported correlation analyses are based on Pearson or Spearman correlation.

The depicted correlations are Pearson correlations. We will add this information to the Methods.

(5) Confidence Intervals: Include confidence intervals for correlations to represent the uncertainty associated with these estimates.

We have added the confidence intervals for all measured correlations to the second revision of our manuscript.

(6) Permutation Statistics: Given the small sample size, I recommend using permutation statistics, as these are exact tests and more appropriate for small datasets.

Our study focuses on a rare population, with a sample size limited by the availability of participants. Our findings provide exploratory insights rather than make strong inferential claims. To this end, we have ensured that our analysis adheres to key statistical assumptions (Shapiro-Wilk as well as Levene’s tests, Supplementary Material S9), and reported our findings with effect sizes, appropriate caution and context.

(7) Adjusted P-Values: Ensure that reported Bonferroni corrected p-values (e.g., p > 0.999) are clearly labeled as adjusted p-values where applicable.

In the revised manuscript, we have changed Figure 4 to say ‘adjusted p,’ which we indeed reported.

(8) Figure 2CFigure 2C still lacks crucial information that the correlation between Glx/GABA ratio and visual acuity was computed solely in the control group (as described in the rebuttal letter). Why was this analysis restricted to the control group? Please provide a rationale.

Figure 2C depicts the correlation between Glx/GABA+ ratio and visual acuity in the congenital cataract reversal group, not the control group. This is mentioned in the Figure 2 legend, as well as in the main text where the figure is referred to (Page 18, Line 475).

The correlation analyses between visual acuity and MRS/EEG measures were only performed in the congenital cataract reversal group since the sighed control group comprised of individuals with vision in the normal range; thus this analyses would not make sense. Table 1 with the individual visual acuities for all participants, including the normally sighted controls, shows the low variance in the latter group.

For variables in which no apiori group differences in variance were predicted, we performed the correlation analyses across groups (see Supplementary Material S12, S15).

We have now highlighted these motivations more clearly in the Methods of the revised manuscript (Page 16, Lines 405-410).

(9 3.4) Interpretation of Aperiodic SignalRelying on previous studies to interpret the aperiodic slope as a proxy for excitation/inhibition (E/I) does not make the interpretation more robust.

How to interpret aperiodic EEG activity has been subject of extensive investigation. We cite studies which provide evidence from multiple species (monkeys, humans) and measurements (EEG, MEG, ECoG), including studies which pharmacologically manipulated E/I balance.

Whether our findings are robust, in fact, requires a replication study. Importantly, we analyzed the intercept of the aperiodic activity fit as well, and discuss results related to the intercept.

Quote:

“(3.4) Interpretation of aperiodic signal:

- Several recent papers demonstrated that the aperiodic signal measured in EEG or ECoG is related to various important aspects such as age, skull thickness, electrode impedance, as well as cognition. Thus, currently, very little is known about the underlying effects which influence the aperiodic intercept and slope. The entire interpretation of the aperiodic slope as a proxy for E/I is based on a computational model and simulation (as described in the Gao et al. paper).

Apart from the modeling work from Gao et al., multiple papers which have also been cited which used ECoG, EEG and MEG and showed concomitant changes in aperiodic activity with pharmacological manipulation of the E/I ratio (Colombo et al., 2019; Molina et al., 2020; Muthukumaraswamy & Liley, 2018). Further, several prior studies have interpreted changes in the aperiodic slope as reflective of changes in the E/I ratio, including studies of developmental groups (Favaro et al., 2023; Hill et al., 2022; McSweeney et al., 2023; Schaworonkow & Voytek, 2021) as well as patient groups (Molina et al., 2020; Ostlund et al., 2021).

- The authors further wrote: We used the slope of the aperiodic (1/f) component of the EEG spectrum as an estimate of E/I ratio (Gao et al., 2017; Medel et al., 2020; Muthukumaraswamy & Liley, 2018). This is a highly speculative interpretation with very little empirical evidence. These papers were conducted with ECoG data (mostly in animals) and mostly under anesthesia. Thus, these studies only allow an indirect interpretation by what the 1/f slope in EEG measurements is actually influenced.

Note that Muthukumaraswamy et al. (2018) used different types of pharmacological manipulations and analyzed periodic and aperiodic MEG activity in humans, in addition to monkey ECoG (Muthukumaraswamy & Liley, 2018). Further, Medel et al. (now published as Medel et al., 2023) compared EEG activity in addition to ECoG data after propofol administration. The interpretation of our results are in line with a number of recent studies in developing (Hill et al., 2022; Schaworonkow & Voytek, 2021) and special populations using EEG. As mentioned above, several prior studies have used the slope of the 1/f component/aperiodic activity as an indirect measure of the E/I ratio (Favaro et al., 2023; Hill et al., 2022; McSweeney et al., 2023; Molina et al., 2020; Ostlund et al., 2021; Schaworonkow & Voytek, 2021), including studies using scalp-recorded EEG from humans.

In the introduction of the revised manuscript, we have made more explicit that this metric is indirect (Page 3, Line 91), (additionally see Discussion, Page 24, Lines 644-645, Page 25, Lines 650-657).

While a full understanding of aperiodic activity needs to be provided, some convergent ideas have emerged. We think that our results contribute to this enterprise, since our study is, to the best of our knowledge, the first which assessed MRS measured neurotransmitter levels and EEG aperiodic activity. “

(10) Additionally, the authors state:"We cannot think of how any of the exploratory correlations between neurophysiological measures and MRS measures could be accounted for by a difference e.g. in skull thickness."(11) This could be addressed directly by including skull thickness as a covariate or visualizing it in scatterplots, for instance, by representing skull thickness as the size of the dots.

We are not aware of any study that would justify such an analysis.

Our analyses were based on previous findings in the literature.

Since to the best of our knowledge, no evidence exists that congenital cataracts go together with changes in skull thickness, and that skull thickness might selectively modulate visual cortex Glx/GABA+ but not NAA measures, we decided against following this suggestion.

Notably, the neurotransmitter concentration reported here is after tissue segmentation of the voxel region. The tissue fraction was shown to not differ between groups in the MRS voxels (Supplementary Material S4). The EEG electrode impedance was lowered to <10 kOhm in every participant (Methods, Page 13, Line 344), and preparation was identical across groups.

(12 3.5) Problems with EEG Preprocessing and AnalysisDownsampling: The decision to downsample the data to 60 Hz "to match the stimulation rate" is problematic. This choice conflates subsequent spectral analyses due to aliasing issues, as explained by the Nyquist theorem. While the authors cite prior studies (Schwenk et al., 2020; VanRullen & MacDonald, 2012) to justify this decision, these studies focused on alpha (8-12 Hz), where aliasing is less of a concern compared of analyzing aperiodic signal. Furthermore, in contrast, the current study analyzes the frequency range from 1-20 Hz, which is too narrow for interpreting the aperiodic signal as E/I. Typically, this analysis should include higher frequencies, spanning at least 1-30 Hz or even 1-45 Hz (not 20-40 Hz).

As previously mentied in the Methods (Page 15 Line 376) and the previous response, the pop_resample function used by EEGLAB applies an anti-aliasing filter, at half the resampling frequency (as per the Nyquist theorem https://eeglab.org/tutorials/05_Preprocess/resampling.html). The upper cut off of the low pass filter set by EEGlab prior to down sampling (30 Hz) is still far above the frequency of interest in the current study (1-20 Hz), thus allowing us to derive valid results.

Quote:

“- The authors downsampled the data to 60Hz to "to match the stimulation rate". What is the intention of this? Because the subsequent spectral analyses are conflated by this choice (see Nyquist theorem).

This data were collected as part of a study designed to evoke alpha activity with visual white-noise, which ranged in luminance with equal power at all frequencies from 1-60 Hz, restricted by the refresh rate of the monitor on which stimuli were presented (Pant et al., 2023). This paradigm and method was developed by VanRullen and colleagues (Schwenk et al., 2020; Vanrullen & MacDonald, 2012), wherein the analysis requires the same sampling rate between the presented frequencies and the EEG data. The downsampling function used here automatically applies an anti-aliasing filter (EEGLAB 2019) .”

Moreover, the resting-state data were not resampled to 60 Hz. We have made this clearer in the Methods of the second revision (Page 15, Line 367).

Our consistent results of group differences across all three EEG conditions, thus, exclude any possibility that they were driven by aliasing artifacts.

The expected effects of this anti-aliasing filter can be seen in the attached Author response image 1, showing an example participant’s spectrum in the 1-30 Hz range (as opposed to the 1-20 Hz plotted in the manuscript), clearly showing a 30-40 dB drop at 30 Hz. Any aliasing due to, for example, remaining line noise, would additionally be visible in this figure (as well as Figure 3) as a peak.

**Author response image 1. sa4fig1:** Power spectral density of one congenital cataract-reversal (CC) participant in the visual stimulation condition across all channels. The reduced power at 30 Hz shows the effects of the anti-aliasing filter applied by EEGLAB’s pop_resample function.

As we stated in the manuscript, and in previous reviews, so far there has been no consensus on the exact range of measuring aperiodic activity. We made a principled decision based on the literature (showing a knee in aperiodic fits of this dataset at 20 Hz) (Medel et al., 2023; Ossandón et al., 2023), data quality (possible contamination by line noise at higher frequencies) and the purpose of the visual stimulation experiment (to look at the lower frequency range by stimulating up to 60 Hz, thereby limiting us to quantifying below 30 Hz), that 1-20 Hz would be the fit range in this dataset.

Quote:

“(3) What's the underlying idea of analyzing two separate aperiodic slopes (20-40Hz and 1-19Hz). This is very unusual to compute the slope between 20-40 Hz, where the SNR is rather low.

"Ossandón et al. (2023), however, observed that in addition to the flatter slope of the aperiodic power spectrum in the high frequency range (20-40 Hz), the slope of the low frequency range (1-19 Hz) was steeper in both, congenital cataract-reversal individuals, as well as in permanently congenitally blind humans."

The present manuscript computed the slope between 1-20 Hz. Ossandón et al. as well as Medel et al. (2023) found a “knee” of the 1/f distribution at 20 Hz and describe further the motivations for computing both slope ranges. For example, Ossandón et al. used a data driven approach and compared single vs. dual fits and found that the latter fitted the data better. Additionally, they found the best fit if a knee at 20 Hz was used. We would like to point out that no standard range exists for the fitting of the 1/f component across the literature and, in fact, very different ranges have been used (Gao et al., 2017; Medel et al., 2023; Muthukumaraswamy & Liley, 2018). “

(13) Baseline Removal: Subtracting the mean activity across an epoch as a baseline removal step is inappropriate for resting-state EEG data. This preprocessing step undermines the validity of the analysis. The EEG dataset has fundamental flaws, many of which were pointed out in the previous review round but remain unaddressed. In its current form, the manuscript falls short of standards for robust EEG analysis. If I were reviewing for another journal, I would recommend rejection based on these flaws.

The baseline removal step from each epoch serves to remove the DC component of the recording and detrend the data. This is a standard preprocessing step (included as an option in preprocessing pipelines recommended by the EEGLAB toolbox, FieldTrip toolbox and MNE toolbox), additionally necessary to improve the efficacy of ICA decomposition (Groppe et al., 2009).

In the previous review round, a clarification of the baseline timing was requested, which we added. Beyond this request, there was no mention of the appropriateness of the baseline removal and/or a request to provide reasons for why it might not undermine the validity of the analysis.

Quote:

“- "Subsequently, baseline removal was conducted by subtracting the mean activity across the length of an epoch from every data point." The actual baseline time segment should be specified.

The time segment was the length of the epoch, that is, 1 second for the resting state conditions and 6.25 seconds for the visual stimulation conditions. This has been explicitly stated in the revised manuscript (Page 13, Line 354).”

Prior work in the time (not frequency) domain on event-related potential (ERP) analysis has suggested that the baselining step might cause spurious effects (Delorme, 2023) (although see (Tanner et al., 2016)). We did not perform ERP analysis at any stage. One recent study suggests spurious group differences in the 1/f signal might be driven by an inappropriate dB division baselining method (Gyurkovics et al., 2021), which we did not perform.

Any effect of our baselining procedure on the FFT spectrum would be below the 1 Hz range, which we did not analyze.

Each of the preprocessing steps in the manuscript match pipelines described and published in extensive prior work. We document how multiple aspects of our EEG results replicate prior findings (Supplementary Material S15, S18, S19), reports of other experimenters, groups and locations, validating that our results are robust.

We therefore reject the claim of methodological flaws in our EEG analyses in the strongest possible terms.

Quote:

“(3.5) Problems with EEG preprocessing and analysis:

- It seems that the authors did not identify bad channels nor address the line noise issue (even a problem if a low pass filter of below-the-line noise was applied).

As pointed out in the methods and Figure 1, we only analyzed data from two occipital channels, O1 and O2 neither of which were rejected for any participant. Channel rejection was performed for the larger dataset, published elsewhere (Ossandón et al., 2023; Pant et al., 2023). As control sites we added the frontal channels FP1 and Fp2 (see Supplementary Material S14)

Neither Ossandón et al. (2023) nor Pant et al. (2023) considered frequency ranges above 40 Hz to avoid any possible contamination with line noise. Here, we focused on activity between 0 and 20 Hz, definitely excluding line noise contaminations (Methods, Page 14, Lines 365-367). The low pass filter (FIR, 1-45 Hz) guaranteed that any spill-over effects of line noise would be restricted to frequencies just below the upper cutoff frequency.

Additionally, a prior version of the analysis used spectrum interpolation to remove line noise; the group differences remained stable (Ossandón et al., 2023). We have reported this analysis in the revised manuscript (Page 14, Lines 364-357).

Further, both groups were measured in the same lab, making line noise (~ 50 Hz) as an account for the observed group effects in the 1-20 Hz frequency range highly unlikely. Finally, any of the exploratory MRS-EEG correlations would be hard to explain if the EEG parameters would be contaminated with line noise.

- What was the percentage of segments that needed to be rejected due to the 120μV criteria? This should be reported specifically for EO & EC and controls and patients.

The mean percentage of 1 second segments rejected for each resting state condition and the percentage of 6.25 long segments rejected in each group for the visual stimulation condition have been added to the revised manuscript (Supplementary Material S10), and referred to in the Methods on Page 14, Lines 372-373.

- The authors downsampled the data to 60Hz to "to match the stimulation rate". What is the intention of this? Because the subsequent spectral analyses are conflated by this choice (see Nyquist theorem).

This data were collected as part of a study designed to evoke alpha activity with visual white-noise, which changed in luminance with equal power at all frequencies from 1-60 Hz, restricted by the refresh rate of the monitor on which stimuli were presented (Pant et al., 2023). This paradigm and method was developed by VanRullen and colleagues (Schwenk et al., 2020; VanRullen & MacDonald, 2012), wherein the analysis requires the same sampling rate between the presented frequencies and the EEG data. The downsampling function used here automatically applies an anti-aliasing filter (EEGLAB 2019) .

- "Subsequently, baseline removal was conducted by subtracting the mean activity across the length of an epoch from every data point." The actual baseline time segment should be specified.

The time segment was the length of the epoch, that is, 1 second for the resting state conditions and 6.25 seconds for the visual stimulation conditions. This has now been explicitly stated in the revised manuscript (Page 14, Lines 379-380).

- "We excluded the alpha range (8-14 Hz) for this fit to avoid biasing the results due to documented differences in alpha activity between CC and SC individuals (Bottari et al., 2016; Ossandón et al., 2023; Pant et al., 2023)." This does not really make sense, as the FOOOF algorithm first fits the 1/f slope, for which the alpha activity is not relevant.

We did not use the FOOOF algorithm/toolbox in this manuscript. As stated in the Methods, we used a 1/f fit to the 1-20 Hz spectrum in the log-log space, and subtracted this fit from the original spectrum to obtain the corrected spectrum. Given the pronounced difference in alpha power between groups (Bottari et al., 2016; Ossandón et al., 2023; Pant et al., 2023), we were concerned it might drive differences in the exponent values. Our analysis pipeline had been adapted from previous publications of our group and other labs (Ossandón et al., 2023; Voytek et al., 2015; Waschke et al., 2017).

We have conducted the analysis with and without the exclusion of the alpha range, as well as using the FOOOF toolbox both in the 1-20 Hz and 20-40 Hz ranges (Ossandón et al., 2023). The findings of a steeper slope in the 1-20 Hz range as well as lower alpha power in CC vs SC individuals remained stable. In Ossandón et al., the comparison between the piecewise fits and FOOOF fits led the authors to use the former, as it outperformed the FOOOF algorithm for their data.

- The model fits of the 1/f fitting for EO, EC, and both participant groups should be reported.

In Figure 3 of the manuscript, we depicted the mean spectra and 1/f fits for each group.

In the revised manuscript, we added the fit quality metrics (average R^2^ values > 0.91 for each group and condition) (Methods Page 15, Lines 395-396; Supplementary Material S11) and additionally show individual subjects’ fits (Supplementary Material S11). “

(14) The authors mention:"The EEG data sets reported here were part of data published earlier (Ossandón et al., 2023; Pant et al., 2023)." Thus, the statement "The group differences for the EEG assessments corresponded to those of a larger sample of CC individuals (n=38) " is a circular argument and should be avoided."The authors addressed this comment and adjusted the statement. However, I do not understand, why not the full sample published earlier (Ossandón et al., 2023) was used in the current study?

The recording of EEG resting state data stated in 2013, while MRS testing could only be set up by the second half of 2019. Moreover, not all subjects who qualify for EEG recording qualify for being scanned (e.g. due to MRI safety, claustrophobia)

References

Bottari, D., Troje, N. F., Ley, P., Hense, M., Kekunnaya, R., & Röder, B. (2016). Sight restoration after congenital blindness does not reinstate alpha oscillatory activity in humans. Scientific Reports. https://doi.org/10.1038/srep24683

Colombo, M. A., Napolitani, M., Boly, M., Gosseries, O., Casarotto, S., Rosanova, M., Brichant, J. F., Boveroux, P., Rex, S., Laureys, S., Massimini, M., Chieregato, A., & Sarasso, S. (2019). The spectral exponent of the resting EEG indexes the presence of consciousness during unresponsiveness induced by propofol, xenon, and ketamine. NeuroImage, 189(September 2018), 631–644. https://doi.org/10.1016/j.neuroimage.2019.01.024

Delorme, A. (2023). EEG is better left alone. Scientific Reports, 13(1), 2372. https://doi.org/10.1038/s41598-023-27528-0

Favaro, J., Colombo, M. A., Mikulan, E., Sartori, S., Nosadini, M., Pelizza, M. F., Rosanova, M., Sarasso, S., Massimini, M., & Toldo, I. (2023). The maturation of aperiodic EEG activity across development reveals a progressive differentiation of wakefulness from sleep. NeuroImage, 277. https://doi.org/10.1016/J.NEUROIMAGE.2023.120264

Gao, R., Peterson, E. J., & Voytek, B. (2017). Inferring synaptic excitation/inhibition balance from field potentials. NeuroImage, 158(March), 70–78. https://doi.org/10.1016/j.neuroimage.2017.06.078

Groppe, D. M., Makeig, S., & Kutas, M. (2009). Identifying reliable independent components via split-half comparisons. NeuroImage, 45(4), 1199–1211. https://doi.org/10.1016/j.neuroimage.2008.12.038

Gyurkovics, M., Clements, G. M., Low, K. A., Fabiani, M., & Gratton, G. (2021). The impact of 1/f activity and baseline correction on the results and interpretation of time-frequency analyses of EEG/MEG data: A cautionary tale. NeuroImage, 237. https://doi.org/10.1016/j.neuroimage.2021.118192

Hill, A. T., Clark, G. M., Bigelow, F. J., Lum, J. A. G., & Enticott, P. G. (2022). Periodic and aperiodic neural activity displays age-dependent changes across early-to-middle childhood. Developmental Cognitive Neuroscience, 54, 101076. https://doi.org/10.1016/J.DCN.2022.101076

Maurer, D., Mondloch, C. J., & Lewis, T. L. (2007). Sleeper effects. In Developmental Science. https://doi.org/10.1111/j.1467-7687.2007.00562.x

McSweeney, M., Morales, S., Valadez, E. A., Buzzell, G. A., Yoder, L., Fifer, W. P., Pini, N., Shuffrey, L. C., Elliott, A. J., Isler, J. R., & Fox, N. A. (2023). Age-related trends in aperiodic EEG activity and alpha oscillations during early- to middle-childhood. NeuroImage, 269, 119925. https://doi.org/10.1016/j.neuroimage.2023.119925

Medel, V., Irani, M., Crossley, N., Ossandón, T., & Boncompte, G. (2023). Complexity and 1/f slope jointly reflect brain states. Scientific Reports, 13(1), 21700. https://doi.org/10.1038/s41598-023-47316-0

Molina, J. L., Voytek, B., Thomas, M. L., Joshi, Y. B., Bhakta, S. G., Talledo, J. A., Swerdlow, N. R., & Light, G. A. (2020). Memantine Effects on Electroencephalographic Measures of Putative Excitatory/Inhibitory Balance in Schizophrenia. Biological Psychiatry: Cognitive Neuroscience and Neuroimaging, 5(6), 562–568. https://doi.org/10.1016/j.bpsc.2020.02.004

Muthukumaraswamy, S. D., & Liley, D. T. (2018). 1/F electrophysiological spectra in resting and drug-induced states can be explained by the dynamics of multiple oscillatory relaxation processes. NeuroImage, 179(November 2017), 582–595. https://doi.org/10.1016/j.neuroimage.2018.06.068

Ossandón, J. P., Stange, L., Gudi-Mindermann, H., Rimmele, J. M., Sourav, S., Bottari, D., Kekunnaya, R., & Röder, B. (2023). The development of oscillatory and aperiodic resting state activity is linked to a sensitive period in humans. NeuroImage, 275, 120171. https://doi.org/10.1016/J.NEUROIMAGE.2023.120171

Ostlund, B. D., Alperin, B. R., Drew, T., & Karalunas, S. L. (2021). Behavioral and cognitive correlates of the aperiodic (1/f-like) exponent of the EEG power spectrum in adolescents with and without ADHD. Developmental Cognitive Neuroscience, 48, 100931. https://doi.org/10.1016/j.dcn.2021.100931

Pant, R., Ossandón, J., Stange, L., Shareef, I., Kekunnaya, R., & Röder, B. (2023). Stimulus-evoked and resting-state alpha oscillations show a linked dependence on patterned visual experience for development. NeuroImage: Clinical, 103375. https://doi.org/10.1016/J.NICL.2023.103375

Schaworonkow, N., & Voytek, B. (2021). Longitudinal changes in aperiodic and periodic activity in electrophysiological recordings in the first seven months of life. Developmental Cognitive Neuroscience, 47. https://doi.org/10.1016/j.dcn.2020.100895

Schwenk, J. C. B., VanRullen, R., & Bremmer, F. (2020). Dynamics of Visual Perceptual Echoes Following Short-Term Visual Deprivation. Cerebral Cortex Communications, 1(1). https://doi.org/10.1093/TEXCOM/TGAA012

Tanner, D., Norton, J. J. S., Morgan-Short, K., & Luck, S. J. (2016). On high-pass filter artifacts (they’re real) and baseline correction (it’s a good idea) in ERP/ERMF analysis. Journal of Neuroscience Methods, 266, 166–170. https://doi.org/10.1016/j.jneumeth.2016.01.002

Vanrullen, R., & MacDonald, J. S. P. (2012). Perceptual echoes at 10 Hz in the human brain. Current Biology. https://doi.org/10.1016/j.cub.2012.03.050

Voytek, B., Kramer, M. A., Case, J., Lepage, K. Q., Tempesta, Z. R., Knight, R. T., & Gazzaley, A. (2015). Age-related changes in 1/f neural electrophysiological noise. Journal of Neuroscience, 35(38). https://doi.org/10.1523/JNEUROSCI.2332-14.2015

Waschke, L., Wöstmann, M., & Obleser, J. (2017). States and traits of neural irregularity in the age-varying human brain. Scientific Reports 2017 7:1, 7(1), 1–12. https://doi.org/10.1038/s41598-017-17766-4